# Input-to-State Stable Coupled Oscillator Networks for Closed-form Model-based Control in Latent Space

**Maximilian Stölzle**
Department of Cognitive Robotics
Delft University of Technology
M.W.Stolzle@tudelft.nl

**Cosimo Della Santina**
Department of Cognitive Robotics
Delft University of Technology
C.DellaSantina@tudelft.nl

## Abstract

Even though a variety of methods have been proposed in the literature, efficient and effective latent-space control (i.e., control in a learned low-dimensional space) of physical systems remains an open challenge. We argue that a promising avenue is to leverage powerful and well-understood closed-form strategies from control theory literature in combination with learned dynamics, such as potential-energy shaping. We identify three fundamental shortcomings in existing latent-space models that have so far prevented this powerful combination: (i) they lack the mathematical structure of a physical system, (ii) they do not inherently conserve the stability properties of the real systems, (iii) these methods do not have an invertible mapping between input and latent-space forcing. This work proposes a novel Coupled Oscillator Network (CON) model that simultaneously tackles all these issues. More specifically, (i) we show analytically that CON is a Lagrangian system - i.e., it possesses well-defined potential and kinetic energy terms. Then, (ii) we provide formal proof of global Input-to-State stability using Lyapunov arguments. Moving to the experimental side, we demonstrate that CON reaches SoA performance when learning complex nonlinear dynamics of mechanical systems directly from images. An additional methodological innovation contributing to achieving this third goal is an approximated closed-form solution for efficient integration of network dynamics, which eases efficient training. We tackle (iii) by approximating the forcing-to-input mapping with a decoder that is trained to reconstruct the input based on the encoded latent space force. Finally, we leverage these three properties and show that they enable latent-space control. We use an integral-saturated PID with potential force compensation and demonstrate high-quality performance on a soft robot using raw pixels as the only feedback information.

## 1 Introduction

Learning how the environment evolves around us from high-dimensional observations (i.e., world models [1]) is essential for achieving both artificial and physical intelligence [2]. For example, world models are required for effectively planning an artificial/robotic agent's actions in complex and unstructured environments [3]. However, learning such dynamics directly in high-dimensional observation space is usually intractable. Seminal works have shown that we can leverage autoencoders to compress the state information into a low-dimensional latent space [4, 5] in which it is much more feasible to learn the dynamics [6, 7, 8, 9, 10]. However, strong limitations still persist when it comes to using these learned models to generate low-level intelligence.

One outstanding challenge is how to perform closed-loop control in the learned latent space - i.e., how to generate control inputs based on a high dimensional sensory input such that a desired movement is generated. Prior works have explored, among other approaches, Reinforcement Learning (RL) [11, 12, 13, 14], Model Predictive Control (MPC) [7, 15, 16, 17], Linear-quadratic Regulators (LQRs) [18, 19, 20] and gradient-based optimization [21] for planning and control towards a target evolution that is given in observation space. However, all existing latent-space control strategies have

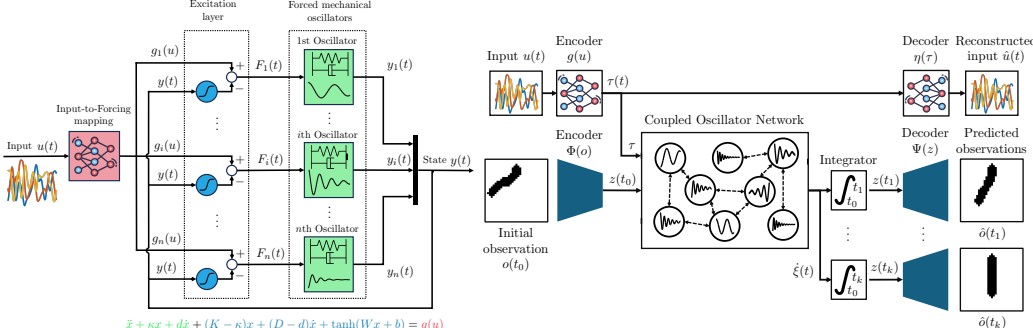

(a) Coupled Oscillator Network (CON)  (b) Learning latent-space dynamics with CON

Figure 1: **Panel (a)**: The proposed CON network consists of $n$ damped harmonic oscillators that are coupled through the neuron-like connection $\tanh(Wx+b)$ and the non-diagonal stiffness $K-k$ and damping coefficients $D-d$, respectively. The state of the network is captured by the positions $x(t)$ and velocities $\dot{x}(t)$ of the oscillators. The time-dependent input is mapped through the (possibly nonlinear) function $g(u)$ to a forcing $\tau$ acting on the oscillators. **Panel (b):** Exploiting Coupled Oscillator Networks (CONs) for learning latent dynamics from pixels: We encode the initial observation $o(t_0)$ and the input $u(t)$ into latent space where we leverage the Coupled Oscillator Network (CON) to predict future latent states. Finally, we decode both the latent-space torques $\tau(t)$ and the predicted latent states $z(t)$.

shortcomings, such as a limited planning horizon and slow control rates (MPC and gradient-based approaches), sample inefficiency (RL), or they pose a requirement for learning linear dynamics [6] (LQR), which is not even possible for systems that are inherently non-linearizable [22]. One interesting avenue is to leverage model-based control approaches, such as potential shaping [23, 24, 10], for effective and computationally efficient control in latent space [25]. For these techniques to be feasible, the dynamical model needs to fulfill four characteristics: (i) the dynamics need to have the mathematical structure of physical systems, (ii) conserve the stability properties of real systems, (iii) the latent state needs to be relatively low-dimensional, and (iv) there needs to exist a well-defined, invertible mapping between the input and the forcing in latent space. However, existing model structures that are used for learning latent dynamics [26] do not meet all of these criteria. Relevant examples are Multilayer Perceptrons (MLPs), Neural ODEs (NODEs) [27, 28], many variants of Recurrent Neural Networks (RNNs) (e.g., LSTMs [29], Gated Recurrent Units (GRUs) [30], etc.), and physics-informed neural networks (e.g., Lagrangian Neural Networks (LNNs) [31, 32, 10], Hamiltonian Neural Networks (HNNs)) [33]. For example, MLPs do not have a physical interpretation and do not provide an invertible mapping of the forcing generated by the input, NODEs are usually not easily stabilizable [34], most RNNs require a relatively high-dimensional latent space (i.e., many hidden states), and energy-shaping control approaches based on LNNs [10] do not come with any formal stability guarantees.

In recent years, oscillatory networks [35, 36, 37, 38, 39] have been shown to exhibit state-of-the-art performance on time sequence modeling tasks while being parameter-efficient, thus fulfilling our requirement (iii). Consequently, we believe that they are a promising option for control-oriented dynamics learning in latent space. Still, these models do not fulfill the remaining requirements that we have listed above. Despite being an interpretable combination of harmonic oscillators, they do not have the structure of a physical system - i.e., they do not possess a well-defined energy function. Moreover, only local stability [35, 37] has been shown, with sufficient conditions that appear to be very stringent. Finally, in addition to training an encoder that maps inputs to latent-space forcing, we propose also training a decoder that learns to reconstruct inputs based on latent-space forcing. This enables us to easily switch between inputs and forcing, which is essential when implementing control strategies.

We resolve all the above-mentioned challenges by proposing Coupled Oscillator Networks (CONs), a new formulation of a coupled oscillator network that is inherently Input-to-State Stability (ISS) stable, for learning the dynamics of physical systems and subsequently exploiting its structure for model-based control in latent space. The network consists of damped, harmonic oscillators connected through elastic springs, damping elements, and a neuron-like coupling force and can be excited by a nonlinear actuation term. We identify a transformation into a set of coordinates from which we can derive the networks' kinetic and potential energy. This allows us to leverage Lyapunov arguments [40] for proving the global asymptotic stability of the unforced system and ISS stability for the forced system under relatively mild assumptions on the network parameters. Even though we constrain the dynamics to a very specific structure, we demonstrate (a) the CON network achieves similar performance as NODEs when learning the dynamics of unactuated, mechanical systems with two

orders of magnitude fewer parameters and (b) that the proposed model achieves, for the complex task of learning the actuated, highly nonlinear dynamics of continuum soft robots [41, 42] directly from pixels, a $60\,\%$ lower prediction error than Coupled Oscillatory Recurrent Neural Network (coRNN) [35] and reaches the SoA performance across all techniques that we tested. Finally, we show some initial results that the proposed CON model is also able to learn the latent dynamics of Partial Differential Equations (PDEs), in this case containing reaction-diffusion [9, 43] dynamics.

Subsequently, we derive an approximate closed-form solution, that is, in parameter regimes in which the linear, decoupled dynamics dominate transient, more accurate than numerical integrators with comparable computational requirements and which increases training speed by 2x with a small decrease in prediction accuracy. Finally, as we can derive the system's potential energy, we can leverage potential shaping [23, 24] to derive a controller that combines an integral-saturated PID controller with a feedforward term compensating potential forces. As the feedback acts on a well-shaped potential field, tuning the feedback gains becomes very simple and out-of-the-box, and the controller exhibits a faster response time and a $26\,\%$ lower trajectory tracking Root Mean Squared Error (RMSE) than a pure feedback controller based on a latent NODE [27] model.

The proposed methodology is particularly well-suited for learning the latent dynamics of mechanical systems with continuous dynamics, dissipation, and a single, attractive equilibrium point. Examples of such systems include many soft robots, deformable objects with dominant elastic behavior, Lagrangian systems immersed in a dominant potential field, or locally other mechanical systems such as robotic manipulators, legged robots, etc. For these systems, we can fully leverage the structural prior of the proposed latent dynamics, including the integrated stability guarantees. If the system is actuated, the learned dynamics can be subsequently exploited for model-based control, as demonstrated in Sec. 5.

The code associated with this paper is available on GitHub[1].

## 2   Input-to-State Stable (ISS) Coupled Oscillator Networks (CONs)

**Formulation.**   The integral component to (coupled) oscillatory RNNs [35, 36, 37, 38] are one-dimensional, potentially damped, harmonic oscillators, which are described by their state $y_i = \begin{bmatrix} x_i & \dot{x}_i \end{bmatrix}^{\mathrm{T}} \in \mathbb{R}^2$, where $x_i$ and $\dot{x}_i$ are the position and velocity of the oscillator, respectively. Then, the oscillator's dynamics are defined by the following Equation of Motion (EOM)

$$m_i\,\ddot{x}_i(t) + d_i\,\dot{x}_i(t) + \kappa_i\,x_i(t) = F_i(t), \qquad \text{with } m_i, \kappa_i, d_i \in \mathbb{R}^+. \tag{1}$$

Here, $m_i$ is the mass, $\kappa_i$ is the stiffness, and $d_i$ is the damping coefficient of the damped harmonic oscillator. $F_i(t) \in \mathbb{R}$ is a (possibly time-dependent) external forcing term acting on the mass.

Even though the state is extremely low dimensional and the number of parameters is small, this single, damped harmonic oscillator can already exhibit a variety of (designable) behaviors: The expressions $\omega_{\mathrm{n},i} = \sqrt{\frac{\kappa_i}{m_i}}$ and $\zeta_i = \frac{d_i}{2\sqrt{\kappa_i\,m_i}}$ let us determine the natural frequency and the damping factor, respectively and allow us to design the transient behavior. For example, $\omega_{\mathrm{n},i}$ lets us isolate a spectrum of the input signal $F_i(t)$ [37] and $\zeta_i$ determines the damping regime: underdamped ($\omega_{\mathrm{n},i} < 1$), critically damped ($\omega_{\mathrm{n},i} = 1$), overdamped ($\omega_{\mathrm{n},i} > 1$). Furthermore, as (damped) harmonic oscillators are omnipresent in nature (and especially in physical systems), they have been intensively studied and are well understood (e.g., characteristics, closed-form solutions, etc.). In this work, we will exploit some of these properties and knowledge to learn stable (latent) dynamics efficiently.

By intercoupling damped harmonic oscillators, we can drastically increase the expressiveness of the dynamical system [35, 37, 38] while preserving some of the intuition and understanding we have for these systems. In this work, we propose a ISS-stable CON consisting of $n$ damped harmonic oscillators that are coupled through both linear and nonlinear terms. The networks' state is defined as $y = \begin{bmatrix} x^{\mathrm{T}} & \dot{x}^{\mathrm{T}} \end{bmatrix}^{\mathrm{T}} \in \mathbb{R}^{2n}$ and its dynamics can be formulated as a 2nd-order Ordinary Differential Equation (ODE)

$$\dot{y}(t) = \begin{bmatrix} \frac{\mathrm{d}x}{\mathrm{d}t} \\ \frac{\mathrm{d}\dot{x}}{\mathrm{d}t} \end{bmatrix} = f(y(t), u(t)) = \begin{bmatrix} \dot{x}(t) \\ g(u(t)) - K x(t) - D\,\dot{x}(t) - \tanh(W\,x(t) + b) \end{bmatrix}, \tag{2}$$

where $K, D \in \mathbb{R}^{n \times n}$ are the linear stiffness and damping matrices, respectively. The neuron-inspired term $\tanh(W\,x(t) + b)$ with $W \in \mathbb{R}^{n \times n}$, $b \in \mathbb{R}^n$ provides nonlinear coupling between the harmonic oscillators. The network is excited by the time-dependent input $u(t) \in \mathbb{R}^m$ through the possibly nonlinear mapping $g : \mathbb{R}^m \to \mathbb{R}^n$. Specifically, we consider in this work a formulation where an input-dependent matrix

---

[1]https://github.com/tud-phi/uncovering-iss-coupled-oscillator-networks-from-pixels

$B(u) \in \mathbb{R}^{n \times m}$ projects the input $u(t)$ to a time-dependent forcing on the oscillators: $\tau = g(u) = B(u)\, u$. Here $B(u)$ could, for example, be parametrized by a MLP.

We specifically designed the network architecture such that (i) the system exhibits a unique and isolated equilibrium and (ii) we can derive expressions for the kinetic and potential energies. These two features allow us to (a) prove Global Asymptotic Stability (GAS) and ISS stability using an established procedure based on strict Lyapunov arguments [44, 45], and (b) implement model-based controller based on potential shaping.

One key insight of this work is that in the coordinates $x(t), \dot{x}(t)$, we cannot derive a potential as the hyperbolic force $\tanh(W x(t) + b)$ is not symmetric. Therefore, we propose a coordinate transformation into $\mathcal{W}$-coordinates: $y_{\mathrm{w}}(t) = \begin{bmatrix} x_{\mathrm{w}}(t) \\ \dot{x}_{\mathrm{w}}(t) \end{bmatrix} = \begin{bmatrix} W\, x(t) \\ W\, \dot{x}(t) \end{bmatrix} \in \mathbb{R}^{2n}$. The coordinate transformation is valid if its Jacobian is full-rank, which is the case if $\mathrm{rank}(W) = n$. In $\mathcal{W}$-coordinates, the dynamics can be rewritten as

$$\dot{y}_{\mathrm{w}}(t) = \begin{bmatrix} \frac{\mathrm{d} x_{\mathrm{w}}}{\mathrm{d}t} \\ \frac{\mathrm{d} \dot{x}_{\mathrm{w}}}{\mathrm{d}t} \end{bmatrix} = f_{\mathrm{w}}(y(t), u(t)) = \begin{bmatrix} \dot{x}_{\mathrm{w}}(t) \\ M_{\mathrm{w}}^{-1}\left(g(u(t)) - K_{\mathrm{w}} x_{\mathrm{w}}(t) - D_{\mathrm{w}} \dot{x}_{\mathrm{w}}(t) - \tanh(x_{\mathrm{w}}(t) + b)\right) \end{bmatrix} \quad (3)$$

with $K_{\mathrm{w}} = K\, W$, $D_{\mathrm{w}} = D\, W$ and $M_{\mathrm{w}} = W^{-1}$.

A difference of this formulation compared to prior work [35, 36, 37, 38] is that (i) the forcing produced by the input term $\tau = g(u)$ is fully separated from the forcing produced by the elastic coupling terms $K_{\mathrm{w}}$, and (ii) the generalized force is symmetric, which we prove in Appendix A.1, allowing us to define a potential energy expression, which we can later on leverage for stability analysis and control.

The equilibria $\bar{y}_{\mathrm{w}} = \begin{bmatrix} \bar{x}_{\mathrm{w}}^{\mathrm{T}} & 0^{\mathrm{T}} \end{bmatrix}^{\mathrm{T}} \in \mathbb{R}^{2n}$ of the unforced network are given by the roots of the characteristic equation $\tanh(\bar{x}_{\mathrm{w}} + b) + K_{\mathrm{w}} \bar{x}_{\mathrm{w}} = 0$.

**Lemma 1.** *Let $K_{\mathrm{w}} \succ 0$. Then, the dynamics defined in* (3) *have a single, isolated equilibrium $\bar{y}_{\mathrm{w}} = \begin{bmatrix} \bar{x}_{\mathrm{w}}^{\mathrm{T}} & 0^{\mathrm{T}} \end{bmatrix}^{\mathrm{T}}$.*

*Proof.* The proof is straightforward and provided in Appendix A.2. $\qquad \square$

Next, we introduce a mapping into the tilde coordinates $\tilde{y}_{\mathrm{w}} = y_{\mathrm{w}} - \bar{y}_{\mathrm{w}}$. The residual dynamics (w.r.t. the equilibrium $\bar{y}_{\mathrm{w}}$) can now be stated as

$$\dot{\tilde{y}}_{\mathrm{w}}(t) = \tilde{f}_{\mathrm{w}}(y, u) = \begin{bmatrix} \dot{\tilde{x}}_{\mathrm{w}}(t) \\ M_{\mathrm{w}}^{-1}\left(g(u(t)) - K_{\mathrm{w}}\left(\bar{x}_{\mathrm{w}} + \tilde{x}_{\mathrm{w}}(t)\right) - D_{\mathrm{w}} \dot{\tilde{x}}_{\mathrm{w}}(t) - \tanh(\bar{x}_{\mathrm{w}} + \tilde{x}_{\mathrm{w}}(t) + b)\right) \end{bmatrix} \quad (4)$$

In the following, we will write $\|A\|$ to denote the induced norm of matrix $A$ and $\lambda_{\mathrm{m}}(A)$, $\lambda_{\mathrm{M}}(A)$ to refer to its minimum and maximum Eigenvalue respectively.

**Global Asymptotic Stability (GAS) for the unforced system.** We first consider the unforced system with $\tau = g(u) = 0$, $\forall t \in [t_0, t_\infty)$ and strive to prove global asymptotic stability [40] for the attractor $\bar{x}$. We propose a strict Lyapunov candidate with skewed level sets [45]

$$V_\mu(\tilde{y}_{\mathrm{w}}) = \frac{1}{2} \tilde{y}_{\mathrm{w}}^{\mathrm{T}} P_{\mathrm{V}} \tilde{y}_{\mathrm{w}} + \sum_{i=1}^{n} \int_0^{\tilde{x}_{\mathrm{w},i}} \tanh(\bar{x}_{\mathrm{w},i} + \sigma + b_i)\, \mathrm{d}\sigma - \sum_{i=1}^{n} \int_0^{\tilde{x}_{\mathrm{w},i}} \tanh(\bar{x}_{\mathrm{w},i} + b_i)\, \mathrm{d}\sigma,$$

$$= \frac{1}{2} \tilde{y}_{\mathrm{w}}^{\mathrm{T}} P_{\mathrm{V}} \tilde{y}_{\mathrm{w}} + \sum_{i=1}^{n} \left(\mathrm{lcosh}(\bar{x}_{\mathrm{w},i} + \tilde{x}_{\mathrm{w},i} + b_i) - \mathrm{lcosh}(\bar{x}_{\mathrm{w},i} + b_i) - \tanh(\bar{x}_{\mathrm{w},i} + b_i)\, \tilde{x}_{\mathrm{w},i}\right), \quad (5)$$

with $P_{\mathrm{V}} = \begin{bmatrix} K_{\mathrm{w}} & \mu\, M_{\mathrm{w}} \\ \mu\, M_{\mathrm{w}}^{\mathrm{T}} & M_{\mathrm{w}} \end{bmatrix} \in \mathbb{R}^{2n \times 2n}$, $\qquad \mathrm{lcosh}(\cdot) = \log(\cosh(\cdot))$, $\qquad$ and $\mu > 0$.

**Lemma 2.** *The scalar function $V_\mu(\tilde{y}_{\mathrm{w}})$ defined in* (5) *is continuously differentiable and verifies the condition $V_\mu(0) = 0$. Furthermore, let $M_{\mathrm{w}}, K_{\mathrm{w}} \succ 0$. Now, if we choose $0 < \mu < \frac{\sqrt{\lambda_{\mathrm{m}}(M_{\mathrm{w}})\, \lambda_{\mathrm{m}}(K_{\mathrm{w}})}}{\|M_{\mathrm{w}}\|} := \mu_{\mathrm{V}}$, then $V_\mu(\tilde{y}_{\mathrm{w}}) > 0 \,\forall \tilde{y}_{\mathrm{w}} \in \mathbb{R}^{2n} \setminus \{0\}$. Additionally, then $V_\mu(\tilde{y}_{\mathrm{w}})$ is radially unbounded as $\|\tilde{y}_{\mathrm{w}}\| \to \infty \Rightarrow V_\mu(\tilde{y}_{\mathrm{w}}) \to \infty$.*

*Proof.* We provide the proof in Appendix A.3 and demonstrate that the bounds on $\mu$ are required for the Lyapunov candidate to be positive-definite. $\qquad \square$

**Theorem 1.** *Let $M_{\mathrm{w}}, K_{\mathrm{w}}$ and $D_{\mathrm{w}}$ be positive definite and suppose the system be unforced: $g(u(t)) = 0$. Then, $\tilde{y}_{\mathrm{w}} = 0$ is globally asymptotically stable for the system dynamics defined* (4) *such that $\dot{V}_\mu(\tilde{y}_{\mathrm{w}}) < 0$, $\quad \forall \tilde{y}_{\mathrm{w}} \in \mathbb{R}^{2n} \setminus \{0\}$.*

*Proof.* First, we show that $\tilde{y}_{\mathrm{w}} = 0$ is an equilibrium of (4):

$$\tilde{f}_{\mathrm{w}}(0,0) = \begin{bmatrix} 0 \\ M_{\mathrm{w}}^{-1}\left(-K_{\mathrm{w}}\,\bar{x}_{\mathrm{w}} - \tanh(\bar{x}_{\mathrm{w}}+b)\right) \end{bmatrix} = \begin{bmatrix} 0 \\ M_{\mathrm{w}}^{-1}\left(-K_{\mathrm{w}}\,\bar{x}_{\mathrm{w}} + K_{\mathrm{w}}\,\bar{x}_{\mathrm{w}}\right) \end{bmatrix} = \begin{bmatrix} 0 \\ 0 \end{bmatrix} \quad (6)$$

Lemma 2 states that we can always choose $\mu$ such that (5) is strict Lyapunov function (e.g., Lipschitz continuous, zero-valued at $\tilde{y}_{\mathrm{w}} = 0$, positive definite, and radially unbounded) [40, 45]. We now evaluate the time-derivative of $\dot{V}_{\mu}(\tilde{y}_{\mathrm{w}})$ in the case of an unforced system (i.e., $g(u(t)) = 0$):

$$\dot{V}_{\mu}(\tilde{y}_{\mathrm{w}}) = \frac{\partial V_{\mu}}{\partial \tilde{y}_{\mathrm{w}}}\,\dot{\tilde{y}}_{\mathrm{w}} = \frac{\partial V_{\mu}}{\partial \tilde{y}_{\mathrm{w}}}\,\tilde{f}_{\mathrm{w}}(\tilde{y}_{\mathrm{w}}) = \tilde{y}_{\mathrm{w}}^{\mathrm{T}}\,P_{\mathrm{V}}\,\dot{\tilde{y}}_{\mathrm{w}} + \left(\tanh(\bar{x}_{\mathrm{w}}+\tilde{x}_{\mathrm{w}}+b) - \tanh(\bar{x}_{\mathrm{w}}+b)\right)^{\mathrm{T}}\,\dot{\tilde{x}}_{\mathrm{w}}$$

$$= -\tilde{y}_{\mathrm{w}}^{\mathrm{T}}\,\underbrace{\begin{bmatrix} \mu\,K_{\mathrm{w}} & \frac{1}{2}\,\mu\,D_{\mathrm{w}} \\ \frac{1}{2}\,\mu\,D_{\mathrm{w}}^{\mathrm{T}} & D_{\mathrm{w}} - \mu\,M_{\mathrm{w}} \end{bmatrix}}_{P_{\dot{\mathrm{V}}}}\,\tilde{y}_{\mathrm{w}} - \mu\left(\tanh(\bar{x}_{\mathrm{w}}+\tilde{x}_{\mathrm{w}}+b)^{\mathrm{T}} + \tilde{x}_{\mathrm{w}}^{\mathrm{T}}\,K_{\mathrm{w}}\right)\tilde{x}_{\mathrm{w}}, \quad (7)$$

$$= -\tilde{y}_{\mathrm{w}}^{\mathrm{T}}\,P_{\dot{\mathrm{V}}}\,\tilde{y}_{\mathrm{w}} - \mu\left(\tanh(\bar{x}_{\mathrm{w}}+\tilde{x}_{\mathrm{w}}+b) - \tanh(\bar{x}_{\mathrm{w}}+b)\right)^{\mathrm{T}}\,\tilde{x}_{\mathrm{w}},$$

$$\leq -\tilde{y}_{\mathrm{w}}^{\mathrm{T}}\,P_{\dot{\mathrm{V}}}\,\tilde{y}_{\mathrm{w}} \leq -\lambda_{\mathrm{m}}\left(P_{\dot{\mathrm{V}}}\right)\|\tilde{y}_{\mathrm{w}}\|_{2}^{2},$$

where we exploited the force balance at equilibrium $K_{\mathrm{w}}\,\bar{x}_{\mathrm{w}} = -\tanh(\bar{x}_{\mathrm{w}}+b)$ and Lemma 7 for defining the upper bound on $\dot{V}_{\mu}(\tilde{y}_{\mathrm{w}})$. Lemma 8 states that $P_{\dot{\mathrm{V}}} \succ 0$ for $0 < \mu < \mu_{\dot{\mathrm{V}}}$. Similarly, Lemma 2 requests that $0 < \mu < \mu_{\mathrm{V}}$. Indeed, both conditions can always be fulfilled by choosing $\mu \in (0, \min\{\mu_{\mathrm{V}}, \mu_{\dot{\mathrm{V}}}\})$. With $P_{\dot{\mathrm{V}}} \succ 0 \Leftrightarrow \lambda_{\mathrm{m}}\left(P_{\dot{\mathrm{V}}}\right) > 0$ [46], we can state that $\dot{V}_{\mu}(\tilde{y}_{\mathrm{w}}) < 0 \,\forall\, \tilde{y}_{\mathrm{w}} \in \mathbb{R}^{2n} \setminus \{0\}$ and conclude that the unforced system is globally asymptotically stable around $\tilde{y}_{\mathrm{w}} = 0$. $\square$

**Global Input-to-State Stability (ISS) for the forced system.** We now take the forcing $g(u)$ into account again and demonstrate that the system states remain proportionally bounded to the initial conditions and as a function of the supremum of the input forcing.

**Theorem 2.** *Suppose $M_{\mathrm{w}}, K_{\mathrm{w}}, D_{\mathrm{w}} \succ 0$, $0 < \theta < 1$, and that we choose $0 < \mu < \min\{\mu_{\mathrm{V}}, \mu_{\dot{\mathrm{V}}}\}$. Then, (4) is globally Input-to-State Stable (ISS) such that the solution $\tilde{y}_{\mathrm{w}}(t)$ verifies*

$$\|\tilde{y}_{\mathrm{w}}\|_2 \leq \beta\left(\|\tilde{y}_{\mathrm{w}}(t_0)\|_2, t - t_0\right) + \gamma\left(\sup_{t_0 \leq t' \leq t} \|g(u(t'))\|_2\right), \qquad \forall\, t \geq t_0 \quad (8)$$

*where $\beta(r,t) \in \mathcal{KL}$, $\gamma(r) = \sqrt{\dfrac{(1+\mu^2)\,\lambda_{\mathrm{M}}(P_{\mathrm{V}})\,r^2 + 4\,\theta\,\sqrt{n}\,\sqrt{1+\mu^2}\,\lambda_{\mathrm{m}}(P_{\dot{\mathrm{V}}})\,r}{\theta^2\,\lambda_{\mathrm{m}}(P_{\mathrm{V}})\,\lambda_{\mathrm{m}}^2(P_{\dot{\mathrm{V}}})}} \in \mathcal{K}$.*

*Proof.* The proof is provided in Appendix A.5. We first demonstrate that the ISS-Lyapunov function is bounded from both sides by $\mathcal{K}_{\infty}$ functions. Subsequently, we derive the energy dissipation of the forced system and establish attracting regions as a function of the norm of the forcing. Outside these regions, it is ensured that the system has a minimal rate of decay and, therefore, converges exponentially fast into the attracting region. $\square$

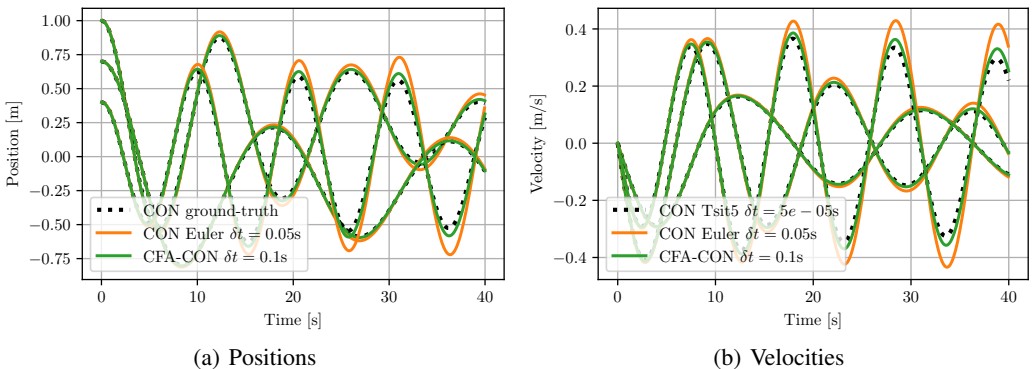

(a) Positions        (b) Velocities

Figure 2: Analysis of approximation error of CFA-CON: we compare the ground-truth solution of a $40\,\mathrm{s}$ rollout of the CON network consisting of three oscillators ($n = 3$) with the *CFA-CON* executed at a time step of $\delta t = 0.1\,\mathrm{s}$ and a solution generated by integrating the ODE at a time step of $\delta t = 0.05\,\mathrm{s}$ with the Euler method.

# 3 An approximate closed-form solution for the rollout of CON

To predict future system states, we need to integrate the ODE in Eq. (2), with the solution given by $y(t_{k+1}) = y_{t_k} + \int_{t_k}^{t_{k+1}} f(y(t'), u(t')) \, dt'$. Unfortunately, a closed-form solution for the nonlinear dynamics $f(y, u)$ does not (yet) exist. Therefore, we traditionally need to revert to (high-order) numerical ODE solvers that are computationally very expensive and introduce additional memory overhead [47]. This considerably increases the training time of models involving such continuous-time dynamics. While the computational time can be reduced by increasing the (minimum) time step of the integrator, this comes at the expense of an integration error, and we lose (part of) the theoretical guarantees and practical characteristics that the nominal ODE provides. In this work, we take an alternative approach by splitting the problem into (i) decoupled linear dynamics that can be cheaply and precisely integrated using a closed-form solution and (ii) the residual, coupled nonlinear dynamics, which we integrate numerically at a slower time scale:

$$\ddot{x}(t) = \underbrace{F - \kappa x(t) - d\,\dot{x}(t)}_{f_{\ddot{x},\mathrm{ld}}(y):\ \text{decoupled, linear dynamics}} + \underbrace{g(u(t) - (K - \kappa)x(t) - (D - d)\,\dot{x}(t) - \tanh(Wx(t) + b)}_{f_{\ddot{x},\mathrm{nld}}(y,u):\ \text{coupled, nonlinear dynamics}} \quad (9)$$

where $\kappa = \mathrm{diag}(K_{11} \ldots K_{nn})$, $d = (D_{11} \ldots D_{nn})$ are the diagonal components of the stiffness and damping matrices, respectively, and $F \in \mathbb{R}^n$ is a constant, external forcing term on the oscillators.

For a short-time-interval $\delta t$, we now approximate (9) as

$$\ddot{x}(t_k + \delta t) \approx f_{\ddot{x},\mathrm{ld}}(y(t_k + \delta t), F(t_k)) \qquad \text{with } F(t_k) = -f_{\ddot{x},\mathrm{nld}}(y(t_k), u(t_k)). \quad (10)$$

For a scalar 2$^{\mathrm{nd}}$-order, linear ODE of form $\dot{y}_i = f_{\mathrm{ld},i}(x(t'), F(t_k))$, a well-known, closed-form solution [48] exists. We exploit this characteristic by formulating the approximate solution as

$$y(t_k + \delta t) \approx f_{\mathrm{CFA-CON}}(y(t_k), u(t_k)) = y(t_k) + \int_{t_k}^{t_k + \delta t} f_{\mathrm{ld}}(y(t'), F(t_k)) \, dt' \quad (11)$$

and denote $f_{\mathrm{CFA-CON}} : \mathbb{R}^n \times \mathbb{R}^m \to \mathbb{R}^n$ as the Closed-Form Approximation of the Coupled Oscillator Network (CFA-CON) model. The implicit assumption behind (10) is that $f_{\ddot{x},\mathrm{ld}}(y) \succ f_{\ddot{x},\mathrm{nld}}(y, u)$ (i.e., the linear, decoupled dynamics dominate the nonlinear, coupled, time-varying dynamics). We refer the interested reader to Appendix B for derivation and implementation details, where we summarize the integration procedure in Algorithm 1. We also provide qualitative results for the integration accuracy in Fig. 2 and quantitative results for the integration accuracy and computational speed-up w.r.t. numerical integrators in Appendix B.

# 4 Learning control-oriented latent dynamics from pixels

We now move towards learning latent dynamical models based on CON and CFA-CON. CONs are an ideal fit for learning latent dynamics as they guarantee that the latent states stay bounded.

We assume to have access to observations in the form of images $o \in \mathbb{R}^{h_o \times w_o \times c_o}$, where $c_o$ denotes the number of channels. Please note that this could also be other high-dimensional observations such as LiDAR scans, point clouds, etc. We now leverage an encoder-decoder architecture to map these high-dimensional observations into a compressed latent space: The encoder $\Phi : \mathbb{R}^{h_o \times w_o \times c_o} \to \mathbb{R}^{n_z}$ with $n_z \ll h_o\,w_o$ identifies a low-dimensional latent representation $z \in \mathbb{R}^{n_z}$ of the images. The decoder $\Psi : \mathbb{R}^{n_z} \to \mathbb{R}^{h_o \times w_o \times c_o}$ approximates the inverse operation by reconstructing an image $\hat{o} \in \mathbb{R}^{h_o \times w_o \times c_o}$ based on the latent representation. To promote the learning of a smooth and monotonic mapping into latent space, we specifically choose to implement the autoencoder here as a $\beta$-Variational Autoencoder (VAE) [5, 49]. Instead of just statically reconstructing the image $\hat{o}(t_k)$, we are interested in predicting future observations $\hat{o}(t_{k+l})$, where $l \in 1 \ldots N$. For this, we train a 2$^{\mathrm{nd}}$-order dynamical model that is, when integrated, able to predict future latent representations $z(t_{k+l})$. This requires us to define a latent state $\xi(t) = \begin{bmatrix} z^{\mathrm{T}}(t) & \dot{z}^{\mathrm{T}}(t) \end{bmatrix}^{\mathrm{T}} \in \mathbb{R}^{2n_z}$ consisting of the latent representation and latent velocity $\dot{z}(t) \in \mathbb{R}^{n_z}$.

We now rely on CON with $n = n_z$ oscillators to provide us with the latent state derivative $\dot{\xi} = f_{\mathrm{w}}(y_{\mathrm{w}}(t), u(t))$, where we defined $\xi = y_{\mathrm{w}}$, and $z = x_{\mathrm{w}}$. To ensure stability, we make use of the Cholvesky decomposition to ensure that $M_{\mathrm{w}}, K_{\mathrm{w}}$ and $D_{\mathrm{w}}$ always remain positive definite (see Theorem 2). It is important to note that we train the encoder, decoder, and dynamical model all jointly. Please refer to Appendix C for more implementation details.

**Training.** It is important to remember that because we are using a $\beta$-VAE [5, 49], the image encoding becomes stochastic, and the encoder neural network actually outputs $\mu_z(o), 2\log(\sigma_z)(o) \in \mathbb{R}^{n_z}$. After executing the reparametrization trick as $z(t_k) \sim \mathcal{N}(\mu_z(t_k), \sigma_z^2(t_k))$, we formulate the loss function, evaluated on each trajectory consisting of $N$ time-steps, as

$$\mathcal{L} = \sum_{k=0}^{N} \left( \underbrace{\frac{\mathrm{MSE}(o(t_k), \Psi(z(t_k)))}{N+1}}_{\text{Static image reconstruction loss}} + \beta \underbrace{\frac{\mathcal{D}_{\mathrm{KL}}\left((\mu_z(t_k), \sigma_z(t_k))\right)}{N+1}}_{\text{Kullback–Leibler divergence}} \right) + \sum_{k=1}^{N} \left( \lambda_{\hat{o}} \underbrace{\frac{\mathrm{MSE}(o(t_k), \Psi(\hat{z}(t_k)))}{N}}_{\text{Dynamic image reconstruction loss}} + \lambda_{\mathrm{z}} \underbrace{\frac{\mathrm{MSE}(z(t_k), \hat{z}(t_k))}{N}}_{\text{Latent dynamics consistency loss}} \right),$$
$$(12)$$

| Model | RMSE M-SP+F [26] ↓ | RMSE S-P+F [26] ↓ | RMSE D-P+F [26] ↓ | RMSE CS ↓ | RMSE PCC-NS-2 ↓ | RMSE PCC-NS-3 ↓ |
|---|---|---|---|---|---|---|
| RNN | $0.2739 \pm 0.0057$ | $0.2378 \pm 0.0352$ | $0.1694 \pm 0.0004$ | $\mathbf{0.1011 \pm 0.0009}$ | $0.1373 \pm 0.0185$ | $0.2232 \pm 0.0075$ |
| GRU [30] | $0.0267 \pm 0.0033$ | $0.1457 \pm 0.0078$ | $0.1329 \pm 0.0005$ | $0.1125 \pm 0.0100$ | $\mathbf{0.0951 \pm 0.0021}$ | $0.2148 \pm 0.0196$ |
| coRNN [35] | $\mathbf{0.0265 \pm 0.0002}$ | $0.1333 \pm 0.0044$ | $0.1324 \pm 0.0016$ | $0.2537 \pm 0.0018$ | $0.2504 \pm 0.0899$ | $0.2474 \pm 0.0018$ |
| NODE [27] | $\mathbf{0.0264 \pm 0.0010}$ | $\mathbf{0.1260 \pm 0.0013}$ | $0.1324 \pm 0.0024$ | $0.2415 \pm 0.0021$ | $0.1867 \pm 0.0561$ | $0.3373 \pm 0.0565$ |
| MECH-NODE | $0.0328 \pm 0.0034$ | $0.1650 \pm 0.0205$ | $0.1710 \pm 0.0111$ | $0.2494 \pm 0.0028$ | $0.1035 \pm 0.0012$ | $0.1900 \pm 0.0024$ |
| CON-S (our) | $0.0303 \pm 0.0053$ | $0.1303 \pm 0.0064$ | $0.1323 \pm 0.0018$ | $0.1993 \pm 0.0646$ | $0.0996 \pm 0.0012$ | $0.1792 \pm 0.0038$ |
| CON-M (our) | $0.0303 \pm 0.0053$ | $0.1303 \pm 0.0064$ | $0.1323 \pm 0.0018$ | $0.1063 \pm 0.0027$ | $0.1008 \pm 0.0006$ | $\mathbf{0.1785 \pm 0.0023}$ |
| CFA-CON (our) | $0.0313 \pm 0.0026$ | $0.1352 \pm 0.0073$ | $\mathbf{0.1307 \pm 0.0012}$ | $0.1462 \pm 0.0211$ | $0.1124 \pm 0.0025$ | $0.1803 \pm 0.0003$ |

Table 1: Benchmarking of CON and CFA-CON at learning latent dynamics against baseline methods. The first three datasets, based on [26], contain samples of a mass-spring with friction (*M-SP + F*), a single pendulum with friction (*S-P + F*), and a double pendulum with friction (*D-P + F*) (all without system inputs). The *CS* dataset considers a continuum soft robot consisting of one segment with three constant planar strains. The *PCC-NS-2* and *PCC-NS-3* datasets contain trajectories of a continuum soft robot made of two and three piecewise constant curvature segments, respectively. We choose the latent dimensions of the models as $n_z = 4$, $n_z = 4$, and $n_z = 12$ for the *M-SP + F*, *S-P + F*, and *D-P + F* datasets, and $n_z = 8$, $n_z = 12$, and $n_z = 12$ for the *PCC-NS-2*, *PCC-NS-3*, and *CS* soft robotic datasets. We report the mean and standard deviation over three different random seeds.

| Dataset | $n_z$ | RNN | GRU [30] | coRNN [35] | NODE [27] | MECH-NODE | CON-S (our) | CON-M (our) | CFA-CON (our) |
|---|---|---|---|---|---|---|---|---|---|
| **M-SP+F** | 4 | 88 | 248 | 40 | 3368 | 3244 | **34** | **34** | **34** |
| **D-P+F** | 12 | 672 | 1968 | 348 | 4404 | 4032 | **246** | **246** | **246** |
| **PCC-NS-2** | 8 | 320 | 928 | **152** | 3856 | 3062 | 676 | 7048 | 7048 |

Table 2: Number of trainable parameters for the various latent dynamic models and the examples of the *M-SP+F* ($n_z = 4$, unactuated), *D-P+F* ($n_z = 12$, unactuated) and *PCC-NS-2* ($n_z = 8$, actuated) datasets. The number of trainable parameters for all models and datasets can be found in Appendix D.

where $\hat{z}(t_k)$ is predicted by $\hat{\xi}(t_k) = \int_{t=t_0}^{t_k} f_\xi(\xi(t'), u(t')) \, \mathrm{d}t'$, and $\xi(t_0) = \begin{bmatrix} z^\mathrm{T}(t_0) & \dot{z}^\mathrm{T}(t_0) \end{bmatrix}^\mathrm{T}$. Here, $z(t_0)$ is given by the encoder, and $\dot{z}(t_0)$ is approximated using finite differences in image-space (see Appendix C.5 for more details). $\beta, \lambda_{\hat{\sigma}}, \lambda_z \in \mathbb{R}$ are loss weights.

**Models.** We train the CON with the input-to-forcing mapping $g(u) = B(u)\, u$, where $B(u)$ is parametrized by a MLP with a hyperbolic tangent activation function applied in between layers. We report results for two variants of the CON model: for the medium-sized *CON-M* and small-sized *CON-S*, the MLP consists of five and two layers with a hidden dimension of 30 and 12, respectively. The model CFA-CON uses the same architecture as *CON-M*. We compare against several popular latent space model architectures: The NODE model uses a MLP with an hyperbolic activation functions and predicts $\dot{\xi}(t) = f_{\mathrm{NODE}}(\xi(t), u(t))$. To make the comparison fair, we parametrize the NODE's MLP in the same fashion as for *CON-M*. The *MECH-NODE* integrates prior knowledge towards learning $2^{\mathrm{nd}}$-order mechanical ODEs and, therefore, predicts $\ddot{z}(t) = f_{\mathrm{MECH-NODE}}(\xi(t), u(t))$. Furthermore, we consider multiple autoregressive models: RNN, GRU, and coRNN and let them parameterize the following transition function: $\xi(t_{k+1}) = f_{\mathrm{ar}}(\xi(t_k), u(t_k))$ As common in the relevant literature [26], we allow the autoregressive models to perform multiple time step transitions before predicting the next sample. For the autoencoder, we use a vanilla Convolutional Neural Network (CNN). More details can be found in Appendix C.

**Datasets.** We consider in total six datasets that are based on simulations of unactuated mechanical systems, and actuated continuum soft robots. The first three, mechanical dataset are based on the work of Botev et al. [26] and contain video sequences of a mass-spring system with friction (*M-SP+F*), a single pendulum with friction (*S-P+F*), and a double pendulum with friction (*D-P+F*). Continuum soft robots have theoretically infinite Degree of Freedom (DOF), evolve with highly nonlinear and often time-dependent dynamical behaviors, and are notoriously difficult to model from first principles [50]. For that reason, it is a very interesting proposition if we could learn latent-space dynamical models directly from video [51] and later leverage them for control [52]. Therefore, we generate three datasets based on the Piecewise Constant Strain (PCS) soft robot model. *CS* considers one segment with constant strain and is modeled using three configuration variables. *PCC-NS-2* and *PCC-NS-3* only consider bending deformations and contain soft robots with two and three segments, respectively. For all datasets, we render images with a resolution of 32x32px of the system's state. More information on the datasets can be found in Appendix C.1.

We tune all hyperparameters for each model and dataset separately using Optuna [53].

**Results.** **Unactuated mechanical datasets:** The results in Tab. 1 show that the *NODE* model slightly outperforms the *CON* network on the *M-SP+F* and *S-P+F* datasets. However, as the datasets do not consider system inputs, we can remove the input mapping from all models (e.g., *RNN*, *GRU*, *coRNN*, *CON*, and *CFA-CON*). With that adjustment, the *CON* network has the fewest parameters among all models, particularly two orders of magnitude less than the NODE model. Therefore, we find it very impressive that the CON network is roughly on par with the NODE model. For the *D-P+F* dataset, we can conclude that the *CFA-CON* model offers the best performance across all methods. Finally, most of the time, the *CON & CFA-CON* networks outperform the other baseline methods that have more trainable parameters. **Actuated continuum soft robot datasets:** The results in Tab. 1 show that *CON-M* matches the performance of the state-of-the-art methods

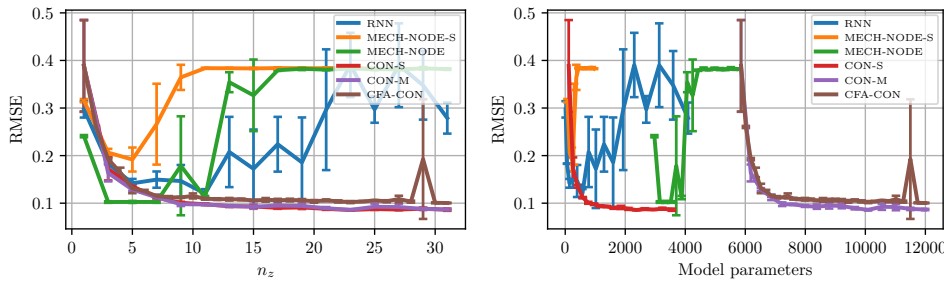

| (a) RMSE vs. latent dimension $n_z$ | (b) RMSE vs. model parameters |

Figure 3: Evaluation of prediction performance of the various models vs. the dimension of their latent representation $n_z$ and the number of trainable parameters of the dynamics model, respectively, on the *PCC-NS-2* dataset. All hyperparameters are tuned for each model separately for $n_z = 8$. The error bar denotes the standard deviation across three random seeds.

across all experiments. In the case of *PCC-NS-3*, *CON-M* even decreases the RMSE error by 6 % w.r.t. the closest baseline method (MECH-NODE). Impressively, the performance is not reduced (but instead often even improved) compared to other models that offer a much larger design space for learning the dynamics (e.g. NODE). Furthermore, *CON-S* and *CFA-CON* often only exhibit slightly lower performance than *CON-M*, even though they have significantly fewer parameters and consider an approximated solution, respectively. Supplementary results (e.g., more evaluation metrics) can be found in Appendix D. We also conduct on the *PCC-NS-2* dataset an analysis concerning the effect of the latent dimension on the performance (see Fig. 3). For this experiment, all hyperparameters were tuned for $n_z = 8$, and we observe that the CON models have a much-improved consistency and smaller variance w.r.t. the baseline methods when the latent dimensionality is increased.

The results for a dataset based on the 1$^{\text{st}}$-order reaction-diffusion PDE [9] are presented in Apx. D.1.

## 5 Exploiting the dynamic structure for latent-space control

We consider the problem setting of guiding the system towards a desired observation $o^{\text{d}}$ by providing a sequence of inputs $u(t_k)$ such that at time $t_N$, the actual observation $o(t_N)$ matches $o^{\text{d}}$. A relatively simple way would be to encode the desired observation into latent space $z^{\text{d}} = \Phi(o^{\text{d}})$ and then to design a feedback controller (e.g., PID) in latent space: $g(u) = \text{PID}(z^{\text{d}} - z(t), -\dot{z})$. Unfortunately, several challenges appear: first, it is not clear how the latent-space force $\tau = g(u)$ can be mapped back to an actual input $u(t)$ as the inverse input-to-forcing mapping $g^{-1}$ is generally not known. Furthermore, relying entirely on a PID controller has several well-known drawbacks, such as poor and slow transient behavior, steady-state errors (in case the integral gain is chosen to be zero), and instability for high proportional and integral gains. We take inspiration from potential shaping strategies [23, 24], which are widely used for effectively controlling (elastic) robots, and, therefore, combine a feedforward term compensating the latent-space potential forces with an integral-saturated, PID-like feedback term. For mapping the desired forcing $\tau$ back to an input $u(t)$, we train an forcing decoder $\eta : \mathbb{R}^n \to \mathbb{R}^m$ that approximates $g^{-1}$. Specifically, we consider here the structure $u = \eta(\tau) = E(\tau)\,\tau$, where $E \in \mathbb{R}^{m \times n}$ is parameterized by an MLP.

The latent-space control law is given by

$$\tau(t) = g(u) = \underbrace{K_{\text{w}}\,z^{\text{d}} + \tanh(z^{\text{d}} + b)}_{\substack{\text{Feedforward term:} \\ \text{compensation of potential forces}}} + \underbrace{K_{\text{p}}\,(z^{\text{d}} - z) - K_{\text{d}}\,\dot{z} + K_{\text{i}} \int_0^t \tanh(\upsilon(z^{\text{d}}(t') - z(t')))\text{d}t'}_{\text{Feedback term: P-satI-D}} \quad (13)$$

where $K_{\text{p}}, K_{\text{i}}, K_{\text{d}} \in \mathbb{R}^{n \times n}$ are the proportional, integral, and derivative control gains, respectively. As integral terms can often lead to instability when applied to nonlinear systems [54], we adopt an integral term saturation [55] with the associated dimensionless gain $\upsilon \in \mathbb{R}$, which ensures that the integral error added at each time step is bounded to the interval $(-1, 1)$. Subsequently, $\tau$ is decoded to the input as $u(t) = \eta(\tau) = E(\tau)\,\tau$. For training this decoder, we add a reconstruction loss to (12): $\mathcal{L}_{\text{u}}(t_k) = \lambda_{\text{u}}\,\text{MSE}(u(t_k), \hat{u}(t_k)) = \lambda_{\text{u}}\,\text{MSE}\,(u(t_k), \eta(g(u(t_k))))$.

**Experimental setup.** We train a CON model with two latent variables ($n_z = 2$) on the *PCC-NS-2* dataset. Analog to the input encoder mapping $B(u)$, the forcing decoder mapping $E(\tau)$ is parametrized by an MLP consisting of five layers with hidden dimension 30 and a hyperbolic tangent activation function. The *CON* model achieves an RMSE of 0.1628 on the test set. We benchmark two controllers on the simulated continuum soft robot consisting of two segments: (i) a pure *P-satI-D* controller (i.e., the feedback term in (13)) that leverages the smooth mapping into the latent representation enabled by the CON dynamic model and the $\beta$-VAE, and (ii) a *P-satI-D+FF* (i.e., (13)) that exploits the structure of the CON dynamics by compensating for the potential forces. The stable closed-loop system dynamics made the control gain tuning very easy, and we selected

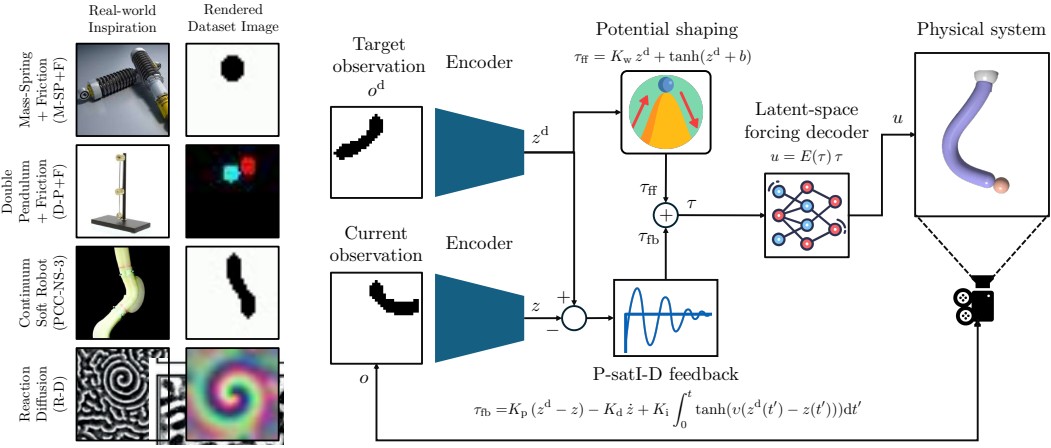

(a) Samples of used datasets      (b) Blockscheme of model-based control in latent space

Figure 4: **Panel (a):** Samples of some of the datasets used as part of the experimental verification, specifically for the results reported in Tab. 1. The real-world Reaction-Diffusion image is adopted from [43]. **Panel (b):** Model-based control in latent space by exploiting the physical structure of the CON model.

$K_\mathrm{p} = 1, K_\mathrm{i} = 2, K_\mathrm{d} = 0.02, \upsilon = 1$ for the *P-satI-D* controller and $K_\mathrm{p} = 0, K_\mathrm{i} = 2, K_\mathrm{d} = 0.05, \upsilon = 1$ for the *P-satI-D+FF* controller, respectively.

Furthermore, we compare the control performance of our model-based controllers with a baseline control strategy based on the *MECH-NODE* ($n_z = 2$) that achieves an error of $0.1104$ on the test set. First, we utilize the same P-satI-D feedback controller as for the CON model to generate the control action $\tau(t)$ in latent space. As the MECH-NODE uses an MLP to parameterize the function $\dot{\xi} = f_\xi(\xi, u)$, we cannot easily map $\tau(t)$ into an input $u(t)$. Therefore, we linearize the latent space dynamics w.r.t to the input as $f_{\xi,\mathrm{ac}}(\xi, u) = f_\xi(\xi, 0) + A(\xi)\, u$, where $A(\xi) = \frac{\partial f_\xi}{\partial u}(\xi, 0)$ is computed using autodiff. Then, $u(t) = A^\mathrm{T}(\xi)\,\tau(t)$. After tuning the control gains, we choose $K_\mathrm{p} = 0.001, K_\mathrm{i} = 0.02, K_\mathrm{d} = 1\mathrm{e}{-}5, \upsilon = 1$.

We train all models (e.g., *MECH-NODE*, *CON*) on three different random seed and choose the best model instance. Please refer to Appendix E for more details on the model selection. We generate a trajectory of 7 setpoints, where $q^\mathrm{d}(t_j) \sim \mathcal{U}(-5\pi, 5\pi)\,\mathrm{rad/m} \in \mathbb{R}^2$ is a sampled configuration of the soft robot. Then, we render an image $o^\mathrm{d}(t_j)$ that represents the target observation for the controller and encode it into latent space to retrieve $z^\mathrm{d} \in \mathbb{R}^2$. At time step $k$, we render an image $o(t_k)$ of the robot's current configuration $q(t_k)$ and encode the image. Subsequently, we evaluate the control law and apply the decoder $u(t_k) = \eta(\tau(t_k)))$, which is finally passed to the simulator that integrates the ground-truth dynamics to the next time-step $t_{k+1}$ considering the actuation $u(t_k)$. The controller runs at $100\,\mathrm{Hz}$, and we simulate the ground-truth dynamics with a *Dopri5* ODE integrator at a time-step of $1\mathrm{e}{-}5\,\mathrm{s}$.

**Results.** As an evaluation metric, we consider the RMSE between the actual and the reference trajectory. The *P-satI-D* applied to the *MECH-NODE* model (baseline) achieves an RMSE of $2.88\,\mathrm{rad/m}$ w.r.t. to the desired configuration $q^\mathrm{d}$ (but unknown to the algorithm). The *P-satI-D* CON controller, which does not exploit the learned latent dynamics for control, exhibits an RMSE of $4.08\,\mathrm{rad/m}$ w.r.t. to the desired configuration $q^\mathrm{d}$. The *P-satI-D+FF* controller exhibits an RMSE of $2.12\,\mathrm{rad/m}$ w.r.t. to the desired configuration $q^\mathrm{d}$. We also visualize the closed-loop trajectories in Fig. 5 and as sequences of stills in Apx. E. We conclude that the nicely structured latent space generated by the $\beta$-VAE allows the *P-satI-D* controller to effectively regulate the system towards the setpoint, although the response time is rather slow. The *P-satI-D+FF* controller is able to exploit the structure of the CON model through its potential shaping feedforward term. With that, *CON P-satI-D+FF* exhibits a faster response time and a $26\,\%$ lower RMSE than the *MECH-NODE P-satI-D* baseline. We provide results for the control of an actuated damped harmonic oscillator in Apx. D.

## 6   Conclusion and Limitations

**Conclusion.** In this work, we propose a new formulation for a coupled oscillator that is inherently input-to-state stable. Additionally, we identify a closed-form approximation, that is able to simulate the network dynamics more accurately compared to numerical ODE integrators with similar computational costs. When learning latent dynamics with CON, we observe that the performance is on par or slightly better compared to SoA methods such as RNNs, NODEs, etc., even though we constrained the solution space to a ISS-stable coupled oscillator structure. Furthermore, we point out that the performance of the CON models is more consistent across latent dimensions compared to the baselines and improved when not specifically tuned for a given dimension. Furthermore, as seen in Tab. 8, the closed-form approximation achieves, with the same number of model parameters, similar accuracies and double the training speed w.r.t. to the continuous-time model. Finally, we demonstrate that even a simple PID-like latent-space controller can effectively regulate the system to a setpoint. By exploiting the

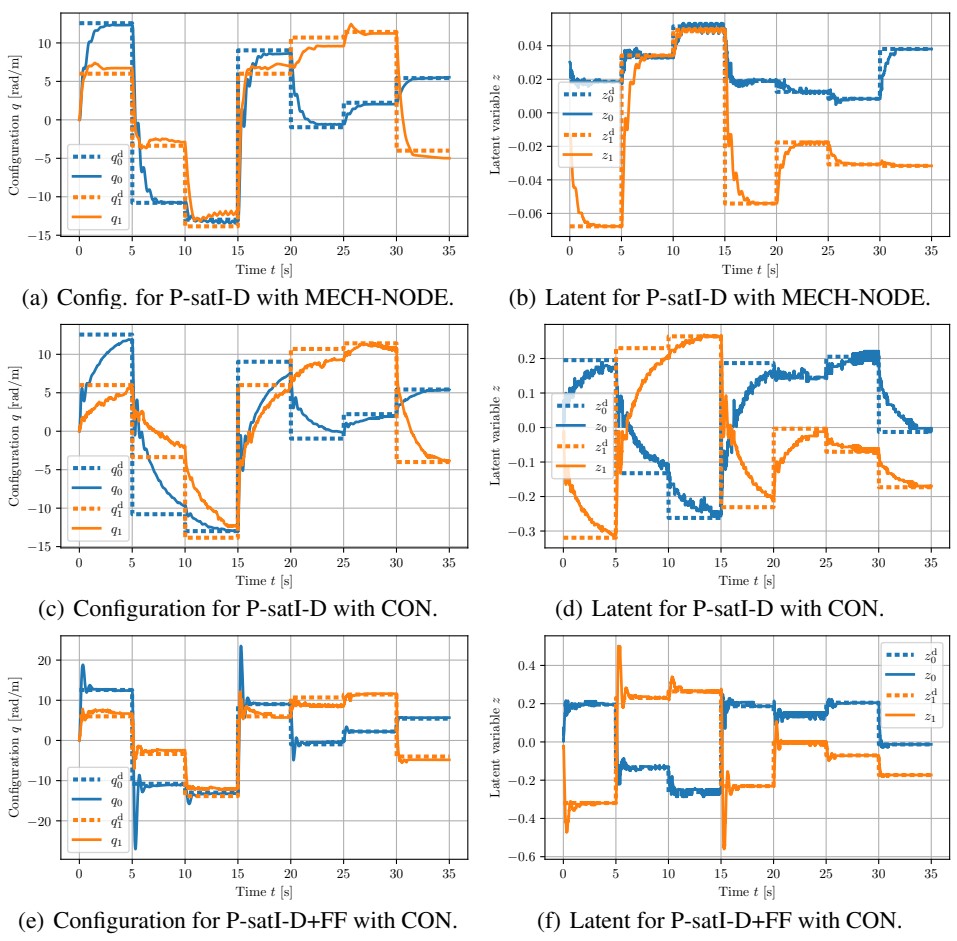

(a) Config. for P-satI-D with MECH-NODE.

(b) Latent for P-satI-D with MECH-NODE.

(c) Configuration for P-satI-D with CON.

(d) Latent for P-satI-D with CON.

(e) Configuration for P-satI-D+FF with CON.

(f) Latent for P-satI-D+FF with CON.

Figure 5: Latent-space control of a continuum soft robot (simulated using two piecewise constant curvature segments) at following a sequence of setpoints: The upper two rows show the performance of a pure P-satI-D feedback controller operating in latent space $z$ learned with the MECH-NODE and CON models, respectively. The lower row displays the results for a latent space controller based on the CON model that additionally also compensates for the learned potential forces.

network structure and compensating for potential forces, regulation performance can be greatly improved, and response time decreased by more than $55\,\%$.

**Limitations.** While we think our proposed method shows great potential and opens interesting avenues for future research, there exist certain limitations. For example, the proposed method of learning (latent) dynamics implicitly assumes that the underlying system adheres to the Markov property (e.g., the full state of the system is observable), that a system with mechanical structure can approximate it, and that it has an isolated, globally asymptotically stable equilibrium. This is, for example, the case for many mechanical systems (e.g., some continuum soft robots, deformable objects, and elastic structures) with continuous dynamics, convex elastic behavior, dissipation, and whose time-dependent effects (e.g., viscoelasticity, hysteresis) are negligible. Even if these conditions are not met globally, the method can be applied to model the local behavior around an asymptotic equilibrium point of the system (e.g., robotic manipulators, legged robots) with added stability benefits for out-of-distribution samples. Alternatively, the method could be extended to relax some of these assumptions, e.g., by allowing for multiple equilibria, zero damping, or by incorporating additional terms to capture discontinuous dynamics (e.g., stick-slip models) or period motions (e.g., limit cycles such as the Van der Pol oscillator). The proposed method might not be suitable for some physical systems, such as nonholonomic systems, partially observable systems, or systems with non-Markovian properties. Examples of such systems include mobile robots and systems with hidden states or delayed observations.

Furthermore, the approximated closed-form solution shows the best integration for situations where linear, decoupled dynamics dominate the transient. For dominant nonlinear, coupled forces, the performance of CFA-CON degrades, and it might be better to revert to numerical integration of the CON ODE. Finally, the control works exceptionally well in the setting where the latent dimension equals the input dimension. We hypothesize that this enables the method to identify a diffeomorphism between the input and the latent-space forcing. Still not investigated is how the performance could degrade if $n_z > m$ (or $n_z < m$ for that matter).

## Acknowledgments and Disclosure of Funding

This work was supported under the European Union's Horizon Europe Program from Project EMERGE - Grant Agreement No. 101070918.

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

# A    Appendix on proof of Input-to-State Stability (ISS)

In the following, we will write $\|A\|$ to denote the induced norm of matrix $A$ and $\lambda_{\mathrm{m}}(A)$, $\lambda_{\mathrm{M}}(A)$ to refer to its minimum and maximum Eigenvalue respectively.

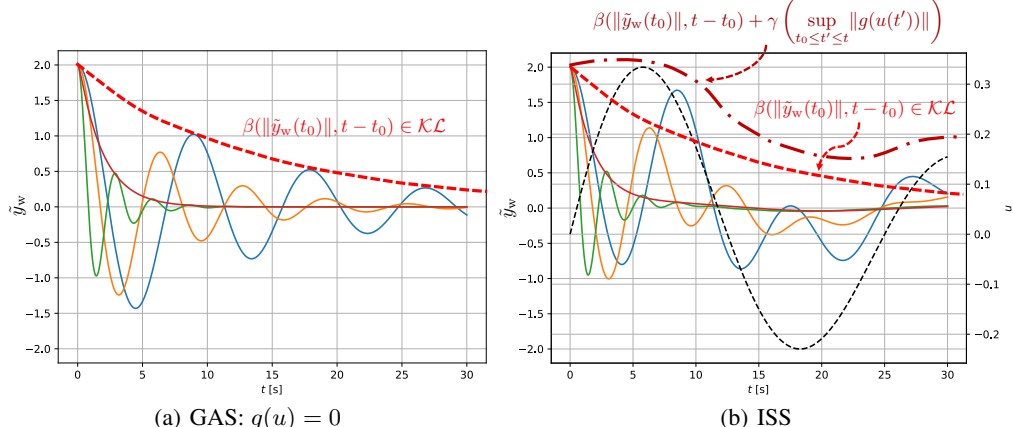

(a) GAS: $g(u) = 0$     (b) ISS

Figure 6: Illustration of global asymptotic stability for the unforced system with $g(u) = 0$ and input-to-state stability for the forced system, where the black dashed line denotes the input $u(t)$.

We introduce some expressions often used throughout this section: The gradient of $V_\mu(\tilde{y}_{\mathrm{w}})$ w.r.t. the residual coordinate $\tilde{y}_{\mathrm{w}}$ is given by

$$\frac{\partial V_\mu}{\partial \tilde{y}_{\mathrm{w}}}(\tilde{y}_{\mathrm{w}}) = P_{\mathrm{V}}\,\tilde{y}_{\mathrm{w}} + \begin{bmatrix} \tanh(\bar{x}_{\mathrm{w}} + \tilde{x}_{\mathrm{w}} + b) \\ 0^n \end{bmatrix} - \begin{bmatrix} \tanh(\bar{x}_{\mathrm{w}} + b) \\ 0^n \end{bmatrix}. \tag{14}$$

Next, the Hessian of the Lyapunov candidate can be derived as

$$H_{\mathrm{V}}(\tilde{x}_{\mathrm{w}}) = \frac{\partial^2 V_\mu}{\partial \tilde{y}_{\mathrm{w}}^2} = \begin{bmatrix} K_{\mathrm{w}} + S_{\mathrm{sech}}^2(\tilde{x}_{\mathrm{w}}) & \mu\,M_{\mathrm{w}} \\ \mu\,M_{\mathrm{w}}^{\mathrm{T}} & M_{\mathrm{w}} \end{bmatrix} \in \mathbb{R}^{2n \times 2n}, \tag{15}$$

where

$$S_{\mathrm{sech}}(\tilde{x}_{\mathrm{w}}) = \mathrm{diag}(\mathrm{sech}(\bar{x}_{\mathrm{w}} + \tilde{x}_{\mathrm{w}} + b)) \in \mathbb{R}^{n \times n} \succ 0 \quad \forall \tilde{x}_{\mathrm{w}} \in \mathbb{R}^n. \tag{16}$$

Furthermore, the Schur complement of $P_{\mathrm{V}}$ is given by

$$S_{P_{\mathrm{V}}} = M_{\mathrm{w}} - \mu^2 M_{\mathrm{w}}^{\mathrm{T}} K_{\mathrm{w}} M_{\mathrm{w}} \tag{17}$$

## A.1    Potential force and energy

**Lemma 3.** *Let $\tilde{x}_{\mathrm{w}} \in \mathbb{R}^n$ be generalized coordinates and $\bar{x}_{\mathrm{w}}, b \in \mathbb{R}^n$ constants. Then, the potential force of system* (4)

$$\tilde{f}_{\mathcal{U}_{\mathrm{w}}}(\tilde{x}_{\mathrm{w}}) = K_{\mathrm{w}}(\bar{x}_{\mathrm{w}} + \tilde{x}_{\mathrm{w}}) + \tanh(\bar{x}_{\mathrm{w}} + \tilde{x}_{\mathrm{w}} + b), \tag{18}$$

*stems from the potential*

$$\mathcal{U}_{\mathrm{w}}(\tilde{x}_{\mathrm{w}}) = \sum_{i=1}^n \int_0^{\tilde{x}_{\mathrm{w},i}} \tanh(\bar{x}_{\mathrm{w},i} + \sigma + b_i)\,\mathrm{d}\sigma - \sum_{i=1}^n \int_0^{\tilde{x}_{\mathrm{w},i}} \tanh(\bar{x}_{\mathrm{w},i} + b_i)\,\mathrm{d}\sigma \in \mathbb{R}. \tag{19}$$

*Proof.*   First, we take the derivative of $\mathcal{U}(\tilde{x}_{\mathrm{w}})$:

$$\frac{\partial \mathcal{U}_{\mathrm{w}}}{\partial \tilde{x}_{\mathrm{w}}} = K_{\mathrm{w}}(\bar{x}_{\mathrm{w}} + \tilde{x}_{\mathrm{w}}) + \tanh(\bar{x}_{\mathrm{w}} + \tilde{x}_{\mathrm{w}} + b) = \tilde{f}_{\mathcal{U}_{\mathrm{w}}}. \tag{20}$$

The Hessian of the potential is given by

$$H_{\mathcal{U}_{\mathrm{w}}}(\tilde{x}_{\mathrm{w}}) = \frac{\partial^2 \mathcal{U}_{\mathrm{w}}}{\partial \tilde{x}_{\mathrm{w}}^2} = \frac{\partial \tilde{f}_{\mathcal{U}_{\mathrm{w}}}}{\partial \tilde{x}_{\mathrm{w}}} = K_{\mathrm{w}} + S_{\mathrm{sech}}^2(\tilde{x}_{\mathrm{w}}) \in \mathbb{R}^{n \times n}. \tag{21}$$

As $K_{\mathrm{w}} \succ 0 \Rightarrow K_{\mathrm{w}} = K_{\mathrm{w}}^{\mathrm{T}}$, we can easily show that the potential force is symmetric:

$$H_{\mathcal{U}_{\mathrm{w}}}^{\mathrm{T}} = K_{\mathrm{w}}^{\mathrm{T}} + S_{\mathrm{sech}}^2(\tilde{x}_{\mathrm{w}})^{\mathrm{T}} = K_{\mathrm{w}} + S_{\mathrm{sech}}^2(\tilde{x}_{\mathrm{w}}) = H_{\mathcal{U}_{\mathrm{w}}}. \tag{22}$$

$\square$

## A.2 Proof of Lemma 1: single & isolated equilibrium

**Lemma 1 restated.** *Let $K_w \succ 0$. Then, the dynamics defined in* (3) *have a single, isolated equilibrium* $\bar{y}_w = \begin{bmatrix} \bar{x}_w^T & 0^T \end{bmatrix}^T$.

*Proof.* We regard the characteristic equation as a function: $h_{eq}(x_w) = \tanh(x_w + b) + K_w x_w$. For there to exist multiple equilibria, $h_{eq}(\bar{x}_w) = 0$ would need to be true for multiple $\bar{x}$. However, we take the partial derivative of $h_{eq}(x_w)$ w.r.t. $x_w$ and see that

$$\frac{\partial h_{eq}}{\partial x_w} = K_w + S_{sech}^2(x_w) \succ 0, \quad \forall x_w \in \mathbb{R}^n \qquad \text{with } S_{sech}(x_w) = \text{diag}(\text{sech}(x_w + b)) \in \mathbb{R}^{n \times n} \quad (23)$$

as $S_{sech}(x_w) \succ 0 \quad \forall x_w \in \mathbb{R}^n$ and $K_w \succ 0$. Therefore, $h_{eq}(x_w)$ is continuously increasing and can only cross the zero line once. □

## A.3 Proof of Lemma 2: Validity of strict Lyapunov candidate $V_\mu(\tilde{y}_w)$

**Lemma 4.** *Suppose $M_w \succ 0, K_w \succ 0$ and $0 < \mu < \frac{\sqrt{\lambda_m(M_w)\lambda_m(K_w)}}{\|M_w\|} := \mu_V$. Then $S_{P_V}$, as defined in* (17), *is positive definite.*

*Proof.* The minimum Eigenvalue of $S_{P_V}$ is bounded by

$$\lambda_m(S_{P_V}) \geq \lambda_m(M_w) - \mu^2 \|M_w^T K_w M_w\|,$$
$$\geq \lambda_m(M_w) - \mu^2 \frac{\|M_w\|^2}{\lambda_w(K_w)} \tag{24}$$

Based on the assumption $M_w \succ 0, K_w \succ 0$, we can state $\frac{\|M_w\|}{\lambda_w(K_w)} > 0$. Therefore, the critical case for the lower bound on $\lambda_m(S_{P_V})$ is $\mu = \frac{\sqrt{\lambda_m(M_w)\lambda_m(K_w)}}{\|M_w\|} := \mu_V$. Hence,

$$\lambda_m(S_{P_V}) > \lambda_m(M_w) - \frac{\lambda_m(M_w)\lambda_m(K_w)}{\|M_w\|^2} \frac{\|M_w\|^2}{\lambda_w(K_w)} = 0 \tag{25}$$

Consequently, the Eigenvalue sensitivity theorem [46] demands that $S_{P_V} \succ 0$. □

**Lemma 5.** *Let $M_w \succ 0, K_w \succ 0$, and $0 < \mu < \frac{\sqrt{\lambda_m(M_w)\lambda_m(K_w)}}{\|M_w\|} := \mu_V$. Then, it follows that $P_V \succ 0$ and $H_V(\tilde{x}_w) \succ 0 \ \forall \tilde{x}_w \in \mathbb{R}^n$.*

*Proof.* By inspecting the expressions for $P_V \succ 0$ and $H_V(\tilde{x}_w) \succ 0 \ \forall \tilde{x}_w \in \mathbb{R}^n$ in Equations (5) and (15), respectively, it can be easily seen that $H_V(\tilde{x}_w) \succeq P_V \ \forall \tilde{x}_w \in \mathbb{R}$. As Lemma 4 states that the Schur complement of $P_V$ is positive definite, it follows that $H_V(\tilde{x}_w) \succeq P_V \succ 0$. □

**Lemma 6.** *Suppose $\bar{x}_w, \tilde{x}_w, b \in \mathbb{R}^n$ and $n \in \mathbb{N}^+$. Then,*

$$h_{V,th}(\tilde{x}_w) = \sum_{i=1}^n \int_0^{\tilde{x}_{w,i}} \tanh(\bar{x}_{w,i} + \sigma + b_i)\,d\sigma - \sum_{i=1}^n \int_0^{\tilde{x}_{w,i}} \tanh(\bar{x}_{w,i} + b_i)\,d\sigma \tag{26}$$

*is a positive semi-definite function.*

*Proof.* Proving $h_{V,th}(\tilde{x}_w) \geq 0$ is equivalent to showing that the scalar function $\check{h}_{V,th}(r) = \int_0^r \tanh(\sigma + a)\,d\sigma - \int_0^r \tanh(a)\,d\sigma \geq 0 \ \forall r, a \in \mathbb{R}$, where we set $r = \tilde{x}_{w,i}$ and $a = \bar{x}_{w,i} + b_i$.

We strive to find the critical points (i.e., minimas and maximas) $\bar{r}$ of $\check{h}_{V,th}(r)$ and, for this, analyze where the first derivative of $\check{h}_{V,th}(r)$ is zero

$$\frac{\partial \check{h}_{V,th}}{\partial r}(\bar{r}) = \tanh(\bar{r} + a) - \tanh(a) = 0, \tag{27}$$

which is the case only for $\bar{r} = 0$. Next, we compute the second derivative at $\bar{r}$ as

$$\frac{\partial \check{h}_{V,th}}{\partial r}(\bar{r}) = \text{sech}^2(\bar{r}) = 1. \tag{28}$$

Thus, $\check{h}_{V,th}(r)$ is convex and its global minimum at $\bar{r} = 0$ takes the value $h_{V,th}(0) = 0$. As a result, $h_{V,th}(\tilde{x}_w)$ is also positive semi-definite. □

### A.3.1 Proof of Lemma 2:

**Lemma 2 restated.** *The scalar function $V_\mu(\tilde{y}_{\mathrm{w}})$ defined in (5) is continuously differentiable and verifies the condition $V_\mu(0) = 0$. Furthermore, let $M_{\mathrm{w}}, K_{\mathrm{w}} \succ 0$. Now, if we choose $0 < \mu < \frac{\sqrt{\lambda_{\mathrm{m}}(M_{\mathrm{w}}) \lambda_{\mathrm{m}}(K_{\mathrm{w}})}}{\|M_{\mathrm{w}}\|} := \mu_V$, then $V_\mu(\tilde{y}_{\mathrm{w}}) > 0 \; \forall \, \tilde{y}_{\mathrm{w}} \in \mathbb{R}^{2n} \setminus \{0\}$. Additionally, then $V_\mu(\tilde{y}_{\mathrm{w}})$ is radially unbounded as $\|\tilde{y}_{\mathrm{w}}\| \to \infty \Rightarrow V_\mu(\tilde{y}_{\mathrm{w}}) \to \infty$.*

*Proof.* **Step 1:** It can be easily seen that $V_\mu(\tilde{y}_{\mathrm{w}})$ in (5) is smooth and continuously differentiable.

**Step 2:** Proof that $V_\mu(0) = 0$.

$$V_\mu(0) = 0 + \sum_{i=1}^{n} \int_0^0 \tanh(\bar{y}_{\mathrm{w},i} + \sigma + b_i) \, \mathrm{d}\sigma - \sum_{i=1}^{n} \int_0^0 \tanh(\bar{y}_{\mathrm{w},i} + b_i) \, \mathrm{d}\sigma = 0. \qquad (29)$$

**Step 3:** Proof that the Lyapunov candidate is positive definite; i.e., $V_\mu(\tilde{y}_{\mathrm{w}}) > 0 \; \forall \tilde{y}_{\mathrm{w}} \in \mathbb{R}^n \setminus \{0\}$.

As the gradient of the Lyapunov candidate, as defined in (14), is zero for $\tilde{y}_{\mathrm{w}} = 0$:

$$\frac{\partial V_\mu}{\partial \tilde{y}_{\mathrm{w}}}(0) = \begin{bmatrix} \tanh(\bar{x}_{\mathrm{w}} + b) \\ 0^n \end{bmatrix} - \begin{bmatrix} \tanh(\bar{x}_{\mathrm{w}} + b) \\ 0^n \end{bmatrix} = 0, \qquad (30)$$

$\tilde{y}_{\mathrm{w}} = 0$ is a critical point of $V_\mu(\tilde{y}_{\mathrm{w}})$. According to Lemma 5, the Hessian in (15) is positive-definite [56]: $H_V(\tilde{y}_{\mathrm{w}}) \succ 0 \; \forall \tilde{y}_{\mathrm{w}} \in \mathbb{R}^{2n}$. With that, (5) is convex and its global minimum is at $\tilde{y}_{\mathrm{w}} = 0$, where $V_\mu(0) = 0$. In summary, we state $V_\mu(\tilde{y}_{\mathrm{w}}) > 0 \; \forall \tilde{y}_{\mathrm{w}} \in \mathbb{R}^n \setminus \{0\}$.

**Step 4:** Proof that the Lyapunov candidate is radially unbounded: i.e., $\|\tilde{y}_{\mathrm{w}}\| \to \infty \Rightarrow V_\mu(\tilde{y}_{\mathrm{w}}) \to \infty$. Lemma 6 is exploited for identifying a lower bound on $V_\mu(\tilde{y}_{\mathrm{w}})$:

$$\begin{aligned} V_\mu(\tilde{y}_{\mathrm{w}}) &= \frac{1}{2}\, \tilde{y}_{\mathrm{w}}^{\mathrm{T}}\, P_V \, \tilde{y}_{\mathrm{w}} + \sum_{i=1}^{n} \int_0^{\tilde{x}_{\mathrm{w},i}} \tanh(\bar{x}_{\mathrm{w},i} + \sigma + b_i) \, \mathrm{d}\sigma - \sum_{i=1}^{n} \int_0^{\tilde{x}_{\mathrm{w},i}} \tanh(\bar{x}_{\mathrm{w},i} + b_i) \, \mathrm{d}\sigma, \\ &\geq \frac{1}{2}\, \tilde{y}_{\mathrm{w}}^{\mathrm{T}}\, P_V \, \tilde{y}_{\mathrm{w}} \; \geq \; \frac{1}{2}\, \lambda_{\mathrm{m}}(P_V) \, \|\tilde{y}_{\mathrm{w}}\|^2. \end{aligned} \qquad (31)$$

Lemma 5 tells us that $P_V \succ 0$ and with that $\lambda_{\mathrm{m}}(P_V) > 0$. Therefore, if $\|\tilde{y}_{\mathrm{w}}\| \to \infty$, it also follows that $V_\mu(\tilde{y}_{\mathrm{w}}) \to \infty$. $\qquad \square$

### A.4 Proof of Theorem 1: Global asymptotic stability of unforced network

The Lemmas introduced below are used to prove Theorem 1.

**Lemma 7.** *Suppose $\bar{x}_{\mathrm{w}}, \tilde{x}_{\mathrm{w}}, b \in \mathbb{R}^n$ and $n \in \mathbb{N}^+$. Then, the function $h_{\dot{V},\mathrm{th}}(\tilde{x}_{\mathrm{w}})$ defined as*

$$h_{\dot{V},\mathrm{th}}(\tilde{x}_{\mathrm{w}}) = (\tanh(\bar{x}_{\mathrm{w}} + \tilde{x}_{\mathrm{w}} + b) - \tanh(\bar{x}_{\mathrm{w}} + b))^{\mathrm{T}} \, \tilde{x}_{\mathrm{w}}, \qquad (32)$$

*is positive semi-definite.*

*Proof.* Proving $h_{\dot{V},\mathrm{th}}(\tilde{x}_{\mathrm{w}}) \geq 0$ is equivalent to proving that the scalar function $\breve{h}_{\dot{V},\mathrm{th}}(r) = (\tanh(r + a) - \tanh(a))\, r \geq 0 \; \forall \, r, a \in \mathbb{R}$, where we set $r = \tilde{x}_{\mathrm{w},i}$ and $a = \bar{x}_{\mathrm{w},i} + b_i$. Now, we expand the hyperbolic tangent:

$$\begin{aligned} \breve{h}_{\dot{V},\mathrm{th}}(r) &= (\tanh(r + a) - \tanh(a))\, r = \left( \frac{e^{2(r+a)} - 1}{e^{2(r+a)} + 1} - \frac{e^{2a} - 1}{e^{2a} + 1} \right) r, \\ &= \frac{2\, e^{2a}\, (e^{2r} - 1)}{e^{2r+4a} + e^{2r+2a} + e^{2a} + 1}\, r \geq 0, \end{aligned} \qquad (33)$$

as the denominator $e^{2r+4a} + e^{2r+2a} + e^{2a} + 1 > 0 \; \forall \, r \in \mathbb{R}$ and as $\mathrm{sign}\left(2\, e^{2a}\, (e^{2r} - 1)\right) = \mathrm{sign}(r)$. For example, $e^{2r} - 1 \geq 0 \, \forall \, r \geq 0$. Analog, $e^{2r} - 1 < 0 \, \forall \, r < 0$. $\qquad \square$

**Lemma 8.** *Let $M_{\mathrm{w}} \succ 0$, $K_{\mathrm{w}} \succ 0$, and $D_{\mathrm{w}} \succ 0$. Also, let $\mu \in \mathbb{R}^+$ be chosen such that $0 < \mu < \frac{\lambda_{\mathrm{m}}(D_{\mathrm{w}})}{\lambda_{\mathrm{m}}(M_{\mathrm{w}}) + \frac{\|D_{\mathrm{w}}\|^2}{4\lambda_{\mathrm{m}}(K_{\mathrm{w}})}} := \mu_{\dot{V}}$. Then, the matrix $P_{\dot{V}} = \begin{bmatrix} \mu K_{\mathrm{w}} & \frac{1}{2}\mu D_{\mathrm{w}} \\ \frac{1}{2}\mu D_{\mathrm{w}}^{\mathrm{T}} & D_{\mathrm{w}} - \mu M_{\mathrm{w}} \end{bmatrix} \in \mathbb{R}^n$ is positive definite.*

*Proof.* The Schur complement of $P_{\dot{V}}$ is given by

$$S_{P_{\dot{V}}} = D_{\mathrm{w}} - \mu M_{\mathrm{w}} - \frac{1}{4}\, \mu D_{\mathrm{w}}^{\mathrm{T}}\, K_{\mathrm{w}}^{-1}\, D_{\mathrm{w}}. \qquad (34)$$

The lower bound on the smallest Eigenvalue of $S_{P_{\dot{V}}}$ can be identified as

$$\lambda_{\mathrm{m}}\left(S_{P_{\dot{V}}}\right) \geq \lambda_{\mathrm{m}}(D_{\mathrm{w}}) - \mu\,\lambda_{\mathrm{m}}(M_{\mathrm{w}}) - \mu\frac{\|D_{\mathrm{w}}\|^2}{4\,\lambda_{\mathrm{m}}(K_{\mathrm{w}})}. \tag{35}$$

As $K_{\mathrm{w}}, D_{\mathrm{w}} \succ 0$, we know that $\frac{\|D_{\mathrm{w}}\|^2}{\lambda_{\mathrm{m}}(K_{\mathrm{w}})} > 0$. Therefore, the case $\mu = \frac{\lambda_{\mathrm{m}}(D_{\mathrm{w}})}{\lambda_{\mathrm{m}}(M_{\mathrm{w}})+\frac{\|D_{\mathrm{w}}\|^2}{4\lambda_{\mathrm{m}}(K_{\mathrm{w}})}} := \mu_{\dot{V}}$ determines the lower bound on $\lambda_{\mathrm{m}}(S_{P_{\dot{V}}})$:

$$\lambda_{\mathrm{m}}\left(S_{P_{\dot{V}}}\right) > \lambda_{\mathrm{m}}(D_{\mathrm{w}}) - \mu_{\dot{V}}\left(\lambda_{\mathrm{m}}(M_{\mathrm{w}}) - \frac{\|D_{\mathrm{w}}\|^2}{4\,\lambda_{\mathrm{m}}(K_{\mathrm{w}})}\right) = 0. \tag{36}$$

We conclude, based on the Eigenvalue sensitivity theorem of symmetric matrices [46], that $S_{P_{\dot{V}}} \succ 0$ and with that $P_{\dot{V}} \succ 0$ [56]. $\qquad\square$

## A.5 Proof of Theorem 2: Proof of Input-to-State Stability (ISS)

**Lemma 9.** *Let $\bar{x}_{\mathrm{w}}, \tilde{x}_{\mathrm{w}}, b \in \mathbb{R}^n$. Then,*

$$h_{\mathrm{V},th}(\tilde{x}_{\mathrm{w}}) = \sum_{i=1}^n \int_0^{\tilde{x}_{\mathrm{w},i}} \tanh(\bar{x}_{\mathrm{w},i} + \sigma + b_i)\,\mathrm{d}\sigma - \sum_{i=1}^n \int_0^{\tilde{x}_{\mathrm{w},i}} \tanh(\bar{x}_{\mathrm{w},i} + b_i)\,\mathrm{d}\sigma \leq 2\,|\tilde{x}_{\mathrm{w}}|. \tag{37}$$

*Proof.* Proving $h_{\mathrm{V},th}(\tilde{x}_{\mathrm{w}}) \leq 2\,|\tilde{x}_{\mathrm{w}}|$ is equivalent to proving that the scalar function $\breve{h}_{\mathrm{V},th}(r) = \int_0^r \tanh(\sigma + a)\,\mathrm{d}\sigma - \int_0^r \tanh(a)\,\mathrm{d}\sigma \leq 2\,|r| \;\forall\, r, a \in \mathbb{R}$, where we set $r = \tilde{x}_{\mathrm{w},i}$ and $a = \bar{x}_{\mathrm{w},i} + b_i$. We perform the integration contained in $\breve{h}_{\mathrm{V},th}(r)$:

$$\breve{h}_{\mathrm{V},th}(r) = \int_0^r \tanh(\sigma + a)\,\mathrm{d}\sigma - \int_0^r \tanh(a)\,\mathrm{d}\sigma,$$
$$\breve{h}_{\mathrm{V},th}(r) = \log(\cosh(r + a)) - \log(\cosh(a)) - \tanh(a)\,r. \tag{38}$$

Next, we demonstrate that the slope of $2\,|r|$ is always larger than the magnitude of the slope of $\breve{h}_{\mathrm{V},th}(r)$:

$$\left|\frac{\partial \breve{h}_{\mathrm{V},th}}{\partial r}\right| = |\tanh(r + a) - \tanh(a)| < 2 = \frac{\partial}{\partial r}\,(2\,|r|). \tag{39}$$

Additionally, $\breve{h}_{\mathrm{V},th}(0) = 2\,|0| = 0$. We conclude that $\breve{h}_{\mathrm{V},th}(r) \leq 2|r| \;\forall\, r \in \mathbb{R}$ and with that, $h_{\mathrm{V},th}(\tilde{x}_{\mathrm{w}}) \leq 2\,|\tilde{x}_{\mathrm{w}}| \;\forall\, \tilde{x}_{\mathrm{w}} \in \mathbb{R}^n$. $\qquad\square$

**Lemma 10.** *Let $M_{\mathrm{w}} \succ 0$ and $K_{\mathrm{w}} \succ 0$. Then, (5) is bounded by the two scalar, class $\mathcal{K}_\infty$ functions $\alpha_1(r) = \frac{1}{2}\lambda_{\mathrm{m}}(P_{\mathrm{V}})\,r^2$ and $\alpha_2(r) = \frac{1}{2}\lambda_{\mathrm{M}}(P_{\mathrm{V}})\,r^2 + 2\sqrt{n}\,r$: $\alpha_1(\|\tilde{y}_{\mathrm{w}}\|_2^2) \leq V_\mu(\tilde{y}_{\mathrm{w}}) \leq \alpha_2(\|\tilde{y}_{\mathrm{w}}\|)_2^2$.*

*Proof.* With Lemma 2, we already showed that $V_\mu(\tilde{y}_{\mathrm{w}})$ is a Lyapunov candidate. Now, we additionally also verify the conditions for ISS-Lyapunov candidates [40].

**Step 1:** Establishing bounds on $V_\mu(\tilde{y}_{\mathrm{w}})$.
We first identify the lower bound of $V_\mu(\tilde{y}_{\mathrm{w}})$ by leveraging Lemma 6:

$$V_\mu(\tilde{y}_{\mathrm{w}}) = \frac{1}{2}\,\tilde{y}_{\mathrm{w}}^{\mathrm{T}}\,P_{\mathrm{V}}\,\tilde{y}_{\mathrm{w}} + \sum_{i=1}^n \int_0^{\tilde{x}_{\mathrm{w},i}} \tanh(\bar{x}_{\mathrm{w},i} + \sigma + b_i)\,\mathrm{d}\sigma - \sum_{i=1}^n \int_0^{\tilde{x}_{\mathrm{w},i}} \tanh(\bar{x}_{\mathrm{w},i} + b_i)\,\mathrm{d}\sigma,$$
$$= \frac{1}{2}\,\tilde{y}_{\mathrm{w}}^{\mathrm{T}}\,P_{\mathrm{V}}\,\tilde{y}_{\mathrm{w}} + h_{\mathrm{V},th}(\tilde{x}_{\mathrm{w}}), \tag{40}$$
$$\geq \frac{1}{2}\,\tilde{y}_{\mathrm{w}}^{\mathrm{T}}\,P_{\mathrm{V}}\,\tilde{y}_{\mathrm{w}} \geq \frac{1}{2}\lambda_{\mathrm{m}}(P_{\mathrm{V}})\,\|\tilde{y}_{\mathrm{w}}\|_2^2 = \alpha_1(\|\tilde{y}_{\mathrm{w}}\|_2).$$

Similarly, we derive an upper bound for $V_\mu(\tilde{y}_{\mathrm{w}})$ exploiting Lemma 9.

$$V_\mu(\tilde{y}_{\mathrm{w}}) = \frac{1}{2}\,\tilde{y}_{\mathrm{w}}^{\mathrm{T}}\,P_{\mathrm{V}}\,\tilde{y}_{\mathrm{w}} + \sum_{i=1}^n \int_0^{\tilde{x}_{\mathrm{w},i}} \tanh(\bar{x}_{\mathrm{w},i} + \sigma + b_i)\,\mathrm{d}\sigma - \sum_{i=1}^n \int_0^{\tilde{x}_{\mathrm{w},i}} \tanh(\bar{x}_{\mathrm{w},i} + b_i)\,\mathrm{d}\sigma$$
$$\leq \frac{1}{2}\lambda_{\mathrm{M}}(P_{\mathrm{V}})\,\|\tilde{y}_{\mathrm{w}}\|_2^2 + 2\,\|\tilde{x}_{\mathrm{w}}\|_1 \leq \frac{1}{2}\lambda_{\mathrm{M}}(P_{\mathrm{V}})\,\|\tilde{y}_{\mathrm{w}}\|_2^2 + 2\sqrt{n}\,\|\tilde{x}_{\mathrm{w}}\|_2 \tag{41}$$
$$\leq \frac{1}{2}\lambda_{\mathrm{M}}(P_{\mathrm{V}})\,\|\tilde{y}_{\mathrm{w}}\|_2^2 + 2\sqrt{n}\,\|\tilde{y}_{\mathrm{w}}\|_2 = \alpha_2(\|\tilde{y}_{\mathrm{w}}\|_2).$$

**Step 2:** Proof that $\alpha_1(r), \alpha_2(r)$ belong to class $\mathcal{K}_\infty$.
According to Lemma 5, $P_{\mathrm{V}} \succ 0$ and with that $\lambda_{\mathrm{m}}(P_{\mathrm{V}}) > 0$. First, we analyze the behavior of $\alpha_1(r)$: as it is strictly increasing and $\alpha_1(0) = 0$, it belongs to class $\mathcal{K}$. Furthermore, we can evaluate $\lim_{r\to\infty} \alpha_1(r) = \infty$. Therefore, $\alpha_1(r) \in \mathcal{K}_\infty$ [40]. $\alpha_2(r)$ is also strictly increasing for $r \in [0, \infty)$, $\alpha_2(0) = 0$, and it is radially unbounded as $\lim_{r\to\infty} \alpha_2(r) = \infty$. For that reason, $\alpha_2(r) \in \mathcal{K}_\infty$ as well. $\qquad\square$

### A.5.1 Proof of Theorem 2

**Theorem 2 restated.** *Suppose* $M_{\mathrm{w}}, K_{\mathrm{w}}, D_{\mathrm{w}} \succ 0$, $0 < \theta < 1$, *and that we choose* $0 < \mu < \min\{\mu_{\mathrm{V}}, \mu_{\dot{\mathrm{V}}}\}\}$. *Then,* (4) *is globally Input-to-State Stable (ISS) such that the solution* $\tilde{y}_{\mathrm{w}}(t)$ *verifies*

$$\|\tilde{y}_{\mathrm{w}}\|_2 \leq \beta\left(\|\tilde{y}_{\mathrm{w}}(t_0)\|_2, t - t_0\right) + \gamma\left(\sup_{t_0 \leq \tau \leq t} \|g(u(\tau))\|_2\right), \qquad \forall\, t \geq t_0 \tag{42}$$

*where* $\beta(r, t) \in \mathcal{KL}$, $\gamma(r) = \sqrt{\frac{(1+\mu^2)\,\lambda_{\mathrm{M}}(P_{\mathrm{V}})\,r^2 + 4\,\theta\,\sqrt{n}\,\sqrt{1+\mu^2}\,\lambda_{\mathrm{m}}(P_{\dot{\mathrm{V}}})\,r}{\theta^2\,\lambda_{\mathrm{m}}(P_{\mathrm{V}})\,\lambda_{\mathrm{m}}^2(P_{\dot{\mathrm{V}}})}} \in \mathcal{K}$.

*Proof.* **Step 1:** Bounds on ISS-Lyapunov candidate.
Lemma 10 provides the $\mathcal{K}_\infty$ functions $\alpha_1(r) = \frac{1}{2}\lambda_{\mathrm{m}}(P_{\mathrm{V}})\,r^2$ and $\alpha_2(r) = \frac{1}{2}\lambda_{\mathrm{M}}(P_{\mathrm{V}})\,r^2 + 2\sqrt{n}\,r$ such that $\alpha_1(\|\tilde{y}_{\mathrm{w}}\|_2^2) \leq V_\mu(\tilde{y}_{\mathrm{w}}) \leq \alpha_2(\|\tilde{y}_{\mathrm{w}}\|)_2^2$.

**Step 2:** Minimum energy dissipation.
Let $0 < \mu < \min\{\mu_{\mathrm{V}}, \mu_{\dot{\mathrm{V}}}\}\}$ as in the proof of Theorem 1. We compute the input-dependent time-derivative of the ISS Lyapunov candidate. We do not repeat the derivations already made as part of (7) (e.g., exploiting Lemmas 7 and 8).

$$\begin{aligned}
\dot{V}_\mu(\tilde{y}_{\mathrm{w}}, u(t)) = &-\tilde{y}_{\mathrm{w}}^{\mathrm{T}}\,P_{\dot{\mathrm{V}}}\,\tilde{y}_{\mathrm{w}} - \mu\,(\tanh(\bar{x}_{\mathrm{w}} + \tilde{x}_{\mathrm{w}} + b) - \tanh(\bar{x}_{\mathrm{w}} + b))^{\mathrm{T}}\,\tilde{x}_{\mathrm{w}} + \tilde{y}_{\mathrm{w}}^{\mathrm{T}}\begin{bmatrix} \mu\,g(u(t)) \\ g(u(t)) \end{bmatrix}, \\
&\leq -\lambda_{\mathrm{m}}\left(P_{\dot{\mathrm{V}}}\right)\|\tilde{y}_{\mathrm{w}}\|_2^2 + \left\|\tilde{y}_{\mathrm{w}}^{\mathrm{T}}\begin{bmatrix} \mu\,g(u(t)) \\ g(u(t)) \end{bmatrix}\right\|_1, \\
&\leq -\lambda_{\mathrm{m}}\left(P_{\dot{\mathrm{V}}}\right)\|\tilde{y}_{\mathrm{w}}\|_2^2 + \|\tilde{y}_{\mathrm{w}}\|_2\left\|\begin{bmatrix} \mu\,g(u(t)) \\ g(u(t)) \end{bmatrix}\right\|_2, \\
&\leq -\lambda_{\mathrm{m}}\left(P_{\dot{\mathrm{V}}}\right)\|\tilde{y}_{\mathrm{w}}\|_2^2 + \sqrt{1+\mu^2}\,\|\tilde{y}_{\mathrm{w}}\|_2\|g(u(t))\|_2,
\end{aligned} \tag{43}$$

where we leveraged Hölder's inequality. We choose $\theta$ such that $0 < \theta < 1$. As a consequence,

$$\dot{V}_\mu(\tilde{y}_{\mathrm{w}}, u(t)) \leq -(1-\theta)\,\lambda_{\mathrm{m}}\left(P_{\dot{\mathrm{V}}}\right)\|\tilde{y}_{\mathrm{w}}\|_2^2, \qquad \forall\,\|\tilde{y}_{\mathrm{w}}\|_2 \geq \frac{\sqrt{1+\mu^2}}{\theta\,\lambda_{\mathrm{m}}\left(P_{\dot{\mathrm{V}}}\right)}\|g(u(t))\|_2 > 0. \tag{44}$$

We define

$$\alpha_3(r) = (1-\theta)\,\lambda_{\mathrm{m}}\left(P_{\dot{\mathrm{V}}}\right)\,r^2, \qquad \text{and}\quad \rho(r) = \frac{\sqrt{1+\mu^2}}{\theta\,\lambda_{\mathrm{m}}\left(P_{\dot{\mathrm{V}}}\right)}\,r. \tag{45}$$

Lemma 8 shows that $\lambda_{\mathrm{m}}\left(P_{\dot{\mathrm{V}}}\right) > 0$. Therefore, $\alpha_3(r)$ is a continuous positive function. Furthermore, as $\mu > 0$, $\rho(r)$ is a strictly increasing for $r \in [0, \infty)$. Additionally with $\rho(0) = 0$ verified, it can be stated that $\rho(r)$ belongs to class $\mathcal{K}$ [40]. We conclude that

$$\dot{V}_\mu(\tilde{y}_{\mathrm{w}}, u(t)) \leq -\alpha_3\left(\|\tilde{y}_{\mathrm{w}}\|_2^2\right), \qquad \forall\,\|\tilde{y}_{\mathrm{w}}\|_2 \geq \rho\left(\|g(u(t))\|_2\right) > 0. \tag{46}$$

**Step 3:** Conclusions.

As a result of Steps 1 and 2, the system is input-to-state stable, and with that, the solution $\tilde{y}_t$ satisfies [40]

$$\|\tilde{y}_{\mathrm{w}}\|_2 \leq \beta\left(\|\tilde{y}_{\mathrm{w}}(t_0)\|_2, t - t_0\right) + \gamma\left(\sup_{t_0 \leq t' \leq t} \|g(u(t'))\|_2\right), \tag{47}$$

with

$$\gamma(r) = \alpha_1^{-1} \circ \alpha_2 \circ \rho(r) = \sqrt{\frac{(1+\mu^2)\,\lambda_{\mathrm{M}}(P_{\mathrm{V}})\,r^2 + 4\,\theta\,\sqrt{n}\,\sqrt{1+\mu^2}\,\lambda_{\mathrm{m}}(P_{\dot{\mathrm{V}}})\,r}{\theta^2\,\lambda_{\mathrm{m}}(P_{\mathrm{V}})\,\lambda_{\mathrm{m}}^2(P_{\dot{\mathrm{V}}})}}. \tag{48}$$

Indeed, based on Theorem 1 and the associated proof, we can easily verify that $\gamma(r)$ is strictly increasing for $r \in [0, \infty)$ and that $\gamma(0) = 0$. As a consequence, $\gamma(r) \in \mathcal{K}$ [40]. $\square$

# B  Appendix on an approximate closed-form solution for coupled oscillator networks

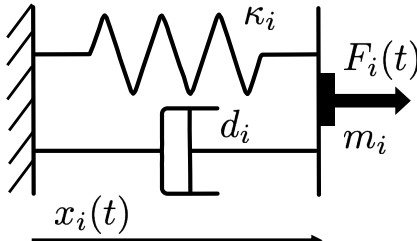

(a) Parameters of a harmonic oscillator

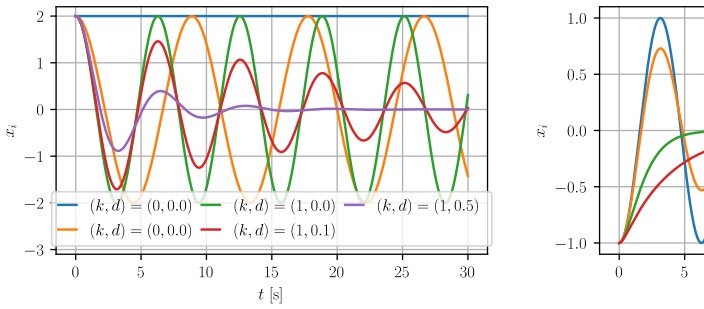

(b) Time evolution of a harmonic oscillator

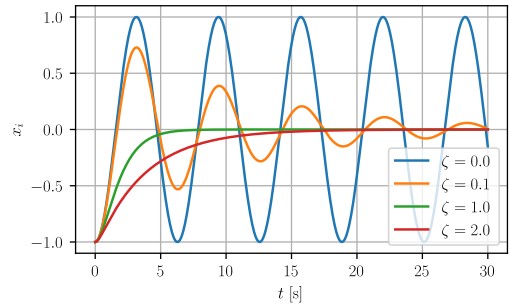

(c) Damping regimes of a harmonic oscillator

Figure 7: **Panel 7(a)**: Parameters of a forced and damped harmonic oscillator: $m_i \in \mathbb{R}^+$ denotes the mass, $\kappa_i \in \mathbb{R}^+$ the stiffness, and $k_i \in \mathbb{R}^+$ the damping coefficient. The position and velocity of the oscillator are measured as $x_i(t) \in \mathbb{R}$ and $\dot{x}_i(t) \in \mathbb{R}$, respectively. The oscillator can be excited by the (potentially time-varying) external forcing $F_i(t) \in \mathbb{R}$. **Panel 7(b):** Time evolution of a 1D harmonic oscillator for different values of $\kappa_i$, $d_i$, all in the undamped or underdamped regime. **Panel 7(c):** The four damping regimes of a harmonic oscillator: undamped ($\zeta = 0$), underdamped ($0 < \zeta < 1$), critically damped ($\zeta = 1$), and overdamped ($zeta > 1$).

## B.1  Closed-form solution to a forced harmonic oscillator

As introduced in (1), we consider the linear dynamics of a 1D forced harmonic oscillator with state $y_i = \begin{bmatrix} x_i & \dot{x}_i \end{bmatrix} \in \mathbb{R}^2$

$$\dot{y}_i = \begin{bmatrix} \frac{\mathrm{d}x_i}{\mathrm{d}t} \\ \frac{\mathrm{d}\dot{x}_i}{\mathrm{d}t} \end{bmatrix} = f_{\mathrm{ld},i}(y_i, F_i) = \begin{bmatrix} \dot{x}_i \\ F_i(t) - \kappa_i\, x_i(t) - d_i\, \dot{x}_i(t) \end{bmatrix}, \tag{49}$$

where $F_i(t) \in \mathbb{R}$ is the externally applied force acting on the oscillator.

The characteristic equation for the unforced dynamics (i.e., $F_i(t) = 0$) can be stated as [48]

$$\lambda^2 + 2\,\zeta_i\,\omega_{\mathrm{n},i}\,\lambda + \omega_{\mathrm{n},i}^2 = 0, \qquad \text{with the solutions } \lambda_{1,2} = -\zeta_i\,\omega_{\mathrm{n},i} \pm \omega_{\mathrm{n},i}\sqrt{\zeta_i^2 - 1}, \tag{50}$$

where $\omega_{\mathrm{n},i} = \sqrt{\kappa_i}$ and $\zeta_i = \frac{d_i}{2\sqrt{\kappa_i}}$ are the natural frequency and the damping factor of the $i$th homogeneous oscillator, respectively. This harmonic oscillator exhibits three regimes: underdamped ($\zeta_i < 1$), critically damped ($\zeta_i = 1$), and overdamped regime ($\zeta_i > 1$).

We approximate the forcing using the Heavyside function $H(t)$: $F_i(t) = F_i(t_k)\, H(t)$, where $F_i(t_k)$ is the constant external forcing as computed by (10). The solution for $\zeta_i \neq 1$ is given by [48]

$$y_i(t_{k+1}) = \begin{bmatrix} x_i(t_{k+1}) \\ \dot{x}_i(t_{k+1}) \end{bmatrix} = \begin{bmatrix} (c_{1,i}\,\cos(\beta_i\,\delta t) + c_{2,i}\,\sin(\beta_i\,\delta t))\, e^{-\alpha_i\,\delta t} + \frac{F_i}{\kappa_i} \\ -\left((c_{1,i}\alpha_i - c_{2,i}\beta_i)\,\cos(\beta_i\,\delta t) + (c_{1,i}\beta_i + c_{2,i}\alpha_i)\,\sin(\beta_i\,\delta t)\right) e^{-\alpha_i\,\delta t} \end{bmatrix}, \tag{51}$$

where $\delta t = t_{k+1} - t_k$, $\alpha_i = \zeta_i\,\omega_{\mathrm{n},i}$, and $\beta_i = \omega_{\mathrm{n},i}\sqrt{1 - \zeta_i^2}$. After enforcing the initial conditions $x_i(t_k), x_i(t_k)$, the integration constants

$$c_{1,i} = x_i(t_k) - \frac{F_i(t_k)}{\kappa_i}, \qquad c_{2,i} = -2\,j\,\frac{\dot{x}_i(t_k) + \alpha_i\left(x_i(t_k) - \frac{F_i}{k_i}\right)}{\Delta\lambda_i}, \tag{52}$$

can be identified with $\Delta\lambda_i = \lambda_{i,2} - \lambda_{i,1} = -2\beta_i j$, where $j$ is the imaginary value. While we could derive the solution for the critically damped case $d_i < 2\sqrt{\kappa_i}$ separately, we instead approximate it in our network dynamics with (51) by setting $\Delta\lambda_i \cong \begin{cases} \mathrm{sign}(-2\beta_i j)\,\epsilon & |2\beta_i j| < \epsilon \\ -2\beta_i j & |2\beta_i j| \geq \epsilon \end{cases}$, where $\epsilon \in \mathbb{R}^+ \ll 1$ is a small, positive value.

## B.2 Algorithmic implementation

We can now leverage the closed form solution to the evolution of a single, decoupled damped harmonic oscillator of (51) to solve the integral in (11)

$$y(t_{k+1}) \approx f_{\mathrm{CFA-CON}}(y(t_k), u(t_k)),$$

$$\begin{bmatrix} x(t_{k+1}) \\ \dot{x}(t_{k+1}) \end{bmatrix} \approx \begin{bmatrix} (c_1 \odot \cos(\beta\,\delta t) + c_2 \odot \sin(\beta\,\delta t)) \odot e^{-\alpha\,\delta t} + \frac{F(t_k)}{\kappa} \\ -\left((c_1 \odot \alpha - c_2 \odot \beta)\cos(\beta\,\delta t) + (c_1 \odot \beta + c_2 \odot \alpha)\sin(\beta\,\delta t)\right) \odot e^{-\alpha\,\delta t} \end{bmatrix}, \quad (53)$$

with

$$\kappa = \mathrm{diag}(K_{11}\ldots K_{nn}),\ d = (D_{11}\ldots D_{nn}),\ \omega_{\mathrm{n}} = \sqrt{\kappa},\ \zeta = \frac{d}{2\sqrt{\kappa}},\ \alpha = \zeta \odot \omega_{\mathrm{n}},\ \beta = \omega_{\mathrm{n}}\sqrt{1-\zeta^2},$$

$$F(t_k) = g(u(t_k)) - (K-\kappa)\,x(t_k) - (D-d)\,\dot{x}(t_k) - \tanh\left(Wx(t_k) + b\right),$$

$$c_1 = x(t_k) - \frac{F(t_k)}{\kappa}, \qquad c_2 = \frac{1}{\beta}\left(\dot{x}(t_k) + \alpha \odot \left(x(t_k) - \frac{F(t_k)}{\kappa}\right)\right). \quad (54)$$

We summarize the approach of integrating/rolling out the CFA-CON dynamics in Algorithm 1.

---

**Algorithm 1** Rollout of CFA-CON.

---

**Inputs:** initial state $y(t_0)$, input sequence $\{u(t_0), \ldots u(t_k), \ldots u(t_N)\}$
**Outputs:** state sequence $\{y(t_0), \ldots y(t_k), \ldots y(t_N)\}$
1: $k \leftarrow 0$
2: **while** $k \leq N$ **do**
3: $\quad (x(t_k), \dot{x}(t_k)) \leftarrow y(t_k)$
4: $\quad F(t_k) \leftarrow g(u(t_k)) - (K-\kappa)\,x(t_k) - (D-d)\,\dot{x}(t_k) - \tanh\left(Wx(t_k)+b\right)$
5: $\quad \omega_{\mathrm{n}},\ \zeta \leftarrow \sqrt{\kappa},\ \frac{d}{2\sqrt{\kappa}}$ $\qquad \triangleright$ Compute the characteristics of the decoupled harmonic oscillators.
6: $\quad \alpha,\ \beta \leftarrow \zeta \odot \omega_{\mathrm{n}},\ \omega_{\mathrm{n}}\sqrt{1-\zeta^2}$
7: $\quad c_1 \leftarrow x(t_k) - \frac{F(t_k)}{\kappa}$ $\qquad \triangleright$ Compute integration constants using initial conditions.
8: $\quad c_2 \leftarrow \frac{1}{\beta}\left(\dot{x}(t_k) + \alpha \odot \left(x(t_k) - \frac{F(t_k)}{\kappa}\right)\right)$
9: $\quad \delta t = t_{k+1} - t_k$ $\qquad\qquad\qquad\qquad\qquad\qquad\qquad \triangleright$ Set time step.
10: $\qquad\qquad\qquad\qquad\qquad\qquad\qquad \triangleright$ Update state with approximated closed-form solution.
11: $\quad x(t_{k+1}) \leftarrow (c_1 \odot \cos(\beta\,\delta t) + c_2 \odot \sin(\beta\,\delta t)) \odot e^{-\alpha\,\delta t} + \frac{F}{\kappa}$
12: $\quad \dot{x}(t_{k+1}) \leftarrow -\left((c_1 \odot \alpha - c_2 \odot \beta)\cos(\beta\,\delta t) + (c_1 \odot \beta + c_2 \odot \alpha)\sin(\beta\,\delta t)\right) \odot e^{-\alpha\,\delta t}$
13: $\quad k \leftarrow k+1$ $\qquad\qquad\qquad\qquad\qquad\qquad\qquad\qquad \triangleright$ Update time index.
14: **end while**

---

## B.3 Approximation bounds for CFA-CON

Lemma 11 demonstrates how, for the particular case of no external input and linearly decoupled oscillators (which we are always free to choose), we can establish bounds on the approximation error when using the closed-form solution instead of the ground-truth coupled oscillator dynamics.

**Lemma 11.** *Suppose that the network is unforced with $g(u(t)) = 0$ and that $K = \mathrm{diag}(\kappa_1, \ldots, \kappa_n)$, $D = \mathrm{diag}(d_1, \ldots, d_n)$ such that the oscillators are not linearly coupled. Then, given any $t \geq 0$, and the initial state $y(0) \in \mathbb{R}^{2n}$, the error between the continuous dynamics $\ddot{x}(t)$ of (2) and the approximated dynamics $\hat{\ddot{x}}(t)$ in (10), is bounded by $\|\ddot{x}(t) - \hat{\ddot{x}}(t)\| \leq 2$.*

*Proof.* (2), (10) and $F = -f_{\ddot{x},\mathrm{nld}}(y(0), 0)$ give us

$$\|\ddot{x}(t) - \hat{\ddot{x}}(t)\| = (f_{\ddot{x},\mathrm{ld}}(y(t), 0) + f_{\ddot{x},\mathrm{nld}}(y(t), 0)) - f_{\ddot{x},\mathrm{ld}}(y(t), F),$$
$$= -Kx(t) - D\dot{x} - \tanh(Wx(t)+b) + Kx(t) + D\dot{x} + \tanh(Wx(0)+b), \quad (55)$$
$$= \|-\tanh(Wx(t)+b) + \tanh(Wx(0)+b)\| \leq 2.$$

$\square$

## B.4 Empirical evaluation of approximation error

In Table 3, we present a comparison of CFA-CON with several other strategies for integrating nonlinear dynamics, such as CON. Following the implicit assumption made in Section B, we consider the case of $g(u) = 0$, $K = \mathrm{diag}(\kappa_1, \ldots, \kappa_n)$ and $D = \mathrm{diag}(d_1, \ldots, d_n)$ but with the hyperbolic coupling between the oscillators active (i.e., a full $W$ matrix). Integrating the dynamics at a very small time step (i.e., $\delta t = 5\mathrm{e}{-5}$ s) with a high-order ODE solver would give us a very accurate solution, but this is computationally infeasible in practice. We, therefore, regard this as the upper bound on the accuracy of the solution. A feasible solution would be to implement either a high-order solver such as Tsit5 at a larger integration time-step, e.g., $\delta t = 1\mathrm{e}{-1}$ s) or a low-order solver with a slightly smaller integration time step, e.g., $\delta t = 5\mathrm{e}{-2}$ s). Therefore, we also benchmark these options. We also benchmark an implementation specialized on the underdamped case (i.e., $\zeta_i < 1$): Closed-Form Approximation of the Underdamped Coupled Oscillator Network (CFA-UDCON). This specialized implementation allows us to avoid using complex numbers in the algorithm and reduces the number of computations necessary for calculating the approximated solution. As a result, we see a considerable increase in the sim-time to real-time factor.

### B.4.1 Integration error

We perform the integration error benchmark over 100 different network configurations, all consisting of 50 oscillators ($n = 50$): First, we sample the natural frequency of the $i$th oscillator from a uniform distribution as $\omega_{\mathrm{n},i} \sim \mathcal{U}(0.05\,\mathrm{Hz}, 0.5\,\mathrm{Hz})$, then we sample $\kappa_i \sim \mathcal{U}(0.2\,\mathrm{N/m}, 2\,\mathrm{N/m})$ such that $K = \mathrm{diag}(\kappa_1, \ldots, \kappa_n) \succ 0$, which lets us determine each mass $m_i = \frac{\kappa_i}{\omega_{\mathrm{n},i}^2}$. Next, the damping ratio is determined as $\zeta_i \sim \mathcal{U}(0.1, 0.9)$ and $\zeta_i \sim \mathcal{U}(0.1, 2.0)$ for the underdamped and general case, respectively. As a result, $D = \mathrm{diag}(d_1, \ldots, d_n) \succ 0$ with $d_i = 2\,\zeta_i\,\sqrt{m_i\,\kappa_i}$ given. Finally, by leveraging the Cholesky decomposition, we sample a $W \succ 0$ and $b_i \sim \mathcal{U}(-1, 1)$. We compute the estimation error of all integrated trajectories with respect to the high-precision solution (i.e., Tsitouras' 5/4 method (Tsit5) at $\delta t = 5\mathrm{e}{-4}$ s). For this, we compute the RMSE for each $60\,\mathrm{s}$ trajectory and then take the mean and standard deviation across the 100 different network configurations.

### B.4.2 Simulation-time to real-time factor

The simulation vs. real-time factor is computed as the simulated rollout duration per second of computational time. For this, we let each method simulate a $60\,\mathrm{s}$ trajectory for 100 times and record the minimum run time on an Intel Core i7-10870H CPU (single core) over 10 trials. Because of computational constraints, we simulated with the high-precision Tsit5 solver the trajectory only 5 times.

### B.4.3 Results

The results in Table 3 show that CFA-CON is $30\,\%$ more accurate than the Euler integrator at half of the speed. Compared against the Tsit5 integrator, CFA-CON exhibits a 1.56x speed increase while being significantly less accurate. For the underdamped case with $\zeta < 1$, the specialized implementation CFA-UDCON is $14.8\,\%$ faster and at the same time $32\,\%$ more accurate than the Euler integrator. Furthermore, CFA-UDCON is 3.7x faster and significantly less accurate than the Tsit5 integrator. We can conclude that in the pure rollout setting (i.e., no backpropagation involved) for a generic CON, the CFA-CON does not show clear advantages to an appropriately tuned Euler or Tsit5 solver. However, the specialized version CFA-UDCON demonstrates a 2.4x speed-up at no reduction of accuracy vs. CFA-CON for underdamped oscillator networks.

| Method | RMSE [m] $\downarrow$ | RMSE $\zeta < 1$ [m] $\downarrow$ | Complexity $\downarrow$ | $\frac{\text{Sim. time}}{\text{Real time}}$ $\uparrow$ |
|---|---|---|---|---|
| CON with Tsit5 at $\delta t = 5\mathrm{e}{-5}$ s | n/a | n/a | $\mathcal{O}\left(\frac{n^{\log_2 7} ph}{\delta t}\right) = \mathcal{O}(3.5\mathrm{e}11)$ | 5.68x |
| CON with Tsit5 at $\delta t = 1\mathrm{e}{-1}$ s | $5\mathrm{e}{-5} \pm 1\mathrm{e}{-5}$ | $8\mathrm{e}{-6} \pm 1\mathrm{e}{-5}$ | $\mathcal{O}\left(n\frac{n^{\log_2 7} ph}{\delta t}\right) = \mathcal{O}(1.8\mathrm{e}8)$ | 11310x |
| CON with Euler at $\delta t = 5\mathrm{e}{-2}$ s | $0.010 \pm 0.003$ | $0.022 \pm 0.005$ | $\mathcal{O}\left(\frac{n^{\log_2 7} h}{\delta t}\right) = \mathcal{O}(7.1\mathrm{e}7)$ | 36500x |
| CFA-CON (our) with $\delta t = 1\mathrm{e}{-1}$ s | $0.007 \pm 0.002$ | $0.015 \pm 0.003$ | $\mathcal{O}\left(\frac{n^{\log_2 7} h}{\delta t}\right) = \mathcal{O}(3.5\mathrm{e}7)$ | 17680x |
| CFA-UDCON (our) with $\delta t = 1\mathrm{e}{-1}$ s | n/a | $0.015 \pm 0.003$ | $\mathcal{O}\left(\frac{n^{\log_2 7} h}{\delta t}\right) = \mathcal{O}(3.5\mathrm{e}7)$ | **41900x** |

Table 3: Benchmarking of various methods for integrating the CON dynamics. The RMSE is computed with respect to the Tsitouras' 5/4 method (Tsit5) (i.e., extremely high accuracy but also extremely high computational complexity). We denote with $n$ the number of oscillators in the network (in this case $n = 50$), with $p$ the order of the numerical ODE solver, and with $\delta t$ the time-step. When stating the complexity, we refer to $h = t_N - t_0$ as the rollout horizon in seconds. In this case, we report the results for a horizon of $h = 60\,\mathrm{s}$. The RMSE column states the RMSE of the various integration strategies with respect to the *CON with Tsit5 at $\delta t = 5\mathrm{e}{-5}s$* solution, which we consider to be the ground-truth. The *RMSE $\zeta < 1$* computes the same metrics, but this time for a dataset that contains only underdamped oscillators. The $\frac{Sim.\ time}{Real\ time}$ column states the ratio between the duration of the simulation achieved (in seconds) per second of real-time (i.e., computational time). We report the mean and standard deviation of the RMSE over 100 different network configurations.

# C Appendix on experimental setup and datasets

## C.1 Datasets

For all datasets, we generate images of size $32 \times 32$px and subsequently normalize the pixels to the interval $[-1, 1]$.

### C.1.1 Unactuated mechanical datasets

We consider multiple mechanical datasets based on a standard implementation included in the *Toy Physics* category of the *NeurIPS 2021 Track on Datasets and Benchmarks* publication by Botev et al. [26]: mass-spring with friction (M-SP+F), a single pendulum with friction (S-P+F), and a double pendulum with friction (D-P+F). All datasets contain 5000 system trajectories in the training set and 1000 trajectories each in the validation and test set. Each trajectory is generated by first randomly initializing the system, then rolling it out for $3\,\mathrm{s}$ using an Euler integrator with a time step size of $5\,\mathrm{ms}$. Samples are recorded at a rate of $20\,\mathrm{Hz}$ (i.e., a time step of $0.05\,\mathrm{s}$). As a result, each trajectory contains 60 images of the system's state. As all of these datasets are unactuated, we can deactivate the input-to-forcing mapping component from all models (e.g., set $g(u) = 0$ for the CON model).

The *M-SP+F* dataset contains motion samples of a damped harmonic oscillator with a mass of $0.5\,\mathrm{kg}$, a spring stiffness of $2\,\mathrm{N/m}$, and a damping coefficient of $0.05\,\mathrm{Ns/m}$. For each trajectory, the initial condition of the mass-spring is randomly sampled by combining a random $\mathrm{sign}(q)$ with a uniformly sampled $|q| \sim \mathcal{U}(0.1\,\mathrm{m}, 1\,\mathrm{m})$. The position of the mass is rendered with a filled circle in a grayscale image.

The *SP+F* and *DP+F* datasets include the evolutions of a single-link pendulum and double-link pendulum, respectively, with a mass of $0.5\,\mathrm{kg}$ attached to the end of each link, which has a length of $1\,\mathrm{m}$. The dataset considers a gravitational acceleration of $3\,\mathrm{m/s^2}$. A rotational damper with coefficient $0.05\,\mathrm{Nms/rad}$ provides the friction. Similarly to the *M-SP+F* dataset, both the sign and the absolute value of the initial configuration are randomly sampled, where $|q(0)| \sim \mathcal{U}(1.3\,\mathrm{rad}, 2.3\,\mathrm{rad})$. The position of each mass is rendered with a filled circle. For the single-link pendulum, this is done in grayscale, and for the double pendulum, each mass is rendered with a different color (i.e., blue and red).

### C.1.2 Actuated continuum soft robot datasets

The shape of slender and deformable rods can be approximated by considering the deformations along the 1D curve of the backbone [57]. While this curve is still infinite-dimensional, it is possible to discretize the backbone into (many) segments with piecewise constant strain [58, 57]. Accordingly, we describe the kinematics of a planar continuum soft robot consisting of $n_\mathrm{b}$ segments with the PCS model [58]. We assume each segment has a length of $100\,\mathrm{mm}$ and a diameter of $20\,\mathrm{mm}$. The PCS model assumes each segment to have constant strain. In the planar case, this means that the shape of the $i$th segment can be parametrized by $\xi_i = \begin{bmatrix} \kappa_{\mathrm{be},i} & \sigma_{\mathrm{sh},i} & \sigma_{\mathrm{ax},i} \end{bmatrix}^\mathrm{T} \in \mathbb{R}^3$ where $\kappa_{\mathrm{be},i}$ is the bending strain (i.e., the curvature) in the unit $\mathrm{rad/m}$, $\sigma_{\mathrm{sh},i}$ is the shear strain (dimensionless), and $\sigma_{\mathrm{ax},i}$ is the axial elongation strain (dimensionless). The robot's configuration is then defined as $q = \begin{bmatrix} \xi_1^\mathrm{T} & \cdots & \xi_i^\mathrm{T} & \cdots & \xi_{n_\mathrm{b}}^\mathrm{T} \end{bmatrix}^\mathrm{T}$. In the case of Piecewise Constant Curvature (PCC), only the bending strain is active as shear strains and axial strains are neglected, and the configuration is now $q \in \mathbb{R}^{n_\mathrm{b}}$. The PCS model generates EOM in the form of [59]

$$B(q)\,\ddot{q} + C(q, \dot{q})\,\dot{q} + G(q) + K_\mathrm{q}\,q + D_\mathrm{q}\,\dot{q} = u(t), \tag{56}$$

where $B(q) \succ 0$ and $C(q, \dot{q})$ are the inertia and Corioli matrices, respectively. $G(q)$ collects the gravitational forces, $K_\mathrm{q} \succ 0$ is the stiffness matrix, and $D_\mathrm{q} \succ 0$ contains the damping coefficients. $u(t) \in \mathbb{R}^{n_\mathrm{b}}$ is an external force acting on the generalized coordinates, and now $m = n_\mathrm{b}$.

We derive the corresponding dynamics for a continuum soft robot of material density $600\,\mathrm{kg/m^3}$, elastic modulus of $20\,000\,\mathrm{Pa}$, shear modulus of $10\,000\,\mathrm{Pa}$, and damping coefficients of $0.000\,01\,\mathrm{Nm^2s}$ for bending strains, $0.01\,\mathrm{Ns}$ for shear strains, and $0.01\,\mathrm{Ns}$ for axial strains, respectively. Gravity is pointing downwards. The implementation of the dynamics in JAX [60] is based on the *JSRM* library [54, 61], and we simulate the robot using a constant integration time step of $0.1\,\mathrm{ms}$. We render grayscale images of the robot with a size of $32 \times 32$px at a rate of $50\,\mathrm{Hz}$ using OpenCV [62]. We generate 10000 trajectories, each of duration $2.0\,\mathrm{s}$ and a sampling time-step of $0.02\,\mathrm{s}$. We use $60\,\%$ training, $20\,\%$ validation, and $20\,\%$ test split. For each trajectory, we randomly sample a constant actuation/input $u \sim \mathcal{U}(-u_{\max}, u_{\max})$. We choose the maximum actuation magnitude to be equal to the sum of the contribution of the potential forces (i.e., elastic and gravitational forces): $u_{\max} = G(q_{\max}) + K\,q_{\max}$ with $q_{\max,i} = \begin{bmatrix} 5\,\pi\,\mathrm{rad/m}, 0.2, 0.2 \end{bmatrix}^\mathrm{T}$.

We generate three datasets based on this continuum soft robot model: in the *CS* dataset, we consider one segment with all three planar strains active (i.e., bending, shear, and elongation). This results in three DOF and six-state variables in the dynamical model. In the case of the *PCC-NS-2* and *PCC-NS-3* datasets, we base the dataset on a simulated system consisting of two planar bending segments, respectively. Each segment is parametrized using Constant Curvature (CC) [63, 64], which results in two configuration variables and a state dimension of four.

### C.1.3 Unactuated PDE reaction-diffusion dataset

We consider the $1^{st}$-order Reaction-diffusion (*R-D*) PDE on which (Champion et al, 2019). [9] evaluated their SINDy Autoencoder on. The PDE of the high-dimensional lambda-omega reaction-diffusion system is defined as

$$
\begin{aligned}
\frac{\partial u}{\partial t} &= \left(1 - (u^2 + v^2)\right) u + \beta \left(u^2 + v^2\right) v + d_1 \left(\frac{\partial^2 u}{\partial q_1^2} + \frac{\partial^2 u}{\partial q_2^2}\right), \\
\frac{\partial v}{\partial t} &= -\beta(u^2 + v^2) u + (1 - (u^2 + v^2)) v + d_2 \left(\frac{\partial^2 v}{\partial q_1^2} + \frac{\partial^2 v}{\partial q_2^2}\right),
\end{aligned}
\tag{57}
$$

where $u(t, q) : \mathbb{R} \times \mathbb{R}^2 \to \mathbb{R}$ and $v(t, q) : \mathbb{R} \times \mathbb{R}^2 \to \mathbb{R}$ are time-dependent two vector fields defined over the spatial domain $q \in \mathbb{R}^2$. We choose the same system parameters and initial condition as Champion et al. [9]: $d_1, d_2 = 0.1$, and $\beta = 1$ and

$$
\begin{aligned}
u(0, q) &= \tanh\left(\sqrt{q_1^2 + q_2^2} \cos\left(\angle(q_1 + iq_2) - \sqrt{q_1^2 + q_2^2}\right)\right), \\
v(0, q) &= \tanh\left(\sqrt{q_1^2 + q_2^2} \sin\left(\angle(q_1 + iq_2) - \sqrt{q_1^2 + q_2^2}\right)\right).
\end{aligned}
\tag{58}
$$

After discretizing the spatial domain into 32 points along each dimension, we solve the PDE with a MATLAB ODE45 solver the solution of $u(t, q)$ and $v(t, q)$ at each time step and grid point. Subsequently, the solution is multiplied with a Gaussian centered at the origin [9]

$$
\begin{aligned}
\bar{u}(t, q) &= \exp(-0.01 (q_1^2 + q_2^2)) \, \bar{u}(t, q), \\
\bar{v}(t, q) &= \exp(-0.01 (q_1^2 + q_2^2)) \, \bar{v}(t, q).
\end{aligned}
\tag{59}
$$

We integrate the system from the specified initial condition for $500\,\mathrm{s}$ and store samples at a time step of $0.05\,\mathrm{s}$. We divide the entire sequence into 99 subsequences each containing 101 samples. We train the models to predict these subsequences that have a horizon of $5.0\,\mathrm{s}$ each.

We stack the solution of $\bar{u}(t, q)$ and $\bar{v}(t, q)$ contained in the two grids $o_\mathrm{u}(t), o_\mathrm{v}(t) \in \mathbb{R}^{32 \times 32}$, respectively, to gather the images $o(t) \in \mathbb{R}^{32 \times 32 \times 2}$ containing two channels. A sample sequence of the generated images is presented in Apx. 17. We use $60\,\%$ of the subsequences (i.e., 59) as our training set, and employ $20\,\%$ (i.e., 19) for the validation and test sets, respectively.

## C.2 Autoencoder architecture

For the encoder and decoder, we rely on a vanilla CNNs implemented as a $\beta$-VAE [5].

**Encoder.** The encoder consists of two convolutional layers with kernel size $(3, 3)$ and stride $(1, 1)$ mapping to $16, 32$, respectively. The features are flattened and then passed to two linear layers with hidden dimension 256 and $n_z$. Each layer (except for the last) is followed by a layer norm [65] and a LeakyReLU nonlinearity.

**Decoder.** The decoder first uses two linear layers to map to hidden dimensions of 256 and 32768, respectively. We then apply two 2D transposed convolutions [66] reducing the number of channels first to 16, and then to 1. Each layer (except for the last linear and last convolutional) is followed by a layer norm [65] and a LeakyReLU nonlinearity. Finally, we apply a sigmoid function to clip the output into the range $[-1, 1]$.

## C.3 Latent dynamic models

In the following section, we provide implementation details for the latent dynamic models that we evaluated as part of this work.

### C.3.1 Coupled Oscillator Network (CON)

We leverage the CON in $\mathcal{W}$-coordinates given by (3) for learning latent space dynamics. Specifically, we consider the input-to-force mapping $g(u)) = B(u) u(t)$, where $B(u) \in \mathbb{R}^{n \times m}$ is parametrized by few-layer MLP. We report results for two different sizes of the MLP: one medium-sized variant consisting of five layers with a hidden dimension of 30 and a small variant with two layers and a hidden dimension of 12. In both cases, we use a hyperbolic tangent as a nonlinearity.

When training the model, we jointly optimize $M_\mathrm{w}^{-1}, K_\mathrm{w}, D_\mathrm{w}, b$ and $g(u)$. However, we also need to make sure that we adhere to the stability constraints $M_\mathrm{w}^{-1}, K_\mathrm{w}, D_\mathrm{w} \succ 0$. For this, we leverage the Cholvesky decomposition [67]. Instead of directly learning the full matrix $A \in \mathbb{R}^{n_z \times n_z}$, we designate the elements of an upper triangular matrix $U \in \mathbb{R}^{n_z \times n_z}$ as the trainable parameters. The Cholesky decomposition demands that $\mathrm{diag}(U_{11}, \ldots, U_{n_z n_z}) > 0$. Therefore, we apply the operation

$$
U_{ii} = \log\left(1 + e^{\breve{U}_{ii} + \epsilon_1}\right) + \epsilon_2,
\tag{60}
$$

where $\breve{U}$ is the learned upper triangular matrix, and $\epsilon_1 = 1\mathrm{e}{-6}$ and $\epsilon_2 = 2\mathrm{e}{-6}$ are two small, positive values. The positive-definite matrix $A$ is now given by $A = U^{\mathrm{T}} U \succ 0$.

### C.3.2 Neural ODEs

We consider two kinds of Neural ODEs [27]: the vanilla $f_{\mathrm{NODE}} : \xi(t) \times u(t) \mapsto \dot{\xi}(t)$ maps latent state and system actuation directly into a time derivative of the latent state. In contrast, for the *MECH-NODE*, we enforce the latent dynamics to have a mechanical structure

$$\dot{\xi}(t) = \begin{bmatrix} \frac{\mathrm{d}z}{\mathrm{d}t} \\ \frac{\mathrm{d}\dot{z}}{\mathrm{d}t} \end{bmatrix} = \begin{bmatrix} \dot{z}(t) \\ f_{\mathrm{MECH-NODE}}(\xi(t), u(t)) \end{bmatrix}. \tag{61}$$

We represent both $f_{\mathrm{NODE}}$ and $f_{\mathrm{MECH-NODE}}$ as MLPs consisting of 5 layers, a hidden dimension of 30, and a hyperbolic tangent nonlinearity.

### C.3.3 Autoregressive models

For the below stated autoregressive models, we divide the integration between two (latent) samples $\xi(t_k)$ and $\xi(t_{k+1})$ into $N_{\mathrm{int}}$ integration steps $\xi(t_k + \delta t), \dots, \xi(t_k + k'\delta t), \dots, \xi(t_k + N_{\mathrm{int}}\delta t)$ where $\delta t$ is the integration step size and $t_{k+1} = t_k + N_{\mathrm{int}}\delta t$. The autoregressive model now describes the transition $\xi(t_{k'+1}) = f_{\mathrm{ar}}(\xi(t_{k'}), u(t_k)) \, \forall k' \in 1, \dots, N_{\mathrm{int}}$.

**RNN.** We implement a standard, single-layer Elman RNN with `tanh` nonlinearity. The hidden state captures the latent state of the system. The latent state transition functions are given by

$$\xi(t_{k'+1}) = \tanh(W_{\mathrm{hh}}\, \xi(t_{k'}) + b_{\mathrm{hh}} + W_{\mathrm{ih}}\, u(t_k) + b_{\mathrm{ih}}), \tag{62}$$

where $W_{\mathrm{hh}} \in \mathbb{R}^{2n_z \times 2n_z}$, $b_{\mathrm{hh}} \in \mathbb{R}^{2n_z}$, $W_{\mathrm{ih}} \in \mathbb{R}^{2n_z \times m}$, and $b_{\mathrm{ih}} \in \mathbb{R}^{2n_z}$.

**GRU.** We implement a standard, single-layer GRU [30] with `sigmoid` activation function where we interpret the latent state of the system as the hidden state of the cell. The latent state transition functions are given by

$$\begin{aligned} r &= \sigma\left(W_{\mathrm{hr}}\, \xi(t_{k'}) + b_{\mathrm{hr}} + W_{\mathrm{ir}}\, u(t_k) + b_{\mathrm{ir}}\right) \\ p &= \sigma\left(W_{\mathrm{hp}}\, \xi(t_{k'}) + b_{\mathrm{hp}} + W_{\mathrm{ip}}\, u(t_k) + b_{\mathrm{ip}}\right) \\ n &= \tanh\left(r \odot \left(W_{\mathrm{hn}}\, \xi(t_{k'}) + b_{\mathrm{hn}}\right) + W_{\mathrm{in}}\, u(t_k) + b_{\mathrm{in}}\right) \\ \xi(t_{k'+1}) &= (1 - p) \odot n + p \odot \xi(t_{k'}) \end{aligned} \tag{63}$$

where $\sigma$ is the sigmoid function, $\odot$ the Hadamard product, $W_{\mathrm{hr}}, W_{\mathrm{hp}}, W_{\mathrm{hn}} \in \mathbb{R}^{2n_z \times 2n_z}$, $W_{\mathrm{ir}}, W_{\mathrm{ip}}, W_{\mathrm{in}} \in \mathbb{R}^{2n_z \times m}$, and $b_{\mathrm{hr}}, b_{\mathrm{ir}}, b_{\mathrm{ip}}, b_{\mathrm{in}} \in \mathbb{R}^{2n_z}$.

**coRNN.** A time-discrete coRNN is defined by the transition function

$$\xi(t_{k'+1}) = \begin{bmatrix} z(t_{k'+1}) \\ \dot{z}(t_{k'+1}) \end{bmatrix} = \begin{bmatrix} z(t_{k'}) + \delta t\, \dot{z}(t_{k'}) \\ \dot{z}(t_{k'}) + \delta t\, \left(-\gamma z(t_{k'}) - \varepsilon \dot{z}(t_{k'}) + \tanh\left(W\xi(t_{k'}) + V u(t_k) + b\right)\right) \end{bmatrix} \tag{64}$$

where $\gamma, \varepsilon \in \mathbb{R}^+$ are positive, scalar hyperparameters representing the stiffness and damping coefficients, respectively. The term $\tanh\left(W\xi(t_{k'}) + V u(t_k) + b\right)$ with $W \in \mathbb{R}^{2n_z \times 2n_z}$, $V \in \mathbb{R}^{n_z \times m}$, and $b \in \mathbb{R}^{n_z}$ contributes nonlinear state-to-state connections. It is implemented with a linear layer operating on $(\xi(t_{k'}), u(t_k))$ followed by a hyperbolic tangent nonlinearity.

**CFA-CON.** We adapt the Alg. 1 for predicting the time evolution in latent-space

$$\xi(t_{k'+1}) = f_{\mathrm{CFA-CON}}(\xi(t_k'), u(t_k)), \tag{65}$$

where $f_{\mathrm{CFA-CON}}$ describe the autoregressive state transition by the CFA-CON model as introduced in Eq. 11.

### C.4 First-order variants of dynamical models

For learning (latent) dynamics of $1^{\mathrm{st}}$-order systems (e.g., the reaction-diffusion dataset *R-D*), it might be beneficial also to formulate the dynamical model to be of $1^{\mathrm{st}}$-order. While this is straightforward for some dynamics that do not explicitly take the order into account (e.g., RNN, GRU, NODE), for other models such as coRNN, CON, and CFA-CON more adjustments are necessary. Namely, we substitute the $\frac{\mathrm{d}z}{\mathrm{d}t}$ component of the ODE with the expression for $\frac{\mathrm{d}\dot{z}}{\mathrm{d}t}$. Furthermore, we remove any terms that depend on the velocity $\dot{z}$ (e.g., damping effects). Below, we report in detail the adapted, $1^{\mathrm{st}}$-order formulations for the coRNN, CON, and CFA-CON models.

**CON.** In the $1^{\mathrm{st}}$-order version, we adapt the standard, $2^{\mathrm{nd}}$-order ODE of the CON network as defined in (3) to

$$\dot{\xi}(t) = \dot{z}(t) = M_{\mathrm{w}}^{-1}\left(g(u(t)) - K_{\mathrm{w}}z(t) - \tanh(z(t) + b)\right). \tag{66}$$

**coRNN.** In the 1$^{\text{st}}$-order version, we define the transition function as

$$\xi(t_{k'+1}) = z(t_{k'+1}) = z(t_{k'}) - \delta t\,\gamma\,z(t_{k'}) + \delta t\,\tanh\left(W\xi(t_{k'}) + Vu(t_k) + b\right). \tag{67}$$

**CFA-CON.** We adapt a 1$^{\text{st}}$-order version of Alg. 1 for predicting the time evolution in latent-space

$$\xi(t_{k'+1}) = z(t_{k'+1}) = z(t_{k'}) + \int_{t_{k'}}^{t_{k'}+\delta t} F(t_{k'}) - \kappa\,z(t')\,\mathrm{d}t',$$
$$F(t_{k'}) = g(u(t_k)) - (K - \kappa)z(t_{k'}) - \tanh(Wz(t_{k'}) + b), \tag{68}$$

where the closed-form solution for the integral is given by

$$\int_{t_{k'}}^{t_{k'}+\delta t} F(t_{k'}) - \kappa\,z(t')\,\mathrm{d}t' = \frac{F(t_{k'})}{\kappa}\left(1 - e^{-\kappa\,\delta t}\right). \tag{69}$$

## C.5 Estimation of the initial latent velocity

For 2$^{\text{nd}}$-order systems and when integrating the evolution of the latent state $\xi(t) = \begin{bmatrix} z^{\mathrm{T}}(t) & \dot{z}^{\mathrm{T}}(t) \end{bmatrix}^{\mathrm{T}}$ in time, we need to have access to an initial latent velocity $\dot{z}(t_0)$ such that we can roll out the latent state $\xi(t)$ in time. A naive approach to estimating such an initial latent velocity would be to encode multiple (at least two) images of the system at the start of the trajectory into latent space and then perform numerical differentiation (e.g., finite differences) in latent space. However, we found the resulting $\dot{z}(t_k)$ to be relatively noisy and susceptible to small encoding errors. Instead, we propose to perform numerical differentiation in image space and then map this velocity into latent space using the encoder's Jacobian. First, we estimate the image-space velocity at $t_k$ using finite differences: $\dot{o}(t_k) \approx \frac{o(t_{k+1}) - o(t_{k-1})}{t_{k+1} - t_{k-1}}$. The latent velocity is then estimated as $\dot{z}(t_k) = \frac{\partial \Phi}{\partial o}(o(t_k))\,\dot{o}(t_k)$, where $\frac{\partial \Phi}{\partial o}$ is obtained with forward-mode automatic differentiation.

## C.6 Training

We implement the network dynamics and the neural networks (e.g., encoder, decoder, and MLPs) in JAX [60] and Flax [68], respectively. When training or inferring time-continuous dynamical models (e.g., NeuralODE, CON), we rely on Diffrax [47] for numerical integration of the ODE using the Dormand-Prince's 5/4 method [69] (Dopri5). For the numerical integration of both the time-continuous and the time-discrete models (e.g., RNN, coRNN, CFA-CON), we use an integration time-step $\delta t$ of $0.025\,\text{s}$ and $0.01\,\text{s}$ for the *Toy Physics* [26] and soft robotic datasets, respectively.

Because of the GPU memory constraints, we limit ourselves to a batch size of 30 and 80 trajectories for the *Toy Physics* [26] and soft robotic datasets, respectively. We implement a learning rate schedule consisting of a warm-up (5 epochs) and a cosine annealing [70] period (remaining epochs). We employ an AdamW optimizer [71, 72] with $\beta_1 = 0.9$, $\beta_2 = 0.999$ for updating both neural network weights (e.g., encoder, decoder) and parameters of the dynamical model (e.g., $K, D_{\mathrm{w}}, M_{\mathrm{w}}$, etc.).

Before training, we conduct a hyperparameter selection study using Optuna [53]. For this, we leverage a Tree-Structured Parzen Estimator [73] for identifying hyperparameters such as the base learning rate, the weight decay, the loss function weights, and model-specific hyperparameters such as the number of MLP layers, the hidden dimension of the MLP layers, the $\gamma$ and $\epsilon$ values for the coRNN model etc. that minimize the RMSE of the predicted images. To reduce computational requirements, we employ the Asynchronous Successive Halving Algorithm [74] to stop unpromising trials early.

## C.7 Evaluation metrics

Similar to other publications in the field [75, 76, 77], we state the RMSE, the Peak Signal-to-Noise Ratio (PSNR) and the Structural Similarity Index Measure (SSIM) [78] between the ground-truth image $o \in \mathbb{R}^{h_{\mathrm{o}} \times w_{\mathrm{o}}}$ and the predicted image image $\hat{o} \in \mathbb{R}^{h_{\mathrm{o}} \times w_{\mathrm{o}}}$. We use the separated test set for all evaluation results.

### C.7.1 Root Mean-Square Error

The RMSE between the two images is given by

$$\text{RMSE}(o, \hat{o}) = \sqrt{\sum_{u=1}^{h_{\mathrm{o}}} \sum_{v=1}^{w_{\mathrm{o}}} \frac{(o_{uv} - \hat{o}_{uv})^2}{h_{\mathrm{o}}\,w_{\mathrm{o}}}}. \tag{70}$$

### C.7.2 Peak Signal-to-Noise Ratio

The PSNR is a function of the total Mean Squared Error (MSE) loss and the maximum dynamic range of the image $L$.

$$\text{PSNR}(o, \hat{o}) = 20 \log_{10}(L) - 10 \log_{10}\left(\sum_{u=1}^{h_\text{o}} \sum_{v=1}^{w_\text{o}} \frac{(o_{uv} - \hat{o}_{uv})^2}{h_\text{o}\, w_\text{o}}\right). \tag{71}$$

As we work with normalized images with pixels in the interval $[-1, 1]$, the dynamic range is $L = 2$.

### C.7.3 Structural Similarity Index Measure

As simple pixel-by-pixel metrics such as RMSE or PSNR tend to average out any encountered errors, this could lead to a situation in which a significant reconstruction error in a part of the image is not seen in the RMSE metric but has a huge impact on the visual appearance of the reconstruction. SSIM [78] incorporates not just the *absolute errors*, but also the strong inter-dependencies between pixels, especially when they are spatially close. The SSIM metric between two observations $o$ and $\hat{o}$ is given by

$$\text{SSIM}(o, \hat{o}) = l^\alpha(o, \hat{o})\, c^\beta(o, \hat{o})\, s^\gamma(o, \hat{o}), \tag{72}$$

where

$$l(o, \hat{o}) = \frac{2\mu_o\mu_{\hat{o}} + C_1}{\mu_o^2 + \mu_{\hat{o}}^2 + C_1}, \quad c(o, \hat{o}) = \frac{2\sigma_o\sigma_{\hat{o}} + C_2}{\sigma_o^2 + \sigma_{\hat{o}}^2 + C_2}, \quad s(o, \hat{o}) = \frac{\sigma_{o\hat{o}} + C_3}{\sigma_o\sigma_{\hat{o}} + C_3}. \tag{73}$$

We use the constants $C_1 = (k_1 L)^2$, $C_2 = (k_2 L)^2$ and $C_3 = C_2/2$, where $L$ signifies the dynamic range as previously used for the PSNR metric, and $k_1 = 0.01$ and $k_2 = 0.03$. The average $\mu$ and the variance $\sigma^2$ is computed with a Gaussian filter with a 1D kernel of size 11 and sigma 1.5. We set the weight exponents $\alpha$, $\beta$, and $\gamma$ for the luminance, contrast, and structure comparisons all to one. We rely on the PIX library [79] for efficiently computing the SSIM metric.

## C.8 Compute resources

We trained the models on several desktop workstations for a total duration of roughly $150\,\text{h}$. In total, we relied on 10x RTX 3090/4090 GPUs, each with 24 GB of VRAM, training the models in parallel. Each workstation contained between 64 and 128 GB of RAM, and we used roughly 100 GB of total storage. Training each model on one random seed took between $45\,\text{min}$ and $4\,\text{h}$ depending on the model type, the integration time constant, and the number of trainable parameters. The hyperparameter tuning we conducted beforehand (only on one random seed) took roughly the same time and computational resources as generating the final results.

For the control experiments, we additionally used a laptop with a 16-core Intel Core i7-10870H CPU and 32 GB RAM. We did not need to use a GPU for evaluating the model during closed-loop control.

# D  Appendix on learning latent dynamics

We report the full set of quantitative results, including the additional evaluation metrics PSNR and SSIM, in Tables 7 to 9. For *PCC-NS-2* in Tab. 8, we additionally also recorded the training steps per second on an Nvidia RTX 3090 GPU with a batch size of 80 (which leads to 8080 images per batch). We plot the results of a sweep across the latent dimensions for the additional evaluation metrics PSNR and SSIM in Fig. 8. Correspondingly, we visualize the number of trainable parameters of each model vs. the latent dimension in Fig. 9. In Figs. 10- 16, we present sequences of stills for the rollout of the trained *CON*(-M) models on the various datasets.

| Model | RMSE ↓ | PSNR ↑ | SSIM ↑ | # Parameters ↓ |
|---|---|---|---|---|
| RNN | $0.2739 \pm 0.0057$ | $4.16 \pm 0.02$ | $0.6958 \pm 0.0122$ | 88 |
| GRU [30] | $0.0267 \pm 0.0033$ | $\mathbf{6.13 \pm 0.09}$ | $0.9861 \pm 0.0022$ | 248 |
| coRNN [35] | $\mathbf{0.0265 \pm 0.0002}$ | $\mathbf{6.13 \pm 0.01}$ | $\mathbf{0.9853 \pm 0.0006}$ | 40 |
| NODE [27] | $\mathbf{0.0264 \pm 0.0010}$ | $\mathbf{6.14 \pm 0.03}$ | $\mathbf{0.9858 \pm 0.0009}$ | 3368 |
| MECH-NODE | $0.0328 \pm 0.0034$ | $5.99 \pm 0.07$ | $0.9821 \pm 0.0024$ | 3244 |
| CON (our) | $0.0303 \pm 0.0053$ | $6.05 \pm 0.13$ | $0.9847 \pm 0.0027$ | $\mathbf{34}$ |
| CFA-CON (our) | $0.0313 \pm 0.0026$ | $6.02 \pm 0.06$ | $0.9843 \pm 0.0008$ | $\mathbf{34}$ |

Table 4: Benchmarking of CON and CFA-CON at learning latent dynamics on the ***M-SP+F* (mass-spring with friction) dataset**. For all models, a latent dimension of $n_z = 4$ is chosen. As this dataset does not consider any inputs, we remove all parameters in the RNN, GRU, coRNN, CON, and CFA-CON models related to the input mapping. *MECH-NODE* is a NODE with prior knowledge about the mechanical structure of the system (i.e., $\frac{\mathrm{d}x}{\mathrm{d}t} = \dot{x}$). We report the mean and standard deviation over three different random seeds and the number of parameters of each latent dynamics model.

| Model | RMSE ↓ | PSNR ↑ | SSIM ↑ | # Parameters ↓ |
|---|---|---|---|---|
| RNN | $0.2378 \pm 0.0352$ | $4.31 \pm 0.15$ | $0.7568 \pm 0.0350$ | 88 |
| GRU [30] | $0.1457 \pm 0.0078$ | $4.78 \pm 0.05$ | $0.9168 \pm 0.0093$ | 248 |
| coRNN [35] | $0.1333 \pm 0.0044$ | $4.86 \pm 0.03$ | $0.9194 \pm 0.0055$ | 40 |
| NODE [27] | $\mathbf{0.1260 \pm 0.0013}$ | $\mathbf{4.91 \pm 0.01}$ | $\mathbf{0.9379 \pm 0.0009}$ | 3368 |
| MECH-NODE | $0.1650 \pm 0.0205$ | $4.67 \pm 0.12$ | $0.8985 \pm 0.0153$ | 3244 |
| CON (our) | $0.1303 \pm 0.0064$ | $4.88 \pm 0.04$ | $0.9175 \pm 0.0095$ | $\mathbf{34}$ |
| CFA-CON (our) | $0.1352 \pm 0.0073$ | $4.85 \pm 0.05$ | $0.9133 \pm 0.0052$ | $\mathbf{34}$ |

Table 5: Benchmarking of CON and CFA-CON at learning latent dynamics on the ***S-P+F* (single pendulum with friction) dataset**. For all models, a latent dimension of $n_z = 4$ is chosen. As this dataset does not consider any inputs, we remove all parameters in the RNN, GRU, coRNN, CON, and CFA-CON models related to the input mapping. *MECH-NODE* is a NODE with prior knowledge about the mechanical structure of the system (i.e., $\frac{\mathrm{d}x}{\mathrm{d}t} = \dot{x}$). We report the mean and standard deviation over three different random seeds and the number of parameters of each latent dynamics model.

| Model | RMSE ↓ | PSNR ↑ | SSIM ↑ | # Parameters ↓ |
|---|---|---|---|---|
| RNN | $0.1694 \pm 0.0004$ | $4.631 \pm 0.002$ | $0.7082 \pm 0.0032$ | 672 |
| GRU [30] | $0.1329 \pm 0.0005$ | $4.858 \pm 0.003$ | $\mathbf{0.8340 \pm 0.0021}$ | 1968 |
| coRNN [35] | $0.1324 \pm 0.0016$ | $4.862 \pm 0.012$ | $0.8229 \pm 0.0039$ | 348 |
| NODE [27] | $0.1324 \pm 0.0024$ | $4.861 \pm 0.016$ | $0.8101 \pm 0.0024$ | 4404 |
| MECH-NODE | $0.1710 \pm 0.0111$ | $4.624 \pm 0.063$ | $0.7170 \pm 0.0439$ | 4032 |
| CON (our) | $0.1323 \pm 0.0018$ | $4.862 \pm 0.013$ | $0.8067 \pm 0.0038$ | $\mathbf{246}$ |
| CFA-CON (our) | $\mathbf{0.1307 \pm 0.0012}$ | $\mathbf{4.873 \pm 0.008}$ | $0.8147 \pm 0.0034$ | $\mathbf{246}$ |

Table 6: Benchmarking of CON and CFA-CON at learning latent dynamics on the ***D-P+F* (double pendulum with friction) dataset**. For all models, a latent dimension of $n_z = 12$ is chosen. As this dataset do not consider any inputs, we remove all parameters in the RNN, GRU, coRNN, CON, and CFA-CON models related to the input mapping. *MECH-NODE* is a NODE with prior knowledge about the mechanical structure of the system (i.e., $\frac{\mathrm{d}x}{\mathrm{d}t} = \dot{x}$). We report the mean and standard deviation over three different random seeds and the number of parameters of each latent dynamics model.

| Model | RMSE ↓ | PSNR ↑ | SSIM ↑ | # Parameters ↓ |
|---|---|---|---|---|
| RNN | **0.1011 ± 0.0009** | **25.92 ± 0.08** | **0.9777 ± 0.0004** | 696 |
| GRU [30] | 0.1125 ± 0.0100 | 24.99 ± 0.74 | 0.9730 ± 0.0040 | 2040 |
| coRNN [35] | 0.2537 ± 0.0018 | 17.93 ± 0.06 | 0.8820 ± 0.0024 | **336** |
| NODE [27] | 0.2415 ± 0.0021 | 18.36 ± 0.08 | 0.8946 ± 0.0023 | 4374 |
| MECH-NODE | 0.2494 ± 0.0028 | 18.08 ± 0.10 | 0.8898 ± 0.0016 | 4002 |
| CON-S (our) | 0.1993 ± 0.0646 | 20.03 ± 2.44 | 0.9218 ± 0.0380 | 1386 |
| CON-M (our) | 0.1063 ± 0.0027 | 25.49 ± 0.22 | 0.9758 ± 0.0011 | 8568 |
| CFA-CON (our) | 0.1462 ± 0.0211 | 22.72 ± 1.17 | 0.9573 ± 0.0103 | 8568 |

Table 7: Benchmarking of CON and CFA-CON at learning latent dynamics on the ***CS (soft robot with one constant strain segment) dataset***. For all models, a latent dimension of $n_z = 12$ is chosen. *CON-S* and *CON-M* are small and medium-sized versions of the CON model, respectively. *MECH-NODE* is a NODE with prior knowledge about the mechanical structure of the system (i.e., $\frac{\mathrm{d}x}{\mathrm{d}t} = \dot{x}$). We report the mean and standard deviation over three different random seeds and the number of parameters of each latent dynamics model.

| Model | RMSE ↓ | PSNR ↑ | SSIM ↑ | # Parameters ↓ | $\frac{\text{Train. steps}}{\text{second}}$ ↑ | Inf. time [ms] ↑ |
|---|---|---|---|---|---|---|
| RNN | 0.1373 ± 0.0185 | 23.27 ± 1.10 | 0.9643 ± 0.0077 | 320 | 1.87 | 02.6 |
| GRU [30] | 0.0951 ± 0.0021 | **26.45 ± 0.19** | 0.9730 ± 0.0040 | 928 | 1.83 | 03.2 |
| coRNN [35] | 0.2504 ± 0.0899 | 18.05 ± 2.66 | 0.9814 ± 0.0006 | **152** | **1.89** | 02.7 |
| NODE [27] | 0.1867 ± 0.0561 | 20.60 ± 2.28 | 0.8774 ± 0.0857 | 3856 | 0.79 | 50.2 |
| MECH-NODE | 0.1035 ± 0.0012 | 25.07 ± 0.06 | 0.9778 ± 0.0004 | 3062 | 0.79 | 50.3 |
| CON-S (our) | 0.0996 ± 0.0012 | 26.05 ± 0.11 | **0.9792 ± 0.0007** | 676 | 0.78 | 50.2 |
| CON-M (our) | 0.1008 ± 0.0006 | 25.95 ± 0.05 | 0.9786 ± 0.0003 | 7048 | 0.60 | 60.1 |
| CFA-CON (our) | 0.1124 ± 0.0025 | 25.01 ± 0.19 | 0.9734 ± 0.0012 | 7048 | 1.12 | 13.6 |

Table 8: Benchmarking of CON and CFA-CON at learning latent dynamics on the ***PCC-NS-2 (soft robot with two constant curvature segments) dataset***. For all models, a latent dimension of $n_z = 8$ is chosen. *CON-S* and *CON-M* are small and medium-sized versions of the CON model, respectively. *MECH-NODE* is a NODE with prior knowledge about the mechanical structure of the system (i.e., $\frac{\mathrm{d}x}{\mathrm{d}t} = \dot{x}$). We report the mean and standard deviation over three different random seeds. Furthermore, we state the number of parameters of each latent dynamics model and the training steps per second on a Nvidia RTX 3090 GPU. Each batch contains 80 trajectories and 8080 images of resolution 32x32px in total. Finally, we report the inference time averaged over 5000 runs for performing a rollout of $2.02\,\mathrm{s}$ (while encoding and decoding all images along the trajectory) on an Nvidia RTX 3090 GPU with a batch size of 1.

## D.1 Results for Reaction-diffusion dataset

As all previous examples exampled ODEs, we strive to test the proposed approach also on a system that is governed by PDEs. Specifically, we consider the Reaction-diffusion (*R-D*) dataset as introduced in Apx. C.1.3. To address the unactuated nature of the dataset, we remove, analog to the *M-SP+F*, *S-P+F*, and *D-P+F* datasets, the input-to-state mapping parameters of the dynamical models (e.g., the $B(u)$ and $E(\tau)$ MLPs for the CON models). Furthermore, the PDE describing the system dynamics is of 1[st]-order. Therefore, we leverage the 1[st]-order versions of the latent dynamics as specified in Apx. C.4.

We report the metrics of the test set evaluations in Tab. 10. Furthermore, we also present a sequence of stills of the rollout of a trained latent dynamics CON model in Fig. 17. We find it impressive that CON with its strong stability guarantees can accurately model the dynamics of a high-dimensional PDE system.

| Model | RMSE ↓ | PSNR ↑ | SSIM ↑ | # Parameters ↓ |
|---|---|---|---|---|
| RNN | $0.2232 \pm 0.0075$ | $19.05 \pm 0.29$ | $0.8955 \pm 0.0083$ | 696 |
| GRU [30] | $0.2148 \pm 0.0196$ | $19.38 \pm 0.76$ | $0.9039 \pm 0.0223$ | 2040 |
| coRNN [35] | $0.2474 \pm 0.0018$ | $18.15 \pm 0.06$ | $0.8877 \pm 0.0011$ | **336** |
| NODE [27] | $0.3373 \pm 0.0565$ | $15.46 \pm 1.34$ | $0.7432 \pm 0.0935$ | 4374 |
| MECH-NODE | $0.1900 \pm 0.0024$ | $20.45 \pm 0.11$ | $0.9315 \pm 0.0011$ | 4002 |
| CON-S (our) | $\mathbf{0.1792 \pm 0.0038}$ | $\mathbf{20.96 \pm 0.18}$ | $\mathbf{0.9392 \pm 0.0023}$ | 1386 |
| CON-M (our) | $\mathbf{0.1785 \pm 0.0023}$ | $\mathbf{20.99 \pm 0.11}$ | $\mathbf{0.9395 \pm 0.0018}$ | 8568 |
| CFA-CON (our) | $0.1803 \pm 0.0003$ | $20.90 \pm 0.01$ | $0.9366 \pm 0.0004$ | 8568 |

Table 9: Benchmarking of CON and CFA-CON at learning latent dynamics on the ***PCC-NS-3* (soft robot with three constant curvature segments) dataset**. For all models, a latent dimension of $n_z = 12$ is chosen. *CON-S* and *CON-M* are small and medium-sized versions of the CON model, respectively. *MECH-NODE* is a NODE with prior knowledge about the mechanical structure of the system (i.e., $\frac{\mathrm{d}x}{\mathrm{d}t} = \dot{x}$). We report the mean and standard deviation over three different random seeds and the number of parameters of each latent dynamics model.

| Model | RMSE ↓ | PSNR ↑ | SSIM ↑ | # Parameters ↓ |
|---|---|---|---|---|
| RNN | $0.3763 \pm 0.0374$ | $3.82 \pm 0.12$ | $0.4463 \pm 0.1358$ | **20** |
| GRU [30] | $0.3232 \pm 0.0368$ | $\mathbf{3.99 \pm 0.13}$ | $0.6798 \pm 0.0949$ | 52 |
| 1$^{\text{st}}$-order coRNN [35] | $\mathbf{0.0741 \pm 0.0001}$ | $5.35 \pm 0.00$ | $\mathbf{0.9724 \pm 0.0014}$ | **20** |
| NODE [27] | $\mathbf{0.0738 \pm 0.0007}$ | $\mathbf{5.36 \pm 0.01}$ | $\mathbf{0.9683 \pm 0.0022}$ | 3064 |
| CON (our) | $0.1110 \pm 0.0160$ | $5.03 \pm 0.12$ | $0.9372 \pm 0.0109$ | 24 |
| CFA-CON (our) | $0.1068 \pm 0.0059$ | $5.05 \pm 0.05$ | $0.9418 \pm 0.0026$ | 24 |

Table 10: Benchmarking of CON and CFA-CON at learning latent dynamics on the ***R-D* (reaction-diffusion) dataset**. For all models, a latent dimension of $n_z = 4$ is chosen. As this dataset does not consider inputs, we remove all parameters in the RNN, GRU, coRNN, CON, and CFA-CON models related to the input mapping. Also, as the *reaction-diffusion* system is governed by 1$^{\text{st}}$-order PDE dynamics, we use specialized, 1$^{\text{st}}$-order version of the *CON*, *CFA-CON*, and *coRNN* dynamics. We report the mean and standard deviation over three different random seeds and the number of parameters of each latent dynamics model.

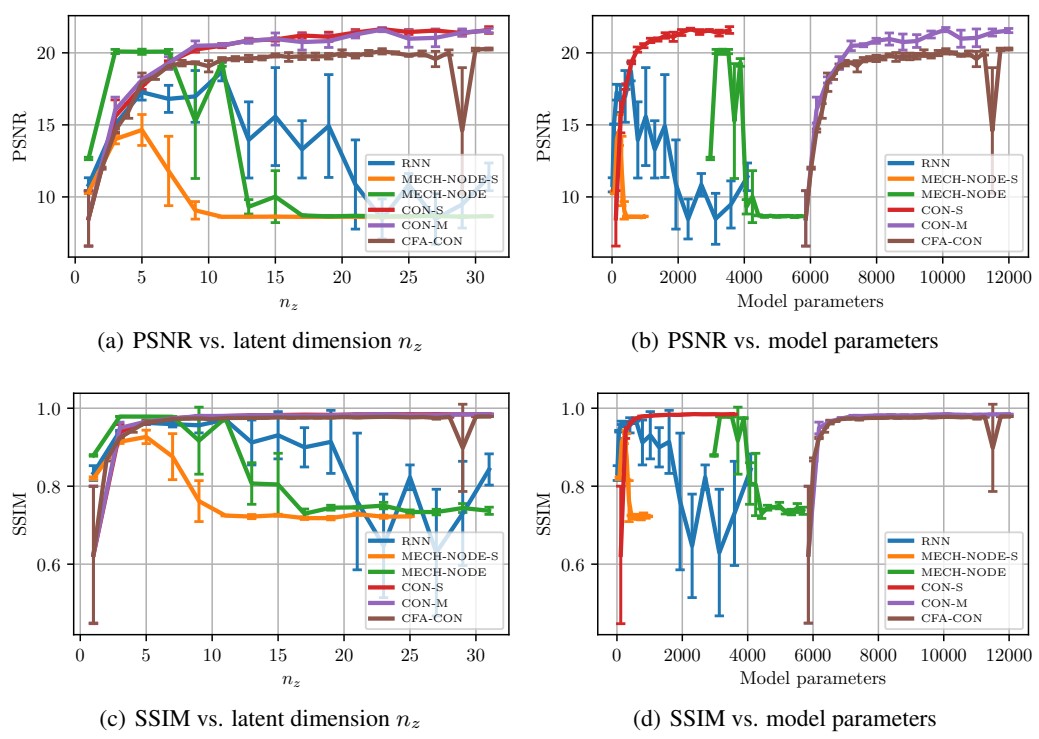

(a) PSNR vs. latent dimension $n_z$

(b) PSNR vs. model parameters

(c) SSIM vs. latent dimension $n_z$

(d) SSIM vs. model parameters

Figure 8: Evaluation of prediction performance of the various models vs. the dimension of their latent representation $n_z$ and the number of trainable parameters of the dynamics model, respectively. We optimize the hyperparameters for the case of $n_z = 8$, and execute the tuning separately for each model and dataset.

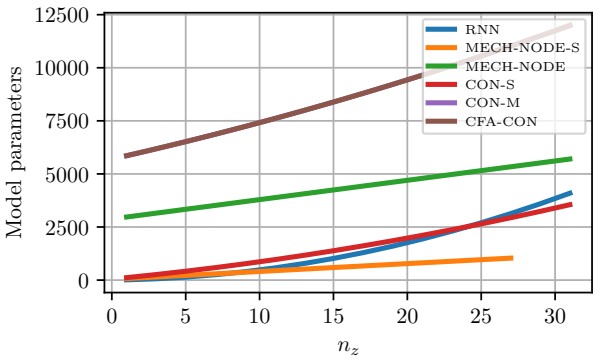

Figure 9: Plot of number of trainable parameters vs. the latent dimension $n_z$ of various models trained on the *PCC-NS-2* dataset. As we have configured them, *CON-M* and *CFA-CON* always have the same number of parameters (i.e., overlaying lines).

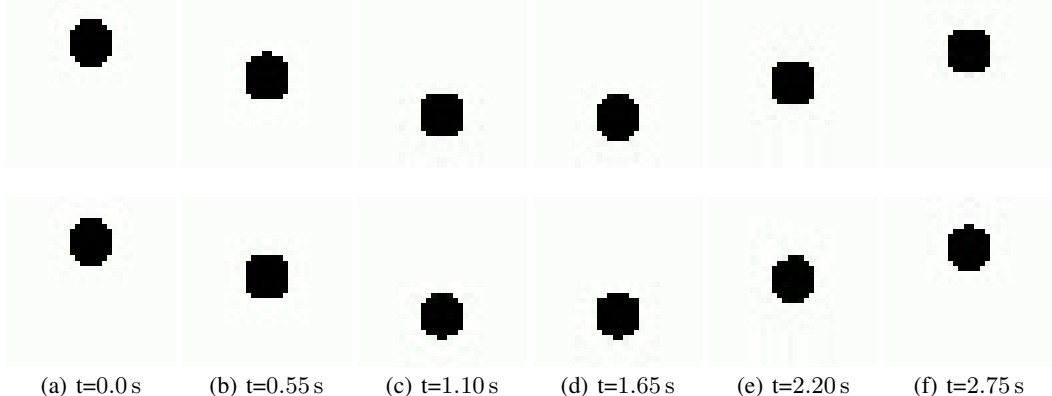

(a) t=0.0 s  (b) t=0.55 s  (c) t=1.10 s  (d) t=1.65 s  (e) t=2.20 s  (f) t=2.75 s

Figure 10: Prediction sequence of a CON model with latent dimension $n_z = 4$ trained on the damped harmonic oscillator (*M-SP+F*) dataset [26]. **Top row:** Ground-truth evolution of the system. **Bottom row:** Predictions of the *CON* model.
The prediction model is given three images centered around $t = 0$ for encoding the initial latent $z(0)$ and estimation of the initial latent velocity $\dot{z}(0)$. Subsequently, we roll out the autonomous network dynamics (i.e., unforced) and compare the decoded predictions with the ground-truth evolution of the system.

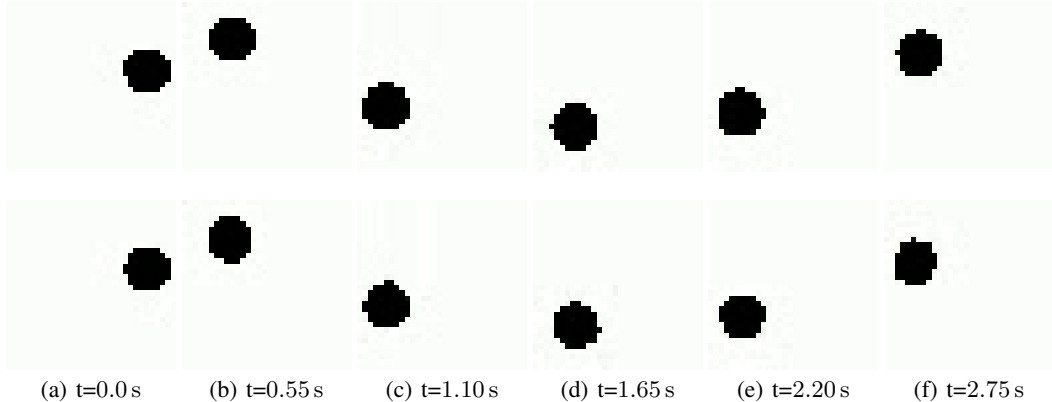

(a) t=0.0 s  (b) t=0.55 s  (c) t=1.10 s  (d) t=1.65 s  (e) t=2.20 s  (f) t=2.75 s

Figure 11: Prediction sequence of a CON model with latent dimension $n_z = 4$ trained on the single pendulum with friction (*S-P+F*) dataset [26]. **Top row:** Ground-truth evolution of the system. **Bottom row:** Predictions of the *CON* model.
The prediction model is given three images centered around $t = 0$ for encoding the initial latent $z(0)$ and estimation of the initial latent velocity $\dot{z}(0)$. Subsequently, we roll out the autonomous network dynamics (i.e., unforced) and compare the decoded predictions with the ground-truth evolution of the system.

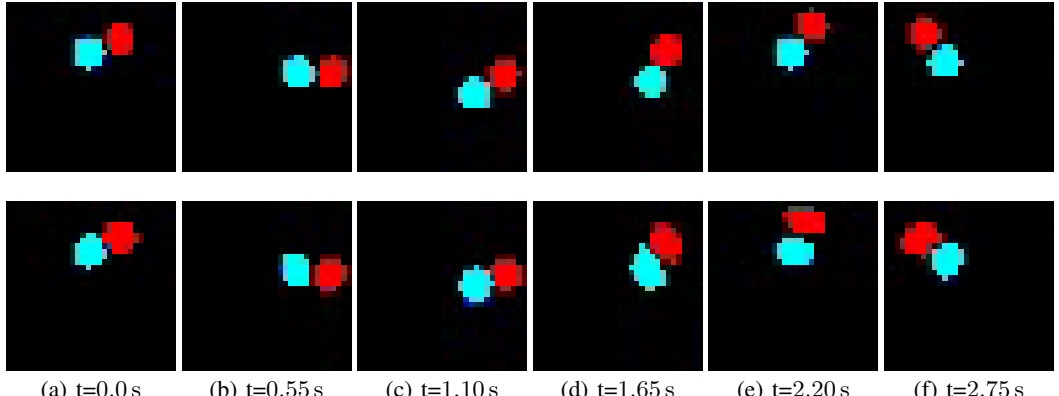

(a) t=0.0 s  (b) t=0.55 s  (c) t=1.10 s  (d) t=1.65 s  (e) t=2.20 s  (f) t=2.75 s

Figure 12: Prediction sequence of a CON model with latent dimension $n_z = 12$ trained on the double pendulum with friction (*D-P+F*) dataset [26]. **Top row:** ground-truth evolution of the system. **Bottom row:** predictions of the *CON* model.

The prediction model is given three images centered around $t = 0$ for encoding the initial latent $z(0)$ and estimation of the initial latent velocity $\dot{z}(0)$. Subsequently, we roll out the autonomous network dynamics (i.e., unforced) and compare the decoded predictions with the ground-truth evolution of the system.

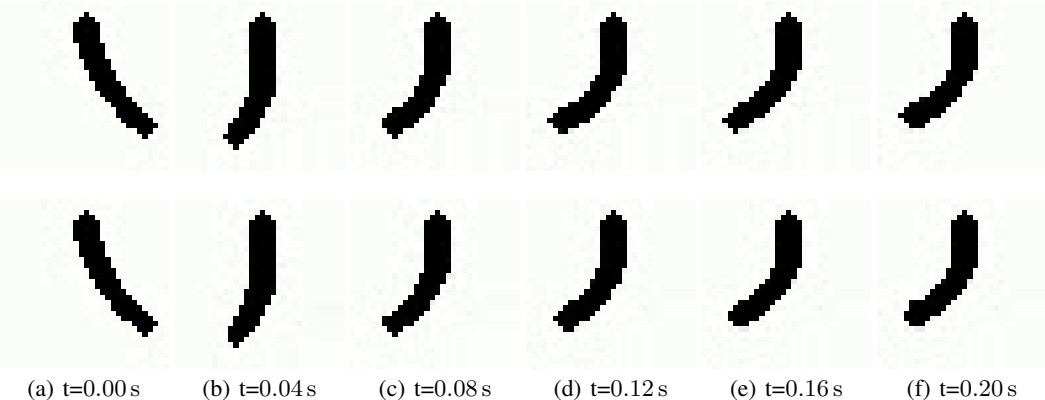

(a) t=0.00 s  (b) t=0.04 s  (c) t=0.08 s  (d) t=0.12 s  (e) t=0.16 s  (f) t=0.20 s

Figure 13: Prediction sequence of a forced CON model with latent dimension $n_z = 12$ trained on the soft robotic *CS* dataset containing trajectories of a simulated constant strain robot with one segment. **Top row:** Ground-truth evolution of the system. **Bottom row:** Predictions of the *CON-M* model.

The prediction model is given three images centered around $t = 0$ for encoding the initial latent $z(0)$ and estimation of the initial latent velocity $\dot{z}(0)$. Subsequently, we roll out the autonomous network dynamics (i.e., unforced) and compare the decoded predictions with the ground-truth evolution of the system.

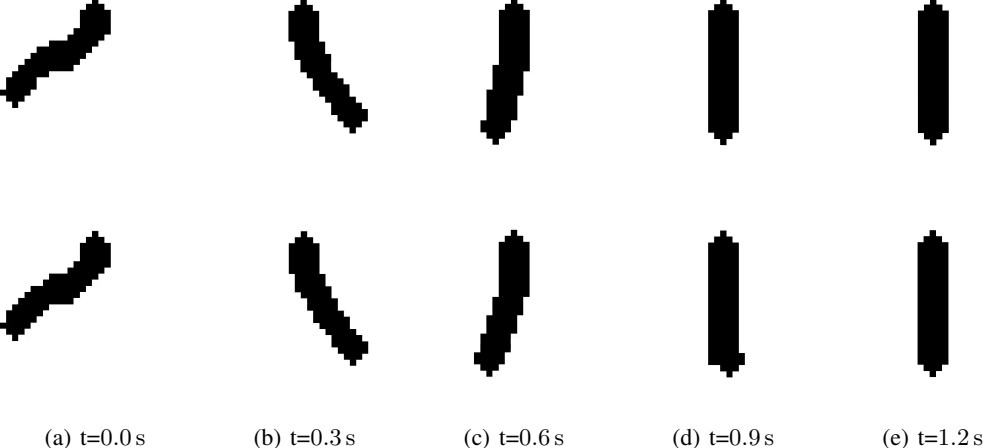

(a) t=0.0 s     (b) t=0.3 s     (c) t=0.6 s     (d) t=0.9 s     (e) t=1.2 s

Figure 14: Prediction sequence of an unforced CON model with latent dimension $n_z = 8$ trained on the *PCC-NS-2* dataset. **Top row:** Ground-truth evolution of the system. **Bottom row:** Predictions of the *CON-M* model.
The prediction model is given three images centered around $t = 0$ for encoding the initial latent $z(0)$ and estimation of the initial latent velocity $\dot{z}(0)$. Subsequently, we roll out the autonomous network dynamics (i.e., unforced) and compare the decoded predictions with the ground-truth evolution of the system.

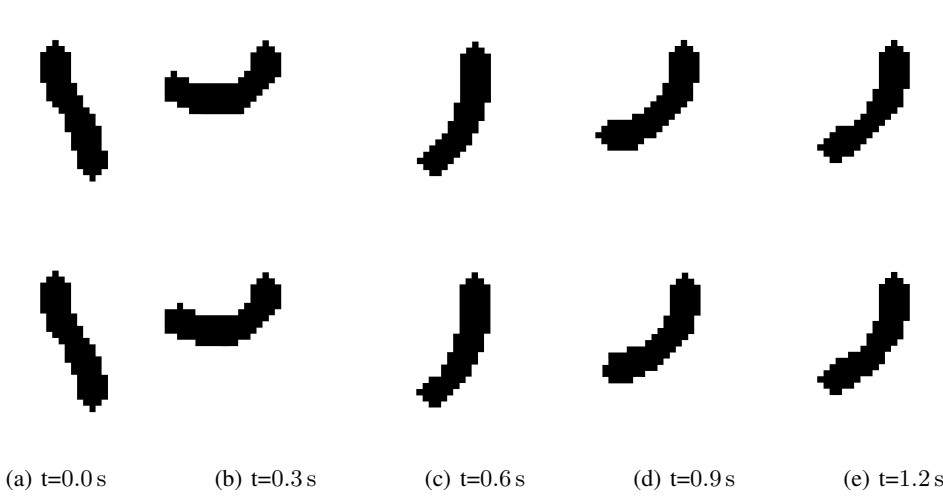

(a) t=0.0 s     (b) t=0.3 s     (c) t=0.6 s     (d) t=0.9 s     (e) t=1.2 s

Figure 15: Prediction sequence of a forced CON model with latent dimension $n_z = 8$ trained on the *PCC-NS-2* dataset. **Top row:** Ground-truth evolution of the system. **Bottom row:** Predictions of the *CON-M* model.
The prediction model is given three images centered around $t = 0$ for encoding the initial latent $z(0)$ and estimation of the initial latent velocity $\dot{z}(0)$. Subsequently, we provide the same constant input $u$ to both the simulator and the network dynamics (i.e., unforced) and compare the decoded predictions with the ground-truth evolution of the system.

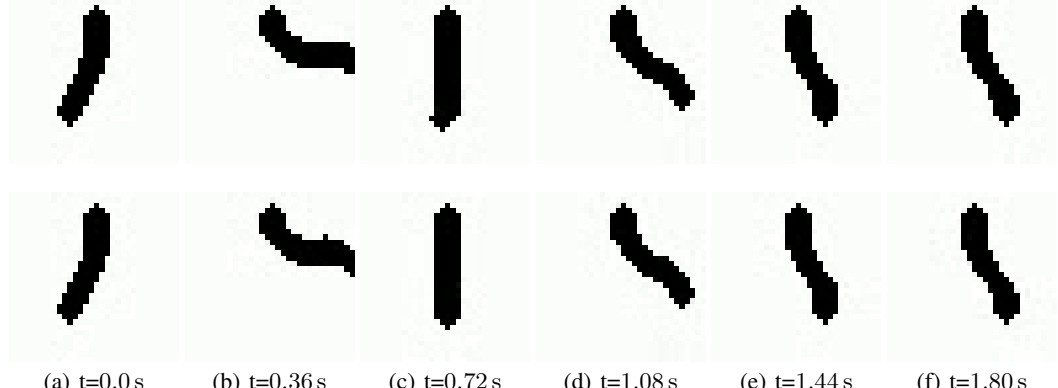

|   (a) t=0.0 s   |   (b) t=0.36 s   |   (c) t=0.72 s   |   (d) t=1.08 s   |   (e) t=1.44 s   |   (f) t=1.80 s   |

Figure 16: Prediction sequence of a forced CON model with latent dimension $n_z = 12$ trained on the soft robotic *PCC-NS-3* dataset containing trajectories of a simulated piecewise constant curvature robot with three segments. **Top row:** Ground-truth evolution of the system. **Bottom row:** Predictions of the *CON-M* model. The prediction model is given three images centered around $t = 0$ for encoding the initial latent $z(0)$ and estimation of the initial latent velocity $\dot{z}(0)$. Subsequently, we roll out the autonomous network dynamics (i.e., unforced) and compare the decoded predictions with the ground-truth evolution of the system.

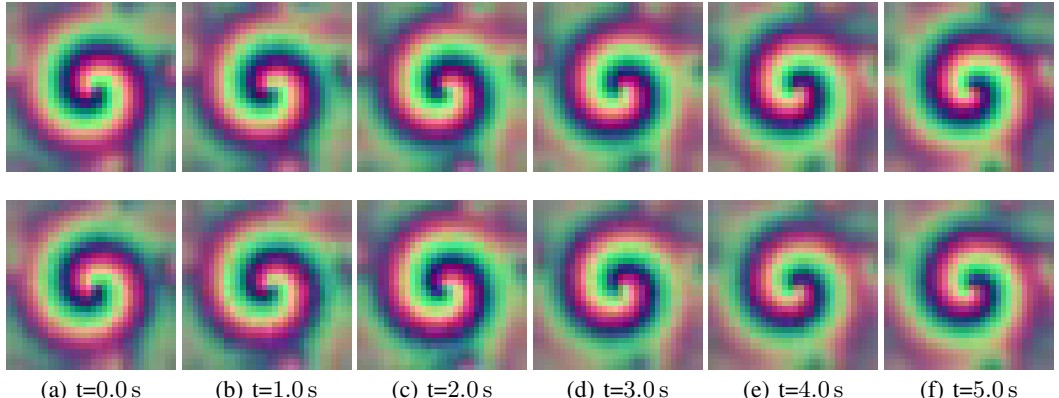

|   (a) t=0.0 s   |   (b) t=1.0 s   |   (c) t=2.0 s   |   (d) t=3.0 s   |   (e) t=4.0 s   |   (f) t=5.0 s   |

Figure 17: Prediction sequence of an unforced, $1^{\text{st}}$-order CON model with latent dimension $n_z = 4$ trained on the reaction-diffusion (*R-D*) dataset. **Top row:** Ground-truth evolution of the system. **Bottom row:** Predictions of the *CON-M* model. We roll out the autonomous, $1^{\text{st}}$-order network dynamics (i.e., unforced) and compare the decoded predictions with the ground-truth evolution of the system.

# E  Appendix on latent-space control

## E.1  Latent-space control of a damped harmonic oscillator

We consider an actuated version of the *M-SP+F* dataset (i.e., a damped harmonic oscillator) and denote it with *M-SP+F+A*. All system, trajectory sampling and rendering parameters remain the same, except that for each trajectory in the dataset we randomly sample a forcing $u \sim \mathcal{U}(-1\,\mathrm{N}, 1\,\mathrm{N})$.

We train a CON model with latent dimension $n_z = 1$ over three random seeds on the *M-SP+F+A* dataset. This means that the network consists of a single oscillator. From the three different random seeds, we choose the model that achieves the best validation loss, which results in an RMSE of 0.0327, a PSNR of 5.99, and SSIM of 0.9796 on the test set.

Fig. 18 shows how the encoder learns an almost linear relationship between the actual configuration of the system and the predicted latent space representation. Furthermore, we notice that both the ground-truth and the learned potential energy are convex and exhibit a global minimum at $q = 0\,\mathrm{m}$.

We compare the performance of *P-satI-D*, *D+FF*, and *P-satI-D+FF* controllers based on the CON model in Fig. 19. For the *P-satI-D* controller, we choose the control gains $K_\mathrm{p} = 10, K_\mathrm{i} = 10, K_\mathrm{d} = 5, \upsilon = 1$. The *D+FF* controller uses $K_\mathrm{d} = 3.5$. Finally, the *P-satI-D+FF* is configured with $K_\mathrm{p} = 2, K_\mathrm{i} = 0.3, K_\mathrm{d} = 3.5, \upsilon = 1$. The results show that the *P-satI-D+FF* controller exhibits thanks to its feedforward term no overshooting and a faster response time than the pure feedback controller *P-satI-D*. The high accuracy of the feedfoward term can be seen from the performance of the *D+FF* controller, that only exhibits relatively small steady-state error. Adding small proportional and integral feedback actions in the *P-satI-D+FF* controller keeps the compliance high while removing the steady-state error and reducing the response time.

Finally, we visualize the behavior of the *P-satI-D+FF* controller as a sequence of stills in Fig. 20.

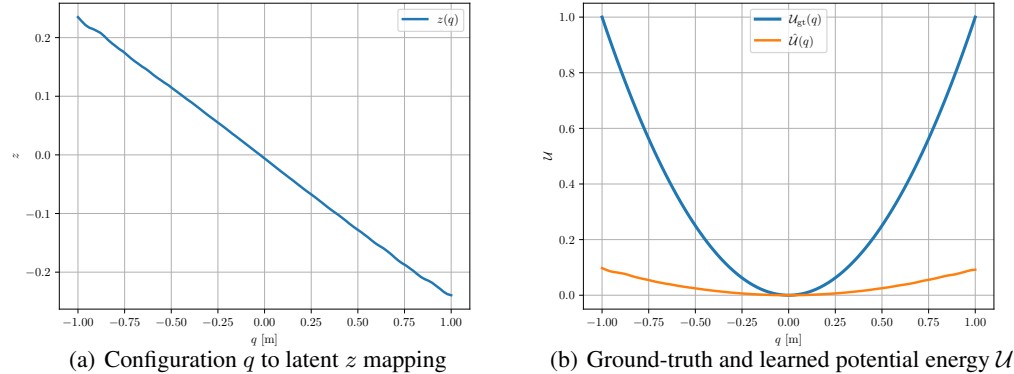

(a) Configuration $q$ to latent $z$ mapping     (b) Ground-truth and learned potential energy $\mathcal{U}$

Figure 18: **Panel (a):** Learned mapping from configuration to latent space for the CON model with $n_z$ (i.e., consisting of a single oscillator) trained on the actuated damped harmonic oscillator (*M-SP+F+A*) dataset. **Panel (b):** The blue line represents the ground-truth potential energy of the damped harmonic oscillator. The orange line represents the learned potential energy of the CON model evaluated vs. the system configuration by rendering and subsequently encoding into latent space each configuration value.

## E.2  Latent-space control of a two segment PCC soft robot

### E.2.1  Potential energy landscape

When leveraging (learned) dynamical models for setpoint regulation, it is essential to accurately estimate the potential energy as this dictates the efficacy of the feedforward terms. Therefore, we qualitatively evaluate the potential energy landscape of the CON latent dynamic model.

In Fig. 21(a), we can see how CON contains a single, isolated, and globally asymptotically stable equilibrium as proven in Appendix A.2 and Section 2, respectively.

Furthermore, we want to verify that the learned potential corresponds to the actual potential energy of the simulated system. An autonomous continuum soft robot with the tip pointing downwards in a straight configuration exhibits an isolated, globally asymptotically stable equilibrium at $q = 0$ (i.e., zero strains) [42]. For this purpose, we can compare the learned potential energy field in Fig. 21(b) with the ground-truth potential energy field in Fig. 21(c). We confirm, based on Fig. 21(b), that, indeed, the learned potential also has its minimum close to/at $q = 0$. Although the field is shaped slightly differently, the potential forces are clearly pointing inwards towards the global attractor.

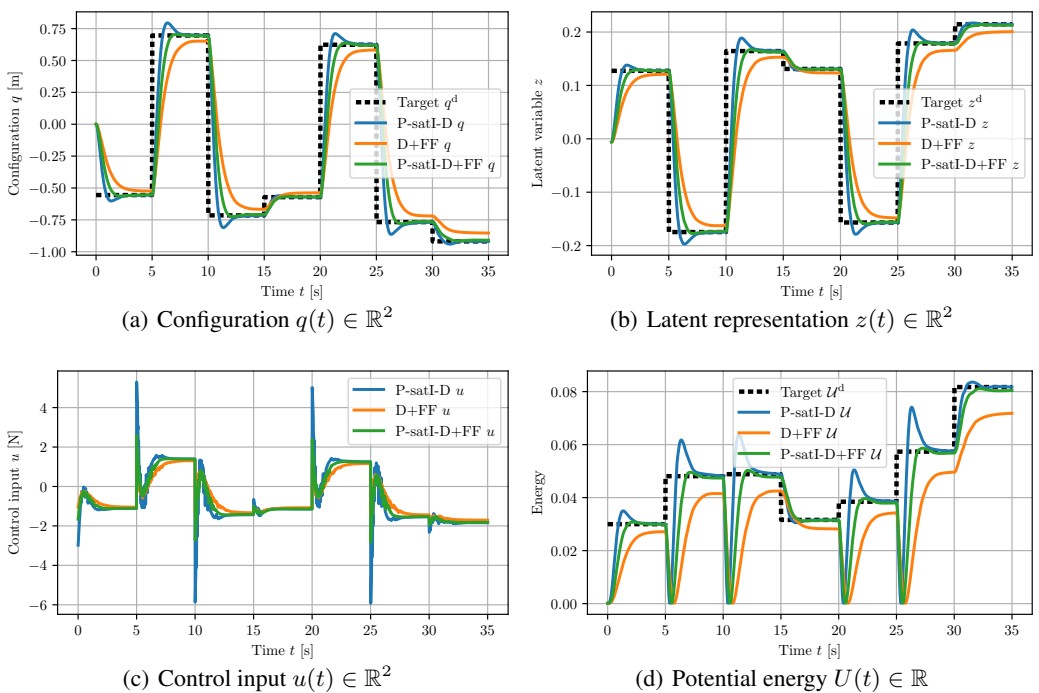

(a) Configuration $q(t) \in \mathbb{R}^2$                     (b) Latent representation $z(t) \in \mathbb{R}^2$

(c) Control input $u(t) \in \mathbb{R}^2$                     (d) Potential energy $U(t) \in \mathbb{R}$

Figure 19: Latent-space control of an actuated damped harmonic oscillator (*M-SP+F+A*) following a sequence of setpoints. We compare multiple controllers based on a trained CON network with $n_z = 1$. The CON model weights are initialized using a random seed of 0. The blue line represents a pure feedback controller (*P-satI-D*). The orange line visualizes the behavior of a feedforward controller with only a damping term applied in feedback (*D+FF*). The green line shows the performance of our proposed combination of feedback and feedforward terms (*P-satI-D+FF*). The dotted and solid lines show the reference and actual values, respectively. For each setpoint, we randomly sample a desired shape $q^{\mathrm{d}}$ and render the corresponding image $o^{\mathrm{d}}$. This image is then encoded to a target latent $z^{\mathrm{d}}$. The controller then computes a latent-space torque $F^{\mathrm{d}}$, which is decoded to an input $u$. Finally, we provide this input to the simulator, which performs a roll-out of the closed-loop dynamics. Important: The robot's configuration (i.e., the first-principle, minimal-order state) is solely used for generating a target image and simulating the closed-loop system.

### E.2.2 Model selection

For the control experiments, we train instances of the *MECH-NODE* and *CON-M* models with latent dimension $n_z = 2$ and with neural network weights initialized with three different random seeds. For *MECH-NODE*, we choose the model with the lowest validation loss (seed 0).

For the *CON* network, we found that model-based control does not perform as well when the latent stiffness $\Gamma_{\mathrm{w}}$ (as visualized in Fig. 21(a)) is significantly larger along one of the Eigenvectors than along the other one. Therefore, we evaluate the Eigenvalues of the learned stiffness matrix in $\mathcal{W}$-coordinates after training: $\lambda_{1,2}(\Gamma_{\mathrm{w}})$. Particularly, we choose the seed that minimizes the normalized standard deviation of the Eigenvalues

$$
\begin{aligned}
\mu_\lambda &= \frac{\lambda_1(\Gamma_{\mathrm{w}}) + \lambda_2(\Gamma_{\mathrm{w}})}{2}, \\
\sigma_\lambda &= \sqrt{\frac{(\lambda_1(\Gamma_{\mathrm{w}}) - \mu_\lambda)^2 + (\lambda_2(\Gamma_{\mathrm{w}}) - \mu_\lambda)^2}{2}}, \\
\mathrm{seed} &= \arg\min \frac{\sigma_\lambda}{\mu_\lambda}.
\end{aligned}
\tag{74}
$$

### E.2.3 Additional control results

Additional results for the *P-satI-D* feedback controller based on the MECH-NODE and CON models are provided in Fig. 23, Fig. 24, respectively and for the *P-satI-D+FF* controller based on the CON model in Fig. 25. Sequences of stills for the *CON P-satI-D+FF* controller are provided in Fig. 22.

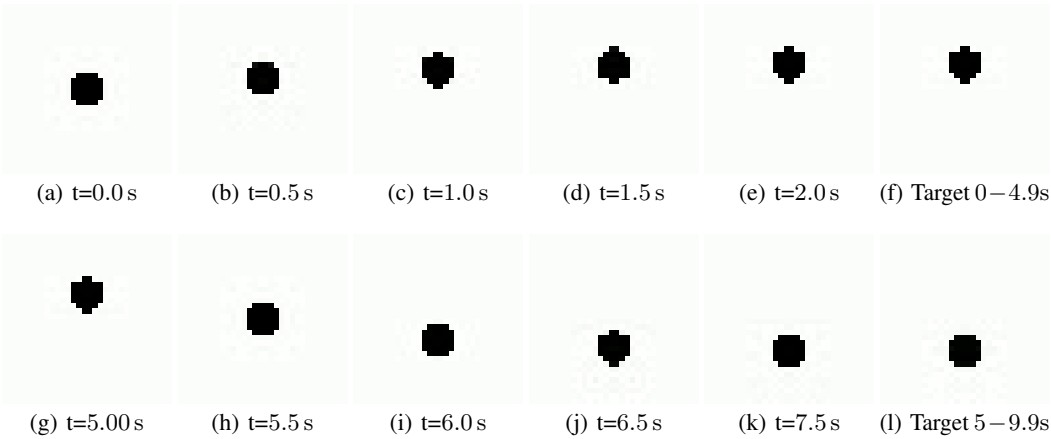

Figure 20: Sequence of closed-loop control of an actuated damped harmonic oscillator (*M-SP+F+A*) with a *P-satI-D+FF* controller based on a trained CON with $n_z = 1$. **Columns 1-4:** show the actual behavior of the closed-loop system. **Column 5:** demonstrates the target image that the control sees for all time instances in the row.

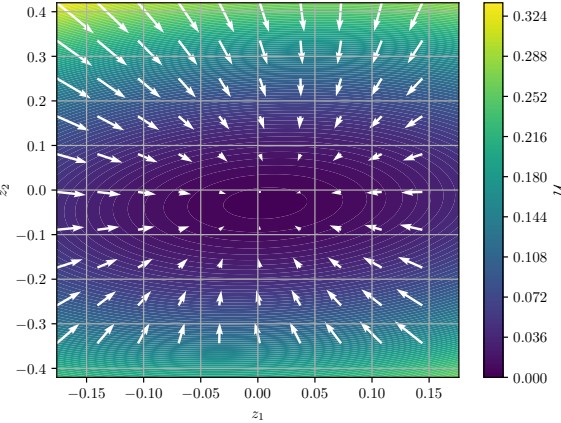

(a) Learned potential energy as a function of $z$

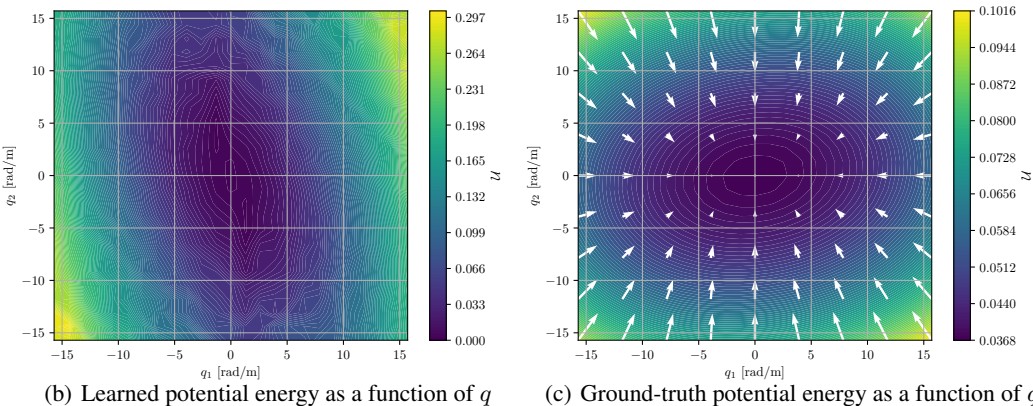

(b) Learned potential energy as a function of $q$      (c) Ground-truth potential energy as a function of $q$

Figure 21: Potential energy landscapes of a CON with $n_z = 2$ trained to learn the latent space dynamics of a continuum soft robots (simulated with two PCC segments). **Panel (a):** Here, we visualize the learned potential energy of CON using the color scale as a function of the latent representation $z = x_{\mathrm{w}} \in \mathbb{R}^2$. The arrows denote the gradient of the potential field $\frac{\partial \mathcal{U}}{\partial z}$ (i.e., the potential force), with the magnitude of the gradient expressed as the length of the arrow. **Panel (b):** Again, we display the learned potential energy of CON using the color scale, but in this case, as a function of the configuration $q \in \mathbb{R}$ of the robot (that is hidden from the model). First, we render an image $o$ of the shape of the robot for each configuration $q = [q_1 \quad q_2]^{\mathrm{T}} \in \mathbb{R}^2$. Then, we encode the image into latent space as $z = \Phi(o)$. This allows us then to compute the potential energy $\mathcal{U}(z)$ of the CON latent dynamics model. **Panel (c):** Here, we display the potential energy and its associated potential forces of the actual (i.e., simulated) system.

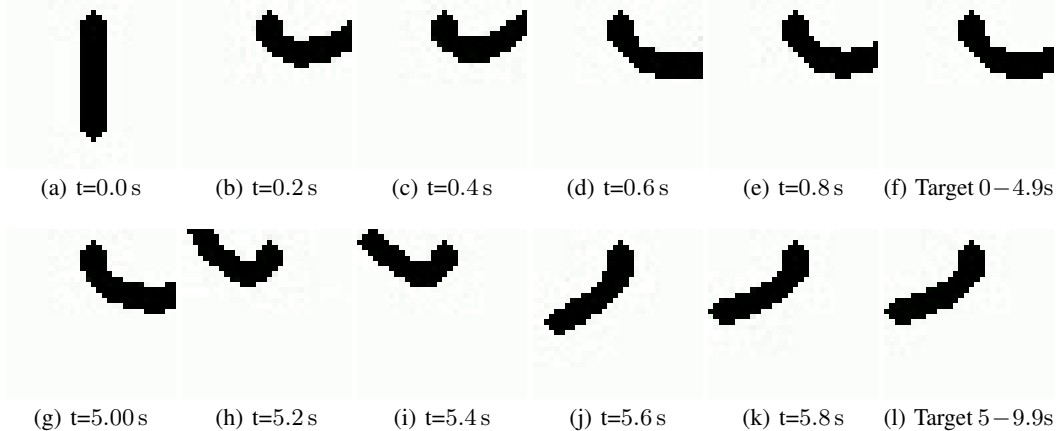

| (a) t=0.0 s | (b) t=0.2 s | (c) t=0.4 s | (d) t=0.6 s | (e) t=0.8 s | (f) Target $0-4.9$s |

| (g) t=5.00 s | (h) t=5.2 s | (i) t=5.4 s | (j) t=5.6 s | (k) t=5.8 s | (l) Target $5-9.9$s |

Figure 22: Sequence of closed-loop control of a continuum soft robot consisting of two constant curvature segments with the *P-satI-D+FF* based on a trained CON with $n_z = 2$. **Columns 1-4:** show the actual behavior of the closed-loop system. **Column 5:** demonstrates the target image that the control sees for all time instances in the row.

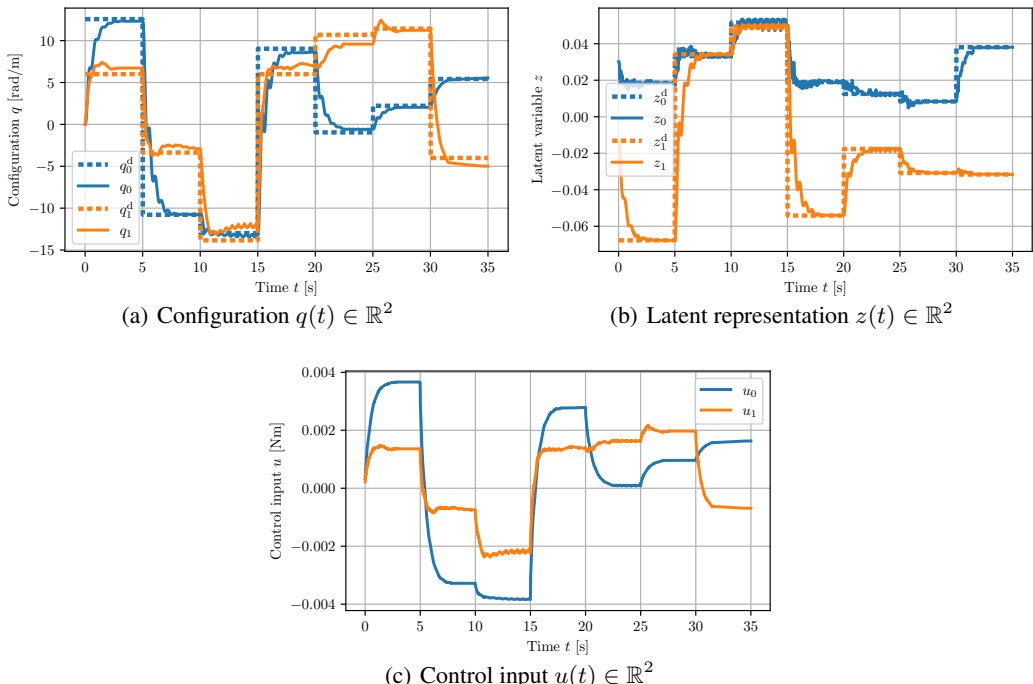

(a) Configuration $q(t) \in \mathbb{R}^2$

(b) Latent representation $z(t) \in \mathbb{R}^2$

(c) Control input $u(t) \in \mathbb{R}^2$

Figure 23: Latent-space control of a continuum soft robot (simulated using two PCC segments) following a sequence of setpoints with a pure **P-satI-D** feedback controller operating in a 2D latent space learned with the **MECH-NODE** model. The CON model weights are initialized using a **random seed of 0**. The dotted and solid lines show the reference and actual values, respectively. For each setpoint, we randomly sample a desired shape $q^{\mathrm{d}}$ and render the corresponding image $o^{\mathrm{d}}$. This image is then encoded to a target latent $z^{\mathrm{d}}$. The controller then computes a latent-space torque $F^{\mathrm{d}}$, which is decoded to an input $u$. Finally, we provide this input to the simulator, which performs a roll-out of the closed-loop dynamics. Important: The robot's configuration (i.e., the first-principle, minimal-order state) is solely used for generating a target image and simulating the closed-loop system.

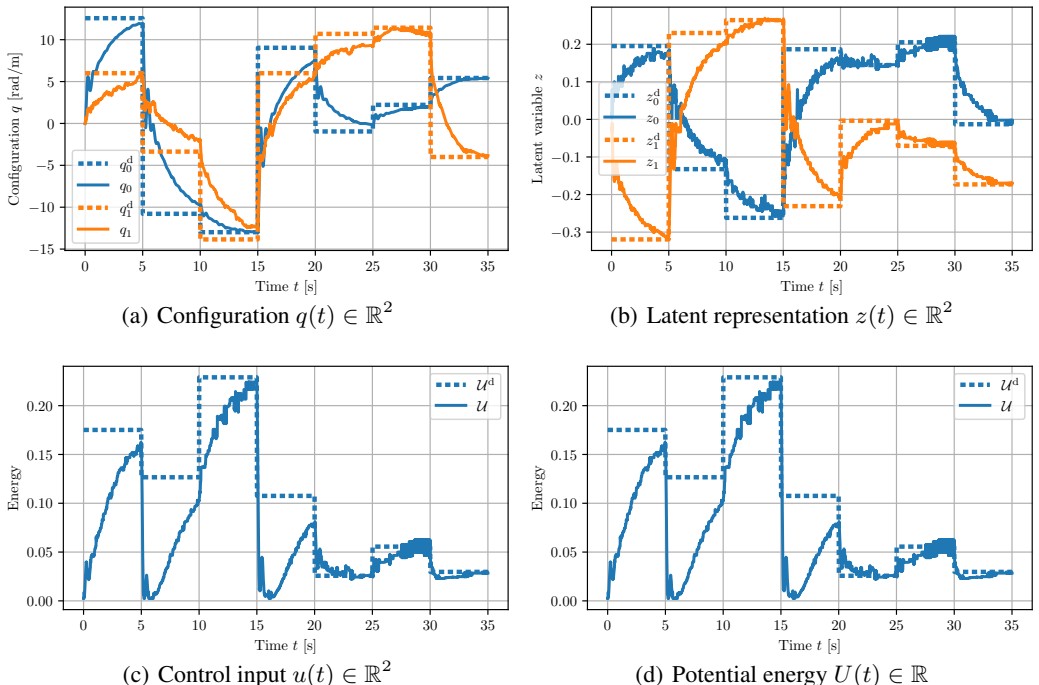

Figure 24: Latent-space control of a continuum soft robot (simulated using two PCC segments) following a sequence of setpoints with a pure **P-satI-D** feedback controller operating in a 2D latent space learned with the **CON** model. The CON model weights are initialized using a random seed of 0. The dotted and solid lines show the reference and actual values, respectively. For each setpoint, we randomly sample a desired shape $q^{\mathrm{d}}$ and render the corresponding image $o^{\mathrm{d}}$. This image is then encoded to a target latent $z^{\mathrm{d}}$. The controller then computes a latent-space torque $F^{\mathrm{d}}$, which is decoded to an input $u$. Finally, we provide this input to the simulator, which performs a roll-out of the closed-loop dynamics. Important: The robot's configuration (i.e., the first-principle, minimal-order state) is solely used for generating a target image and simulating the closed-loop system.

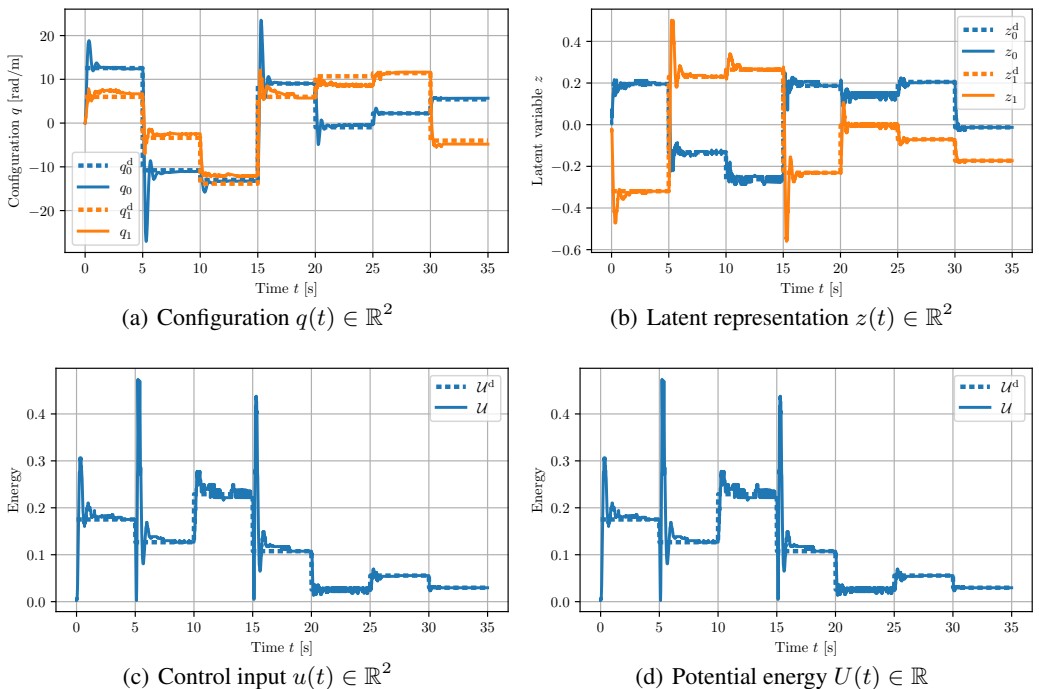

(a) Configuration $q(t) \in \mathbb{R}^2$

(b) Latent representation $z(t) \in \mathbb{R}^2$

(c) Control input $u(t) \in \mathbb{R}^2$

(d) Potential energy $U(t) \in \mathbb{R}$

Figure 25: Latent-space control of a continuum soft robot (simulated using two PCC segments) following a sequence of setpoints with a pure **P-satI-D+FF** feedback & feedforward controller operating in a 2D latent space learned with the **CON** model. The CON model weights are initialized using a random seed of 0. The dotted and solid lines show the reference and actual values, respectively. For each setpoint, we randomly sample a desired shape $q^{\mathrm{d}}$ and render the corresponding image $o^{\mathrm{d}}$. This image is then encoded to a target latent $z^{\mathrm{d}}$. The controller then computes a latent-space torque $F^{\mathrm{d}}$, which is decoded to an input $u$. Finally, we provide this input to the simulator, which performs a roll-out of the closed-loop dynamics. Important: The robot's configuration (i.e., the first-principle, minimal-order state) is solely used for generating a target image and simulating the closed-loop system.

# F Extended discussion on future applications and limitations

## F.1 Systems for which we would expect the proposed method to work

**Mechanical systems with continuous dynamics, dissipation, and a single, attractive equilibrium point.** The proposed method is a very good fit for mechanical systems with continuous dynamics, dissipation, and a single, attractive equilibrium point. In this case, the real system and the latent dynamics share the energetic structure and stability guarantees. Examples of such systems include many soft robots, deformable objects with dominant elastic behavior, and other mechanical structures with elasticity.

**Local modeling of (mechanical) systems that do not meet the global assumptions.** Even if the global assumptions of the proposed method are not met, the method can still be applied to model the local behavior around a local asymptotic equilibrium point of the system (i.e., in the case of multi-stability). For example, the method could be used to model the behavior of a robotic leg locally in contact with the ground, a cobot's interaction with its environment, etc.

## F.2 Systems for which we could envision the proposed method to work under (minor) modifications

**Mechanical systems without dissipation.** The proposed method would currently not work well for mechanical systems without any dissipation, as (a) the original system will likely not have a globally asymptotically stable equilibrium point, and more importantly, (b) we currently force the damping learned in latent space to be positive definite. However, these systems are not common in practice as friction and other dissipation mechanisms are omnipresent, and the proposed method can learn very small damping values (e.g., the mass-spring+friction system). A possible remedy could be to relax the positive definiteness of the damping matrix in the latent space, allowing for zero damping. This would allow the method to work for systems without dissipation, such as conservative systems. Examples of such systems include a mass-spring system without damping, the n-body problem, etc.

**Systems with discontinuous dynamics.** The proposed method might underperform for systems with highly discontinuous dynamics, such as systems with impacts, friction, or other discontinuities. In these cases, the latent dynamics might not capture the real system's behavior accurately, and the control performance of feedforward + feedback will very likely be worse than pure feedback. Again, the method should be able to capture local behavior well. A possible remedy for learning global dynamics could be to augment the latent dynamics with additional terms that capture the discontinuities, such as contact and friction models (e.g., stick-slip friction).

**Systems with multiple equilibrium points.** The original system having multiple equilibria conflicts with the stability assumptions underlying the proposed CON latent dynamics. In this case, as, for example, seen on the pendulum+friction and double pendulum + friction results, the method might work locally but will not be able to capture the global behavior of the system. A possible remedy could be to relax the global stability assumptions of the CON network. For example, the latent dynamics could be learned in the original coordinates of CON while allowing $W$ also to be negative definite. This would allow the system to have multiple equilibria & attractors. Examples of such systems include a robotic arm under gravity, pendula under gravity, etc.

**Systems with periodic behavior.** The proposed method will likely not work well for systems with periodic behavior, as they do not have a single, attractive equilibrium point. Examples of such systems include a mass-spring system with a periodic external force, a pendulum with a periodic external force, some chemical reactions, etc. Again, it is likely possible to apply the presented method to learning a local behavior (i.e., not completing the full orbit). A possible remedy could be to augment the latent dynamics with additional terms that capture the periodic behavior, such as substituting the harmonic oscillators with Van der Pol oscillators to establish a limit cycle or a supercritical Hopf bifurcation.

## F.3 Systems for which we would not expect the proposed method to work

**Nonholonomic systems.** The proposed method likely would not work well for nonholonomic systems, as both structure (e.g., physical constraints) and stability characteristics would not be shared between the real system and the latent dynamics. Examples of such systems include vehicles, a ball rolling on a surface, and many mobile robots.

**Partially observable and non-markovian systems.** As the CON dynamics are evaluated based on the latent position and velocity encoded by the observation of the current time step and the observation-space velocity, we implicitly assume that the system is (a) fully observable and (b) fulfills the Markov property. This assumption might not hold for partially observable systems, such as systems with hidden states or systems with delayed observations. Examples of such cases include settings where the system is partially occluded or in situations without sufficient (camera) perspectives covering the system. Furthermore, time-dependent material properties, such as viscoelasticity or hysteresis, that are present and significant in some soft robots and deformable objects are not captured by the method in its current formulation.

