# OpenReview forum: "Input-to-State Stable Coupled Oscillator Networks for Closed-form Model-based Control in Latent Space"
_NeurIPS.cc/2024/Conference — NeurIPS 2024 spotlight_

### Official Review · Reviewer_Ys99 · 2024-07-13

**Soundness:** 3
**Presentation:** 3
**Contribution:** 3
**Rating:** 7
**Confidence:** 4

**Summary:**

In this paper, the authors introduce Coupled Oscillator Networks (CONs), extending the idea of coRNNs but with the difference that the generalized force is symmetric, enabling the definition of a potential energy expression which can be exploited for energy shaping based control methods. It is proven, that this model is asymptotically stable and, given an input, input-to-state-stable. It is shown that CONs provide similar performance compared to other state of the art methods. Then, the proposed network is used together with a variational autoencoder to learn a low-dimensional representation of the dynamics of a soft robot. Finally, a PID controller with an energy shaping part is designed to steer the soft robot to a given desired state. The main contribution is the new network structure which allows us to exploit the potential energy expression.

**Strengths:**

- It was a pleasure to read the article as it is nicely written, and all steps are explained in detail and clarity.
- Computing the solution of the CON by splitting the dynamics into a linear part (where the solution can be computed in closed form) and a nonlinear part is an original idea.
- The new structure of the proposed network that allows us to exploit energy-based control methods open doors for many interesting applications and might be a significant contribution.

**Weaknesses:**

- Theorem 1 seems to be trivial as it elaborates on the (nonlinear) coupling of passive systems (damped mass-spring systems) which is always passive since it does not “produce” any energy and thus, is inherently stable. (Same for Theorem 2). In this sense, the theoretical contribution seems limited. Is there anything I missed?
- The statement in Line 266 “PID controller has several well-known drawbacks, such […] steady-state errors (in case the integral gain is chosen to be zero)” is misleading as in this case we would say it’s a PD controller.
- The main contribution seems to be the new network structure that allows to exploit the potential energy for energy shaping methods. The VAE and controller are existing methods. Thus, a more detailed elaboration on the performance and limitation of CONs would be beneficial as it is evaluated for soft robotics data sets only.

**Questions:**

- Can you add the inference time for the different methods in Table 1? If that’s not possible, could you give some general comments on the inference time of CONs?
- Figure 3 visualizes that the controller with feed-forward part leads to heavy oscillations in the systems. Is that due to a poorly tuned controller or do you see the reason in the CON model?

UPDATE: New score after discussion

**Limitations:**

Yes

---

> ### Author Rebuttal · Authors · 2024-08-06
>
> # Response to Reviewer Ys99 (R1)
>
> We thank the Reviewer very much for the kind words, for their interest in our research activities, and for the very insightful comments that they have provided.
> Because of space constraints, we respond to the request for inference times and questions about the oscillations of the controller in the global rebuttal.
>
> ## Is the stability of CON trivial?
>
> > R1: Theorem 1 seems to be trivial as it elaborates on the (nonlinear) coupling of passive systems (damped mass-spring systems), which is always passive since it does not “produce” any energy and thus, is inherently stable. (Same for Theorem 2). In this sense, the theoretical contribution seems limited. Is there anything I missed?
>
> We thank the Reviewer for their question and for allowing us to elaborate further on this topic. In short, we differ with the Reviewer's technical arguments, and, consequently, with their conclusions on the triviality of Theorem 1.
> In essence, we recognize that the Reviewer's arguments are well funded in the context of port-Hamiltonian theory. However, substantial extra properties/assumptions are needed to yield to the Reviewer's conclusion. Thus, the PH way of proving Theorem 1 would be interesting and proper, but it would definitely not result in the trivial proof the Reviewer is hinting at.
> More precisely, we point out two main points of technical disagreement, which we list below.
>
> ### (A): Passivity does not imply Stability
>
> Passivity, even in the scalar case, does not imply (global) stability. Additional assumptions on the (global) convexity of the energy landscape are needed, which would be hard to impose. As a trivial counterexample, think of a damped mass placed atop an infinite hill.
> To further illustrate this point, we propose two simple passive-but-not-globally-stable CONs that deviate slightly from the assumptions in Theorem 1 and 2.
>
> 1. Take a passive CON as stated in Eq. (2) in the scalar case with $K=1$ and $D=0.4$. Conflicting with the assumptions in Theorem 1, we select $W=-5$. Now, depending on the choice of $b$, this scalar system either has 1, 2, or 3 equilibria (see Fig. R1 in global response). All global stability guarantees are lost if the system has multiple attractors.
> 2. We study a scalar CON with $K=-1$, positive damping $D=0.4$, and $W = 0$, $b=0$ resulting in the EOM $\ddot{x} + K x + D \dot{x} = \tau$. We take the passive output $o = \dot{x}$, and we prove passivity according to Def. 6.3 (Khalil, 2001) using the storage function $V(x) = K x^2 + \dot{x}^2$: $\dot{V}(x, \tau) = \tau \dot{x} - D \dot{x}^2 \leq \tau o. $
> However, this system is unstable, as it can be easily assessed by looking at the linearization at the equilibrium. Fig. R2 reports the globally repulsive vector field, which exhibits a single unstable equilibrium at the origin.
>
> Therefore, we conclude that passivity is not sufficient to prove global asymptotic stability.
>
> ### (B): Stable harmonic oscillators do not imply Stable Networks
>
> Even if the individual systems (i.e., the harmonic oscillators) are stable, care needs to be taken when coupling them not to create an unstable network.
>
> We illustrate this with the following example: consider a CON of dimensionality two with $K = [[1.0, -1.4],[-1.4, 1.0]]$, $D = \mathrm{diag}(0.4, 0.4)$ $W = \mathrm{diag}(3, 3)$,
> It can be easily shown that the oscillators individually with the EoM $\ddot{x}_i + k x_i + d \dot{x}_i + \tanh(w x_i)= 0,$ where $k=1$, $d=0.4$, and $w=3$ are globally asymptotically stable.
>
> However, the linear stiffness matrix is negative definite, and therefore, the system is not globally asymptotically stable. This is illustrated in Fig. R3 of the global response.
>
> ## PID vs. PD
>
> > R1: The statement in Line 266 “PID controller has several well-known drawbacks, such […] steady-state errors (in case the integral gain is chosen to be zero)” is misleading as in this case we would say it’s a PD controller.
>
> We thank the Reviewer for their comment and for pointing out the mistake. We agree that this sentence is indeed badly written and confusing. In the final version of the paper, we will remove the subsentence "_steady-state errors (in case the integral gain is chosen to be zero)_".
>
> ## Performance and Limitations of CON
>
> > R1: The main contribution seems to be the new network structure that allows to exploit the potential energy for energy shaping methods. The VAE and controller are existing methods.
>
> We thank the Reviewer for their comment. We want to stress that the proposed network imposes a beneficial inductive bias for conserving global stability and ISS. Therefore, we consider the two proofs (i.e., GAS and ISS) important contributions of the paper.
> Furthermore, we also regard the closed-form approximation of the CON dynamics as an important tool for deploying oscillator networks in practice.
>
> > R1: Thus, a more detailed elaboration on the performance and limitation of CONs would be beneficial as it is evaluated for soft robotics data sets only.
>
> For this rebuttal, we have performed additional experiments involving non-soft-robotic datasets. They demonstrate that the CON network can also learn the latent dynamics of other mechanical systems, such as a mass spring, a pendulum, and a double pendulum with friction, effectively and with SOA performance. We refer to the global rebuttal for more details.
>
> In totality, the results provided in the paper and in the rebuttal show that the performance of the CON model is on par with other SOA methods while adding physical structure and stability guarantees.
> As detailed in Section 6.2, the strong assumptions needed to provide global stability guarantees are the primary limitation of the presented CON network, making it unsuitable for applications where complex attractor dynamics (e.g., multiple attractors, strange attractors, etc.) are required.
> Relaxing the stability assumptions could make the CON network also suitable for these applications.

---

> > ### Comment · Reviewer_Ys99 · 2024-08-11
> >
> > Thank your for your response! You are right that passivity does not imply *asymptotic* stability but most often stability. That said, your counterexamples do not convince me because of the following:
> >
> > 1. Example: It is pretty obvious that for a negative W the system's equilibrium can be unstable as you introduce a negative stiffness coefficient in the system. Thus, I'd still still say that the result seems trivial as the systems seems always stable (asymptotic stable) for positive damping and stiffness coefficients.
> >
> > 2. Example: Your analysis is wrong. Following Khalil, the storage function must be a "positive semidefinite function" that is obviously not the case in your example for a negative K. I don't want to look into the details, but I think that for the linear system as shown in example 2, passivity leads to (asymptotic) stability. Please provide a valid counter example if my believe is wrong.
> >
> > Finally, it's the same story for deriving stability from the single oscillators to the network. Of course you can make the stiffness matrix negative definite to achieve a non-stable network but again, if all parameters are positive (definite), meaning defined so that you would expect the system to be stable, I think that the system is stable. I'd change me opinion if you can provide a counter example where the parameters are positive (definite) but would lead to an unstable system / network.

---

> > > ### Author Response · Authors · 2024-08-13
> > > **Technical response to Reviewer Ys99: Example 2**
> > >
> > > ## Example 2: A system where the W-coordinate transformation does not help
> > >
> > > The proof underlying Theorem 1 is only (relatively) simple as the coordinate transformation $x_\mathrm{w} = W x$ helps us to identify the potential energy function in the $\mathcal{W}$ coordinates, where the hyperbolic nonlinearity operates elementwise. However, this is not always possible. For example, consider the system $\ddot{x} + K x + D \dot{x} + W^{\mathrm{T}} \tanh(Wx + b) = 0$, where $K$, $D$, $W \succ 0$ are positive definite matrices.
> > >
> > > Unfortunately, the unactuated dynamics in the \mathcal{W} coordinates are now given by
> > >
> > > $M_w \ddot{x}_w + K_w x_w + D_w \dot{x}_w+ W^T \tanh(x_w + b) = 0$,
> > >
> > > for which we cannot easily derive a potential energy function (i.e., integrate), as the hyperbolic term $W^\mathrm{T} \tanh(x_\mathrm{w} + b)$ is not (easily) separable.
> > >
> > > Remark: In contrast to Example 1, this system actually has a single equilibrium point.
> > >
> > > This example motivates why we chose the proposed CON network architecture such that we can (easily) derive the kinetic and potential energy terms, subsequently prove GAS & ISS, and perform model-based control.

---

> ### Author Response · Authors · 2024-08-13
> **General response to Reviewer Ys99**
>
> > Finally, it's the same story for deriving stability from the single oscillators to the network. Of course you can make the stiffness matrix negative definite to achieve a non-stable network but again, if all parameters are positive (definite), meaning defined so that you would expect the system to be stable, I think that the system is stable. I'd change me opinion if you can provide a counter example where the parameters are positive (definite) but would lead to an unstable system / network.
>
> We appreciate the direct nature of the Reviewer's comment, which rises indeed a compelling point with which however we still disagree. In our next response, we will try again to convince the Reviewer of our argument, although proving that something is _not trivial_ is quite a challenging task, as triviality is very subjective.
>
> So, before delving into it, we would like to take the chance of zooming out. Indeed, at the moment, we see the risk that we are debating on a relatively narrow point while it may be that we could essentially agree on several key aspects. Or at least, we believe we can work to find a common ground on these while we continue in parallel our discussion.
>
> 1. We believe that Theorem 2 is the main _stability proof_ contribution of the paper. To the best of our knowledge, we are the first in this community/problem setting to provide explicit input-to-state stability guarantees, including convergence rates into the region of attraction (Theorem 2). We believe that the proof of Theorem 2 is far from trivial.
>
> Does the Reviewer disagree on this point?
>
> 2. In this sense, we agree with the Reviewer that the proof of Theorem 1 is comparably simpler (although not trivial!) and less impactful than the one of Theorem 2.
>
> Would renaming "_Theorem 1_" as a "_Lemma_" or even as a "_Proposition_" better reflect the opinion of the Reviewer on the importance of this contribution?
>
> 3. Finally, we realize now that we did not make a great job, with the manuscript and with our previous answers, in conveying the message that: the relatively simplicity of the proof of Theorem 1 is no accident.
> The proof of Theorem 1 could be derived via _standard_ arguments from the control of mechanical systems, _because_ we designed the CON network in a particular way. So the Theorem 1 statement with its relative simplicity implicitly stresses this point.
>
> If the Reviewer agrees on this point, we are happy to include in the revision any remarks or changes that they see fit to better bring this point home.
>
> Specifically, the proof in its current form is possible because (a) a coordinate transformation exists that allows us to identify a Lyapunov candidate, and (b) the system has a potential energy. We see these two design choices a contribution by themselves as they allow the proof to be simpler as it could have been otherwise.
>
> We want to stress that already small modifications to the network formulation and the underlying assumptions in Theorem 1 would have made the proof much more difficult (or even impossible).

---

> ### Author Response · Authors · 2024-08-13
> **Technical response to Reviewer Ys99: Example 1**
>
> > Finally, it's the same story for deriving stability from the single oscillators to the network. Of course you can make the stiffness matrix negative definite to achieve a non-stable network but again, if all parameters are positive (definite), meaning defined so that you would expect the system to be stable, I think that the system is stable. I'd change me opinion if you can provide a counter example where the parameters are positive (definite) but would lead to an unstable system / network.
>
> We are now ready to go back to our technical discussion, with the discussed counterexamples. These, hopefully, also better illustrate our point #3. Essentially, our argument here is that even positive-definite stiffness & damping matrices are not sufficient to claim global asymptotic stability. This should answer the Reviewer's direct technical question.
>
> In the following, we will give two examples of modifications to the CON network (in the original coordinates) that would have, according to the Reviewer's argument, appeared to be stable but for which global stability proof would be hard to prove.
>
> ## Example 1: A system with multiple equilibria and without a valid potential energy
>
> We consider the slightly modified network dynamics $\ddot{x} + K x + D \dot{x} + W^{-1} \tanh(Wx + b) = 0$, where $K, D, W \succ 0$ are positive definite matrices.
>
> The equilibria of this system are given by the characteristic equation $K \bar{x} + W^{-1} \tanh(W \bar{x} + b) = 0$.
> For the system to be globally asymptotically stable, it would need to have a single equilibrium point. However, if we simulate the system with $K = [[5.0, -2.2], [-2.2, 1.0] \succ 0$ (positive-definite as symmetric and positive eigenvalues $0.027$ and $5.973$), $D = \mathrm{diag}(0.2, 0.2) \succ 0$, $W = [[1.0, -2.2], [-2.2, 5.0]] \succ 0$ (positive-definite as symmetric and positive eigenvalues $0.027$ and $5.973$), we notice that the system is actually bistable with two attractors at $\bar{x}_1 = [-212.6, -475.2]$ and $\bar{x}_2 = [212.6, 475.2]$. This is illustrated in the time series plot and phase portrait attached at https://anonymous.4open.science/r/neurips24-20062-rebuttal-7770. Therefore, the system is **not** globally asymptotically stable.
>
> For $\tau_\mathrm{pot} = -K x -W^{-1} \tanh(Wx + b)$ to be a valid potential force, it would need to satisfy the property $\frac{\partial \tau_\mathrm{pot}}{\partial x} = \left ( \frac{\partial \tau_\mathrm{pot}}{\partial x} \right )^\mathrm{T}$. Therefore, we derive $\frac{\partial \tau_\mathrm{pot}}{\partial x} = -K - W^{-1} \mathrm{diag}(\mathrm{sech}^2(Wx + b)) W$. Its transpose is given by $\left ( \frac{\partial \tau_\mathrm{pot}}{\partial x} \right )^\mathrm{T} = -K - W^\mathrm{T} \mathrm{diag}(\mathrm{sech}^2(Wx + b)) W^{-\mathrm{T}}$. Therefore, $\tau_\mathrm{pot}$ only stems from a potential iff $W^{-\mathrm{T}} = W \rightarrow W^\mathrm{T} W = \mathbb{I}$ (i.e., $W$ is orthogonal). This is not the case for a general positive-definite $W$.
>
> This example motivates that even if the matrices are positive-definite, the system can still have multiple equilibria (i.e., lose global asymptotic stability) and not have a valid potential energy function.

---

> ### Comment · Reviewer_Ys99 · 2024-08-13
>
> Thank you for your response! I hope that you don't see my direct nature as rude because that was not at all my intention.
>
> 1. Yes, I agree
> 2. and 3. No need to rename it but I definitely like the perspective that you intentionally designed the network so that you can find a "simple" proof. That would be great to emphasize in the paper.
>
> Especially thank you for the new example. I appreciate your work here and now understand better the details. I will raise my score by 3 points.

---

### Official Review · Reviewer_W9L3 · 2024-08-01

**Soundness:** 4
**Presentation:** 3
**Contribution:** 3
**Rating:** 8
**Confidence:** 3

**Summary:**

- This paper proposes Coupled Oscillator Networks (CONs); Networks consisting of coupled one-dimensional dampened harmonic oscillators, coupled through a neuron-like connection.
- It is shown that under some constraints, the unforced coupled oscillators have a single, globally stable equilibrium and the forced system is globally ISS stable.
- As the CON has no closed-form solution, the authors propose to split the solution in one decoupled linear, analytically solvable term and a residual non-linear term which is. For this system, a closed-form approximation is derived (CON-CFA).
- High dimensional observations are encoded via a VAE into a compressed latent space that is trained to reconstruct future observations that are predicted via a CON latent dynamics model. This system is trained jointly.
- “CONs are an ideal fit for learning latent dynamics as they guarantee that the latent states stay bounded.”
- The approaches (CON with two different latent dimensions,S and M, and CON-CFA) are evaluated on a simulated soft-robots dataset and benchmarked against other latent-space dynamics methods based on neural ODEs ore autoregressive models.
- When predicting system state, CON-M performs on par with sota methods, CON-S and CON-CFA perform almost as good, while cheaper computationally.
- For control, potential-shaping strategies are combined with PID, resulting in a latent-space control law. A forcing decoder is then trained to to predict control inputs. This approach strongly outperforms a simple latent space PID controller.

**Strengths:**

- Very interesting and (to my knowledge) novel approach of controlling robots in a latent dynamics space, modeled through a network of coupled oscillators with provable stability guarantees.
- Extensive proofs and detailed descriptions of implementation.
- Promising experimental evaluation.
- Very concise and complete presentation of their approach.

**Weaknesses:**

- Limited experimental evaluation on a single simulated soft robot.
- It is unclear how the obtained results generalize to other robots, in particular with less smooth contact dynamics.

**Questions:**

- Have you tried the method on other (non-soft) robots? Do you have any intuition on how your method would perform on environments with many contact forces?
- Have you considered non-physical systems?

**Limitations:**

Many limitations of the method are clearly stated.

---

> ### Author Rebuttal · Authors · 2024-08-06
>
> # Response to Reviewer W9L3 (R2)
>
> We thank the Reviewer for the careful reading and the encouraging comments.
> In the following, we will respond to the questions raised by the reviewer, which also relate to the weaknesses mentioned by the reviewer.
>
> ## Application of CON to non-soft robots
>
> > R2: Have you tried the method on other (non-soft) robots?
>
> We thank the Reviewer for inquiring about results on non-soft robots. We provide additional results on various mechanical/robotic systems in the global rebuttal.
>
> > R2: Do you have any intuition on how your method would perform on environments with many contact forces?
>
> First, we want to stress that we have not conducted any experiments with contact-rich systems yet and reserve this interesting challenge for future work.
> We hypothesize that the highly discontinuous dynamics of contact-rich systems could present a challenge for the method in its present form.
> Still, we envision that the method could be augmented to handle such systems. For example, we could add stick-slip friction or similar mechanisms to the CON dynamics to increase their expressiveness while maintaining the physical structure.
> Furthermore, contact-rich systems often have multiple equilibria, which would be a violation of the stability conditions introduced in this work. As touched on in Section 6 of the paper, these strong global stability guarantees would probably need to be relaxed.
>
> ## Application of CON to non-physical systems
>
> > R2: Have you considered non-physical systems?
>
> We thank the Reviewer for their interest in this topic. While we do think that the CON / CFA-CON models could be potentially useful in other applications where strong stability guarantees are required, we strive to focus in this paper on learning the dynamics of physical/mechanical systems as we can here leverage (a) the shared stability characteristics between the original and the latent space systems, and (b) exploit the mechanical structure of the CON model for control.
> To make this focus clear, we will, in the final paper, modify the sentence starting on line 69 to (with changes marked in **bold**):
>
> _We resolve all the above-mentioned challenges by proposing Coupled Oscillator Networks (CONs), a new formulation of a coupled oscillator network that is inherently Input-to-State Stability (ISS) stable, **for learning the dynamics of physical systems,** and subsequently exploiting its structure for model-based control in latent space._
>
> This work was specifically focused on how to impose structural biases that enforce second order Lagrangian structure and stability properties. However, we believe that the proposed model can be used more broadly, beyond mechanical systems, whenever strong stability guarantees are needed. This will be the focus of future research.

---

> > ### Comment · Reviewer_W9L3 · 2024-08-13
> >
> > Thank you for the response.
> >
> > **Application of CON to non-soft robots**
> >
> > I agree with the authors' view that their method might underperform for systems with highly discontinuous dynamics. Since this paper is framed as a method for prediction and control of systems, with soft robots being only one application area (rather than a method for the control of soft robots), I would appreciate a lucid analysis of classes of systems/application areas in which the method might perform strongly/weakly (including a clear reasoning why; and potentially a remedy left to future work). Please consider including this, both from a systems theory and an application standpoint into the paper.
> >
> > **Application of CON to non-physical systems**
> >
> > Thank you for explicitly limiting the application area to physical systems.

---

> > > ### Author Response · Authors · 2024-08-13
> > > **Response to Reviewer W9L3 Part 2**
> > >
> > > ## Planned additions to the Appendix
> > >
> > > Furthermore, we could envision adding a more detailed discussion (similar to the one below), or alternatively a table, to the Appendix:
> > >
> > > ### Systems for which we would expect the proposed method to work
> > >
> > > - **Mechanical systems with continuous dynamics, dissipation, and a single, attractive equilibrium point:** The proposed method is a very good fit for mechanical systems with continuous dynamics, dissipation, and a single, attractive equilibrium point. In this case, the real system and the latent dynamics share both the energetic structure and stability guarantees. Examples of such systems include many soft robots, deformable objects with dominant elastic behavior, other mechanical structures with elasticity, etc.
> > >
> > > - **Local modeling of (mechanical) systems that do not meet the global assumptions:** Even if the global assumptions of the proposed method are not met, the method can still be applied to model the local behavior around a local asymptotic equilibrium point of the system (i.e., in the case of multi-stability). For example, the method could be used to model the behavior of a robotic leg locally in contact with the ground, a cobot's interaction with its environment, etc.
> > >
> > > ### Systems for which we could envision the proposed method to work under (minor) modifications
> > >
> > > - **Mechanical systems without dissipation:** The proposed method would currently not work well for mechanical systems without any dissipation, as (a) the original system will likely not have a globally asymptotically stable equilibrium point, and more importantly, (b) we currently force the damping learned in latent space to be positive definite. However, these systems are not common in practice as friction and other dissipation mechanisms are omnipresent, and the proposed method can learn very small damping values (e.g., the mass-spring+friction system).
> > > A possible remedy could be to relax the positive definiteness of the damping matrix in the latent space, allowing for zero damping. This would allow the method to work for systems without dissipation, such as conservative systems. Examples of such systems include a mass-spring system without damping, the n-body problem, etc.
> > >
> > > - **(Mechanical) systems with discontinuous dynamics:** The proposed method might underperform for systems with highly discontinuous dynamics, such as systems with impacts, friction, or other discontinuities. In these cases, the latent dynamics might not capture the real system's behavior accurately, and the control performance of feedforward + feedback will very likely be worse than pure feedback. Again, the method should be able to capture local behavior well. A possible remedy for learning global dynamics could be to augment the latent dynamics with additional terms that capture the discontinuities, such as contact and friction models (e.g., stick-slip friction).
> > >
> > > - **(Mechanical) systems with multiple equilibrium points:** The original system having multiple equilibria conflicts with the stability assumptions underlying the proposed CON latent dynamics. In this case, as, for example, seen on the pendulum+friction and double pendulum + friction results, the method might work locally but will not be able to capture the global behavior of the system. A possible remedy could be to relax the global stability assumptions of the CON network. For example, the latent dynamics could be learned in the original coordinates of CON while allowing $W$ also to be negative definite. This would allow the system to have multiple equilibria & attractors. Examples of such systems include a robotic arm under gravity, pendula under gravity, etc.
> > >
> > > - **(Mechanical) systems with periodic behavior:** The proposed method will likely not work well for systems with periodic behavior, as they do not have a single, attractive equilibrium point. Examples of such systems include a mass-spring system with a periodic external force, a pendulum with a periodic external force, some chemical reactions, etc. Again, it is likely possible to apply the presented method to learning a local behavior (i.e., not completing the full orbit). A possible remedy could be to augment the latent dynamics with additional terms that capture the periodic behavior, such as substituting the harmonic oscillators with Van der Pol oscillators to establish a limit cycle or a supercritical Hopf bifurcation.

---

> > > ### Author Response · Authors · 2024-08-13
> > > **Response to Reviewer W9L3 Part 3**
> > >
> > > ### Systems for which we would not expect the proposed method to work
> > >
> > > - **Nonholonomic systems:** The proposed method likely would not work well for nonholonomic systems, as both structure (e.g., physical constraints) and stability characteristics would not be shared between the real system and the latent dynamics. Examples of such systems include vehicles, a ball rolling on a surface, and many mobile robots.
> > >
> > > - **Partially observable and non-markovian systems:** As the CON dynamics are evaluated based on the latent position and velocity encoded by the observation of the current time step and the observation-space velocity, we implicitly assume that the system is (a) fully observable, and (b) fulfills the Markov property. This assumption might not hold for partially observable systems, such as systems with hidden states or systems with delayed observations. Examples of such cases include settings where the system is partially occluded or in situations without sufficient (camera) perspectives covering the system. Furthermore, time-dependent material properties, such as viscoelasticity or hysteresis, that are present and significant in some soft robots and deformable objects are not captured by the method in its current formulation.

---

> ### Author Response · Authors · 2024-08-13
> **Response to Reviewer W9L3 Part 1**
>
> > _RW9L3:_ I agree with the authors' view that their method might underperform for systems with highly discontinuous dynamics. Since this paper is framed as a method for prediction and control of systems, with soft robots being only one application area (rather than a method for the control of soft robots), I would appreciate a lucid analysis of classes of systems/application areas in which the method might perform strongly/weakly (including a clear reasoning why; and potentially a remedy left to future work). Please consider including this, both from a systems theory and an application standpoint into the paper.
>
> We appreciate the Reviewer's feedback and suggestion. We fully agree with the Reviewer that specific categories or examples of systems for which the proposed method might be suitable or unsuitable should be discussed in the paper. In the following, we will detail the planned changes.
>
> ## Planned changes to the Introduction
>
> First, we will add the following sentence to the end of the introduction (i.e., Section 1):
>
> > The proposed methodology is particularly well-suited for learning the latent dynamics of mechanical systems with continuous dynamics, dissipation, and a single, attractive equilibrium point. Examples of such systems include many soft robots, deformable objects with dominant elastic behavior, other mechanical structures with elasticity, or locally for other mechanical systems such as robotic manipulators, legged robots etc. For these systems, we can fully leverage the structural prior of the proposed latent dynamics including the integrated stability guarantees. If the system is actuated, the learned dynamics can be subsequently exploited for model-based control, as demonstrated in Section 5.
>
> ## Planned changes to the _Limitations_ section
>
> We will also extend the _Limitations_ section (i.e., Section 6):
>
> > **Limitations.** While we think our proposed method shows great potential and opens interesting avenues for future research, there are currently certain limitations. For example, the proposed method of learning (latent) dynamics implicitly assumes that the underlying system adheres to the Markov property (e.g., the full state of the system is observable), that it can be approximated by a system with mechanical structure, and that it has an isolated, globally asymptotically stable equilibrium. This is, for example, the case for many mechanical systems (e.g., some continuum soft robots, deformable objects, and elastic structures) with continuous dynamics, convex elastic behaviour, dissipation and whose time-dependent effects (e.g., viscoelastcity, hysteresis) are negligible. Even if these conditions are not met globally, the method can be applied to model the local behavior around an asymptotic equilibrium point of the system (e.g., robotic manipulators, legged robots) with added stability benefits for out-of-distribution samples. Alternatively, the method could be extended to relax some of these assumptions, e.g., by allowing for multiple equilibria, zero damping, or by incorporating additional terms to capture discontinuous dynamics (e.g., stick-slip models) or period motions (e.g., limit cycles such as the Van der Pol oscillator). For some physical systems, such as nonholonomic systems, partially observable systems, or systems with non-Markovian properties, the proposed method might not be suitable. Examples of such systems include mobile robots and systems with hidden states or delayed observations. Finally, the application of this method to non-physical systems, such as financial systems, social networks, or other complex systems, is out of the scope of this work. _THE DISCUSSION OF OTHER LIMITATIONS OF THIS PAPER WHICH ARE ALREADY MENTIONED IN THE INITIAL PAPER SUBMISSION CONTINUES HERE..._

---

### Author Rebuttal · Authors · 2024-08-06

# Global Rebuttal

## Performance of CON on non-soft-robotic datasets (R1 & R2)

> R1: Thus, a more detailed elaboration on the performance and limitations of CONs would be beneficial as it is evaluated for soft robotics data sets only.

> R2: Have you tried the method on other (non-soft) robots?

We thank both reviewers for their interest in the performance of CON on non-soft-robotic datasets.
For this rebuttal, we compared the performance of CON against the baseline method on three additional mechanical, non-soft-robotic datasets: a mass-spring with friction (_M-SP+F_) (i.e., a damped harmonic oscillator), a single pendulum with friction (_S-P+F_), and a double pendulum with friction (_D-P+F_). These datasets are based on an interesting publication by Botev et al. (2021) [25], which appeared in the _NeurIPS 2021 Track on Datasets and Benchmarks_ and benchmarks various models for learning latent space dynamics.

The results, which we will refer to as Table R1 of the global response PDF, show that the NODE model slightly outperforms the CON network on the _M-SP+F_ and _S-P+F_. However, as the datasets do not consider system inputs, we can remove the input mapping from all models (e.g., RNN, GRU, coRNN, CON, and CFA-CON). With that adjustment, the CON network has the fewest parameters among all models and particularly two orders of magnitude less than the NODE model. Therefore, we find it very impressive that the CON network is roughly on par with the NODE model.
For the _D-P+F_ dataset, we can conclude that the CFA-CON model offers the best performance across all methods.
Finally, most of the time, the CON & CFA-CON networks outperform the other baseline methods that have more trainable parameters.

## Inference time of CON (R1)

> R1: Can you add the inference time for the different methods in Table 1? If that’s not possible, could you give some general comments on the inference time of CONs?

We thank the reviewer for their question about the inference time of the various methods. We note that the number of training steps per second of all methods included in Table 1 was already reported in the original submission in Table 4 of Appendix D.

For this rebuttal, we performed additional evaluations of the inference time (i.e., without computation of loss function and gradient descent) of the various models and report the results in Table R2 of the global response PDF.

## Oscillations of the controller with FF term. (R1)

> R1: Figure 3 visualizes that the controller with a feed-forward part leads to heavy oscillations in the systems. Is that due to a poorly tuned controller or do you see the reason in the CON model?

We thank the Reviewer for their question and for raising the topic. When tuning the gains PID-like controllers, a trade-off naturally exists between transient behavior (e.g., oscillations and overshooting) and response time.
In this case, we chose gains that minimized the response time but allowed for stable behavior. The oscillations are caused by a combination of (a) the underdamped nature of the system and (b) the magnitude of the proportional term.
Importantly, to have a fair comparison, we kept the gains of the feedback controller the same for both the _P-satI-D_ and _P-satI-D + FF_ cases. A higher proportional term is beneficial for the response time (and the performance) of the _P-satI-D_,
while it leads to overshooting and oscillations in the _P-satI-D + FF_ case. We stress that this is not an inherent problem of the feedback controller but can be mitigated by tuning the feedback gains differently.
For this rebuttal, we tuned a controller with reduced proportional and increased damping term and the results, included as Fig. R4 in the global response PDF, show that the oscillations and overshooting are both significantly reduced.

---

### Decision · Program_Chairs · 2024-09-25

**Decision:**

Accept (spotlight)

**Comment:**

The paper presents a very interesting combination of traditional control methods and machine learning, applied to robotic control (non-soft robot results were added at rebuttal time). Unfortunately there were only 2 reviews but both are very enthusiastic and I concur with their assessment. The authors should make sure that the final version reflects the clarifications provided in the rebuttal.